

# Does objective cluster analysis serve as a useful precursor to seasonal precipitation prediction at local scale? Application to western Ethiopia

Ying Zhang[1], Semu Moges[2], Paul Block[1]

[1]Department of Civil and Environmental Engineering, University of Wisconsin-Madison, Madison, 53706, USA
[2]School of Civil and Environmental Engineering, Addis Ababa University, Addis Ababa, 1000, Ethiopia

*Correspondence to*: Paul Block (paul.block@wisc.edu)

**Abstract.** Prediction of seasonal precipitation can provide actionable information to guide management of various sectoral activities. For instance, it is often translated into hydrological forecasts for better water resources management. However, many studies assume homogeneity in precipitation across an entire study region, which may prove ineffective for operational and local-
level decisions, particularly for locations with high spatial variability. This study proposes advancing local-level seasonal precipitation predictions by first conditioning on regional-level predictions, as defined through objective cluster analysis, for western Ethiopia. To our knowledge, this is the first study predicting seasonal precipitation at high resolution in this region, where lives and livelihoods are vulnerable to precipitation variability given the high reliance on rain-fed agriculture and limited water resources infrastructure. The combination of objective cluster analysis, spatially high-resolution prediction of seasonal
precipitation, and a modeling structure spanning statistical and dynamical approaches makes clear advances in prediction skill and resolution, as compared with previous studies.





# 1 Primer on prediction models and cluster analysis

Seasonal precipitation prediction can provide potentially actionable information to guide management of various sectoral activities. For instance, precipitation prediction is often translated into a hydrological forecast, which can be used to optimize reservoir operations, provide early flood or drought warning, inform waterway navigation, etc. As a primary input to soil
moisture, precipitation prediction is also essential to agricultural management – farmers can take advantage of anticipated preferable climatic conditions or avoid unnecessary costs under expected undesirable conditions. Two types of models are commonly used for seasonal precipitation prediction: statistical and dynamical. Dynamical models, such as general circulation models (GCMs), include complex physical climate processes, while statistical models are purely data-driven, relating observations and hydroclimate variables directly.

While both modeling approaches have produced skillful seasonal predictions for a variety of applications (e.g. Barrett, 1993; Hammer et al., 2000; Shukla et al., 2016), each has noteworthy drawbacks. Dynamical models often require a significant amount of time to build and parameterize, whereas statistical models require considerably fewer resources (e.g. Mutai et al., 1998; Gissila et al., 2004; Block and Rajagopalan, 2007; Diro et al., 2008; Diro et al., 2011b; Block and Goddard, 2012). Dynamical
models also suffer from their high sensitivity to initial uncertain conditions, particularly given a long lead time. Consequently, a number of simulations are typically produced, each with unique initial conditions, to provide a range of possible outcomes (e.g. Roeckner et al., 1996; Anderson et al., 2007). Furthermore, the outputs from dynamical models often require additional bias correction, typically using statistical methods, to better match observations (e.g. Ines and Hansen, 2006; Block et al., 2009; Teutschbein and Seibert, 2012). Statistical models, on the other hand, are highly dependent on substantial high-quality historical
data to capture hydroclimatic patterns and signals, particularly extreme conditions, which is often not available. Additionally, statistical models are often linear by construction, and may not well capture non-linear complex interactions and feedbacks. The physical nature of dynamical models, however, allows for prediction under non-stationary conditions, and also when insufficient historical data is available, whereas statistical models, by construction, typically rely on stationary relationships (Schepen et al., 2012).


Given these features of both model types for seasonal prediction, many studies have explored the combination of statistical and dynamical model outputs (e.g. Coelho et al., 2004; Block and Goddard, 2012; Schepen et al., 2012). In general, the combined predictions are typically superior to individual models, however this is not always the case, and is dependent on location, predicted seasons, lead time, comparable model skill, etc. (e.g. Metzger et al., 2004).


The spatial extent selected for statistical seasonal prediction is critical. It is not uncommon to simply assume homogeneity in precipitation across an entire study region; however, this limits addressing potential spatial variability. While this may be suitable for very broad regional planning, it is often ineffectual for operational and local-level decisions, particularly for locations with high spatial variability. This prompts the need for delineation of sub-regional scale homogeneous regions, often
defined through cluster analysis. Defining these homogeneous regions, however, is a non-trivial process. There are a variety of methods to delineate homogeneous regions, including comparing annual cycles (e.g. unimodal and bimodal distributions in precipitation) between stations (or grid-cells), comparing station correlations with regional averages, applying empirical orthogonal functions (EOF), various clustering techniques, and other methods of increasing complexity (e.g. Parthasarathy et al., 1993; Mason, 1998; Landman and Mason, 1999; Gissila et al., 2004; Diro et al., 2008; Diro et al., 2011b; Singh et al., 2012). In
addition, delineation of the sub-region size is also important to consider. Smaller sized homogeneous sub-regions do not



necessarily lead to improved predictions, as the noise at overly small scales can dominate any real signals representing spatial coherency of precipitation. For additional discussion regarding defining homogeneous sub-regions and cluster analysis, the reader is referred to Zhang et al. (2016) and Badr et al. (2015).

## 2 Application to western Ethiopia and objectives of the study

Ethiopia is vulnerable to fluctuations in precipitation given its reliance on rain-fed agriculture and limited water resources infrastructure. The majority of agriculture and infrastructure are in western Ethiopia, where water resources are relatively rich compared to other parts of the country (Awulachew et al., 2007). However, precipitation is highly varied temporally and spatially in the *Kiremt* season – the major rainy season spanning June through September (JJAS) – making skillful seasonal predictions challenging, particularly at local scales (e.g. Gissila et al., 2004; Block and Rajagopalan, 2007). Operational precipitation
predictions in Ethiopia have been issued by its National Meteorological Agency (NMA) since 1987 using an analog methodology (i.e. locating a similar climate scenario in the past – an analog – to predict future conditions), however this approach has produced only marginally skillful outcomes (Korecha and Sorteberg, 2013). For NMA's prediction, the country is divided into eight homogeneous regions for which NMA produces independent predictions. Similarly, others have also addressed seasonal prediction in Ethiopia contingent on both temporal and spatial precipitation patterns. Gissila et al. (2004) divide
Ethiopia into four regions conditioned on the seasonal cycle and interannual variability coherence prior to prediction, while Diro et al. (2009) apply a similar approach but with dynamic cluster boundaries, allowing for different delineations for each rainy season. Segele et al. (2015) consider statistical precipitation predictions across Ethiopia as a whole, as well as for northeastern Ethiopia and at two Ethiopian cities. Block and Rajagopalan (2007) predict the average summertime (JJAS) precipitation over the upper Blue Nile basin – a region they claim is homogenous at inter-annual time scales. Korecha and Barnston (2007) select
an all-Ethiopia average precipitation index to characterize predictability broadly, with minimal attention to operational-level predictions. All of these studies focus on predicting regional average precipitation based on subjective clustering methods applying a limited number of stations or coarsely gridded data; no local predictions at a finer spatial scale are explored.

This study moves forward by exploring local-level seasonal precipitation prediction through the use of regional-level predictions,
based on previous cluster analyses over western Ethiopia (Zhang et al., 2016). The advantages of defining homogeneous regions for seasonal prediction at operational (small) scales will be demonstrated by comparing approaches with and without undertaking a cluster analysis *a priori*. The combination of objective cluster analysis, spatially high-resolution prediction of seasonal precipitation, and a modeling structure spanning statistical and dynamical approaches makes clear advances compared to previous studies.

## 3 Modeling high-resolution seasonal prediction


To evaluate high-resolution seasonal precipitation prediction comparing with versus without cluster analysis *a priori*, statistical models are developed and compared with bias-corrected dynamical model predictions. Four scenarios are evaluated based on two criteria – (1) *clustered* vs. *non-clustered* and (2) *direct* vs. *indirect*. In the *clustered* case, predictions are produced for each homogeneous region (cluster) given a unique set of predictors. In the *non-clustered* case, the entire study region is considered as
one cluster and thus only one set of predictors is utilized for predictions. For the *direct* case, precipitation is predicted directly at the local level (grid scale); for the *indirect* case, the average precipitation within each homogeneous region is predicted first (as





an intermediary), and then regressed to local-level (grid scale) predictions. Combinations of the two criteria form four scenarios – *clustered direct* (C-D), *non-clustered direct* (NC-D), *clustered indirect* (C-I), and *non-clustered indirect* (NC-I) predictions.

### 3.1 Cluster analysis

Using a k-means clustering technique, western Ethiopia – the major agricultural region of the country – is divided into eight
homogeneous regions (Fig. 1), conditioned on the interannual variability of total precipitation in JJAS, the same variable that is to be predicted. Precipitation is based on a 0.1˚×0.1˚ gridded precipitation dataset from NMA (Dinku et al., 2014), consisting of 7320 grid-cells across 1983–2011 (29 years). Given the high-resolution gridded dataset, k-means clustering is performed for a range of predefined numbers of clusters; the optimal number of clusters is identified by comparing the within-cluster sum of square errors (WSS). During the clustering process, each grid-cell is assigned and reassigned to clusters until the WSS is
minimized. This does not require any subjective delineation or manual delineation of boundaries between clustered stations or grid-cells; instead, an automated and objective delineation is performed. Readers are referred to Zhang et al. (2016) for more details.

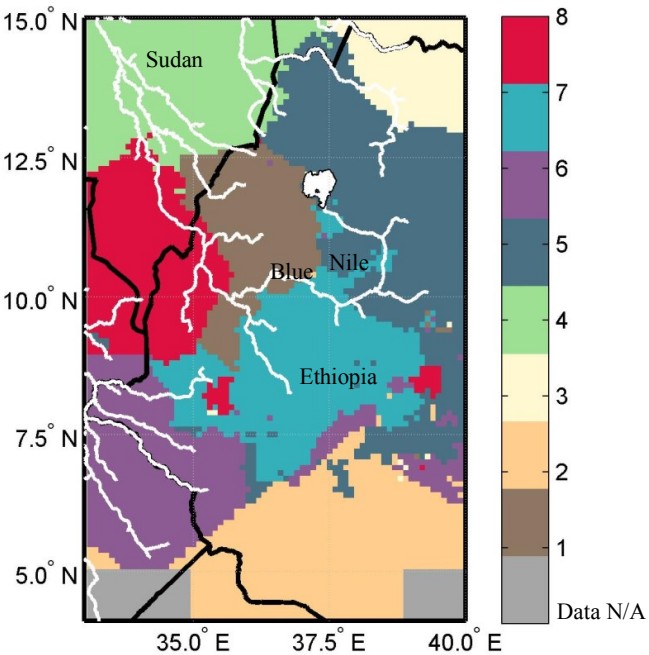

**Figure 1: Regionalization map of 8 homogeneous regions marked by different colors, with country boundary and river profile. After**
**Zhang et al. (2016)**

### 3.2 Statistical modeling approach

Multiple linear regression (MLR) is favored by many as a statistical modeling approach given its well-developed theory, simple model structure, efficient processing, and often skillful outcomes (e.g. Omondi et al., 2013; Camberlin and Philippon, 2002; Diro et al., 2008). As mentioned, only a few studies have focused on seasonal precipitation prediction in Ethiopia (Gissila et al., 2004;
Block and Rajagopalan, 2007; Korecha and Barnston, 2007; Diro et al., 2008; Diro et al., 2011b; Segele et al., 2015), and almost





all of them include the applications of MLR. This study also applies MLR to predict seasonal precipitation, yet differentiates from other studies by applying predictions to pre-defined homogeneous regions and further translating to local-level predictions.

Large-scale climate variables are often evaluated as potential predictors in statistical seasonal precipitation prediction models, commonly including sea surface temperatures (SST) in the equatorial Pacific Ocean representing the well-known of the El Nino-Southern Oscillation (ENSO) (Stone et al., 1996). For Ethiopia, the ENSO phenomenon is considered a significant indicator of precipitation variability, particularly in the main JJAS rainy season (e.g. NMSA, 1996; Camberlin, 1997; Bekele, 1997; Segele and Lamb, 2005; Diro et al., 2011a; Elagib and Elhag, 2011). In addition to ENSO, the effect of Indian Ocean SST and regional atmospheric pressure systems such as the St. Helena, Azores, and Mascarene Highs also have notable influence on Ethiopia's precipitation variability (e.g. Kassahun, 1987; Tadesse, 1994; NMSA, 1996; Shanko and Camberlin, 1998; Goddard and Graham, 1999; Latif et al., 1999; Black et al., 2003; Segele and Lamb, 2005). Consequently, season-ahead (March-May) or month-ahead (May) large-scale climate variables that are physically relevant in potentially modulating moisture transport to the basin (or cluster) are selected as potential predictors. Four climate variables are selected here for further evaluation based on outcomes of the aforementioned prediction studies: SST, sea level pressure (SLP), geopotential height (GH) at 500mb, and surface air temperature (SAT). All climate variables are from the National Centers for Environmental Prediction and National Center for Atmospheric Research (NCEP/NCAR) reanalysis dataset (Kalnay et al., 1996) at a 2.5°×2.5° grid scale.

Predictor selection and statistical modeling are developed according to the following five steps – for the region as a whole (non-clustered) and for each pre-defined cluster (Fig. 2):

(1) Precipitation observations for JJAS averaged across the region and each cluster are spatially correlated independently with each global climate variable (e.g. Fig. 3).

(2) For each spatial correlation, regions with justifiable climatic associations and statistically significant correlations at the 95% level are identified and selected (Table 1).

(3) For each climate variable region selected (Table 1), data within the region are spatially averaged for 1983-2011 (defined as "pre-predictors").

(4) Pre-predictors are combined and transformed (for the region or each cluster separately) through principal component analysis (PCA; Jolliffe, 2002).

(5a) The top principal components (PCs) from the PCA are used as predictors – the direct inputs into the MLR model, otherwise known as the principal component regression (PCR). For the *direct* case, PCR is used to directly predict the grid-level precipitation; for the *indirect* case, PCR is used to predict the intermediate cluster-level precipitation.

(5b) For the *indirect* case only, cluster-level predictions are regressed to the grid-level.

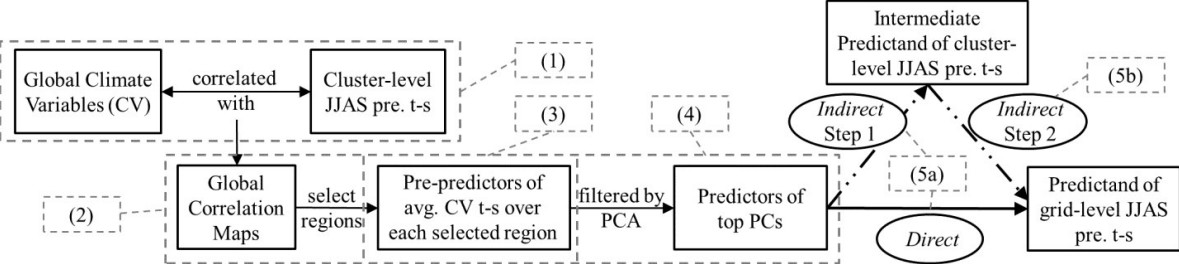

**Figure 2: Flow chat of data processing for predictors into the statistical model. Numbers framed by dash lines correspond to the procedures listed in the context. Note: pre. – precipitation, t-s – time-series, avg. – average.**

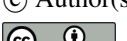



**Table 1: Climate Variables (C.V.) in May over different regions for each cluster (C1 ~ C8) and region as a whole (non-cluster) used as predictors, with corresponding correlation between the climate variable averaged over the region and the cluster-level JJAS seasonal total precipitation time-series shown (only cells with correlation values shown are used as pre-predictors). After Zhang et al. (2016)**

| C.V. | SST | | | | SLP | | | | | | GH at 500mb | | | | | | SAT | # of pre-predictors |
|---|---|---|---|---|---|---|---|---|---|---|---|---|---|---|---|---|---|---|
| Region | EP | NI | SI | E/SA | LO | EP | AH | SH | MH | AM | LO | EP | AH | SH | MH | AM | LO | |
| C1 | -0.46 | -0.46 | -0.55 | -0.48 | | | 0.45 | 0.45 | -0.51 | | | 0.52 | | | | | 0.50 | 9 |
| C2 | | -0.43 | -0.51 | -0.43 | | | | 0.58 | | | | 0.55 | | | 0.50 | | | 6 |
| C3 | | | -0.58 | -0.59 | | | | -0.50 | | | | | 0.57 | | | | | 4 |
| C4 | | | -0.60 | | | | | | | -0.58 | | | 0.49 | | | | | 3 |
| C5 | -0.52 | 0.52 | -0.54 | 0.59 | -0.50 | 0.61 | | | | | | 0.67 | 0.67 | 0.54 | 0.53 | | 0.67 | 11 |
| C6 | | 0.56 | -0.51 | | | 0.64 | | | | | | 0.66 | | -0.51 | | | | 5 |
| C7 | | 0.63 | -0.59 | 0.65 | | | | 0.44 | | | | | 0.65 | 0.65 | | | 0.44 | 7 |
| C8 | | | -0.44 | 0.53 | -0.46 | 0.55 | | | | | | | 0.63 | | | | 0.48 | 6 |
| Non-cluster | -0.47 | | -0.47 | -0.52 | | 0.47 | | | | -0.41 | | | 0.54 | 0.58 | 0.52 | | 0.52 | 9 |

Note: EP - equatorial Pacific region, NI - North Indian Ocean, SI - South Indian Ocean, E/SA - Equatorial/South Atlantic Ocean
LO - local region, AH - Azores High, SH - St Helena High, MH - Mascarene High, AM - SW Asian Monsoon




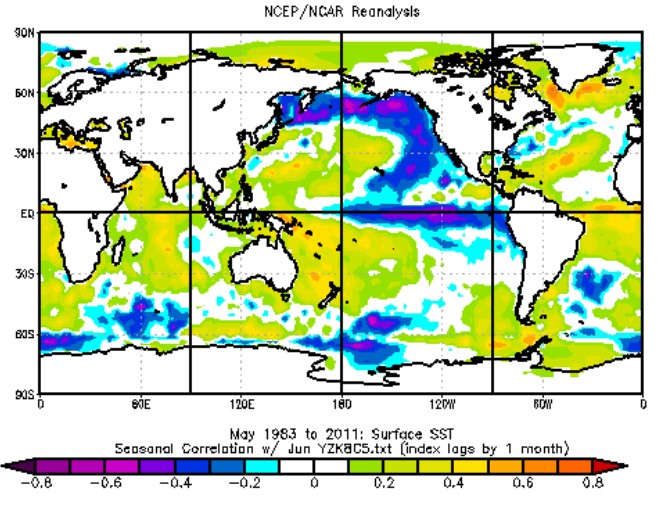

**Figure 3: An example of global correlation map built using the correlations between the cluster-level average JJAS precipitation time-series (Cluster 5 in this example) and global sea surface temperature (SST) in May during predictor selection process.**

5     PCA is a common approach in climate modeling to reduce the dimensionality of predictors and remove multi-collinearity, while simultaneously extracting the most dominant signals from the potential predictors, typically reflected in the first few PCs. Since PCA is independent of the predictand, retaining the first few PCs as predictors, in lieu of the original variables, also helps to reduce artificial prediction skill. A scree test (Jolliffe, 2002) is performed to determine the optimal number of PCs to retain as predictors and the amount of variance explained in the predictors.

    PCR is performed in a "drop-one-year" cross-validation mode to reduce over-fitting effects and therefore avoid overestimation of prediction skill. This requires reconstructing the principal components for the dropped year, and then multiplying the coefficient estimates with each reconstructed PC respectively in order to obtain the final predicted value for the dropped year (e.g. Block and Rajagopalan, 2009; Wilks, 2011). Q-Q plots are evaluated to verify normally distributed residuals (results not included).

    For the four scenarios, the model structures are quite similar but have subtle differences which could lead to significantly different outcomes (Table 2). Under the NC-D (Eq. (1a, b)) and C-D scenarios (Eq. (2a, b)), the time-series of JJAS seasonal total precipitation in each grid-cell (i.e. at local level) is used as the direct predictand ($Y_{i,t}$); however, the NC-D and C-D scenarios differ, as the former uses the same predictors ($X_t$) across all the grid-cells, while the latter uses different predictors

20     according to the cluster to which the grid-cell is assigned ($X_{j,t}$). In the indirect case, the cluster-level time-series of JJAS seasonal total precipitation (the time-series averaged over all grid-cells that belong to a given cluster, $Y_{m,t}$ or $Y_{j,t}$) is first predicted (Eq. (3a, b) and (4a, b)). The predicted intermediate product ($\widetilde{Y}_{m,t}$ or $\widetilde{Y}_{j,t}$) is then used as the only regressor in the second step to estimate the grid-level precipitation ($\widetilde{Y}_{i,t}$ or $\widetilde{Y}_{i\in j,t}$ for every j; Eq. (3c, d) and (4c, d)). Again, for the C-I scenario, predictors in the first step are unique for each of the eight clusters and grid-cells within that cluster ($X_{j,t}$), while predictors are identical for all grid-cells ($X_t$)

25     under the NC-I scenario.





**Table 2: Equations of linear regression panel models under four scenarios**

|  | **Non-clustered** |  | **Clustered** |  |
|---|---|---|---|---|
| **Direct** | $Y_{i,t} = \tilde{\alpha}_i + \tilde{\beta}_i X_t + \varepsilon_{i,t}$ | ...... (1a) | $Y_{i \in j,t} = \tilde{\alpha}_i + \tilde{\beta}_i X_{j,t} + \varepsilon_{i,t}$ | ...... (2a) |
| **Direct** | $\tilde{Y}_{i,t} = \tilde{\alpha}_i + \tilde{\beta}_i X_t$ | ...... (1b) | $\tilde{Y}_{i \in j,t} = \tilde{\alpha}_i + \tilde{\beta}_i X_{j,t}$ | ...... (2b) |
| **Indirect** | $Y_{m,t} = \tilde{\alpha} + \tilde{\beta} X_t + \varepsilon_t$ | ...... (3a) | $Y_{j,t} = \tilde{\alpha}_j + \tilde{\beta}_j X_{j,t} + \varepsilon_{j,t}$ | ...... (4a) |
| **Indirect** | $\tilde{Y}_{m,t} = \tilde{\alpha} + \tilde{\beta} X_t$ | ...... (3b) | $\tilde{Y}_{j,t} = \tilde{\alpha}_j + \tilde{\beta}_j X_{j,t}$ | ...... (4b) |
| **Indirect** | $Y_{i,t} = \tilde{\eta}_i + \tilde{\gamma}_i \tilde{Y}_{m,t} + \nu_{i,t}$ | ...... (3c) | $Y_{i \in j,t} = \tilde{\eta}_i + \tilde{\gamma}_i \tilde{Y}_{j,t} + \nu_{i,t}$ | ...... (4c) |
| **Indirect** | $\tilde{Y}_{i,t} = \tilde{\eta}_i + \tilde{\gamma}_i \tilde{Y}_{m,t}$ | ...... (3d) | $\tilde{Y}_{i \in j,t} = \tilde{\eta}_i + \tilde{\gamma}_i \tilde{Y}_{j,t}$ | ...... (4d) |

where Y- predictand of JJAS seasonal total precipitation; X- two predictors of top two PCs;
$\varepsilon, \nu$ - error terms; $\tilde{Y}$ - predicted values of JJAS seasonal total precipitation; $\tilde{\alpha}, \tilde{\beta}, \tilde{\eta}, \tilde{\gamma}$ - estimated coefficients; i- grid-cell index; t- time (year) index; j- cluster index; $i \in j$- grid-cell i that belongs to cluster j; m- mean over entire study region that is equivalently the only one cluster.

**3.3 Dynamical modeling approach**

The North American Multi-Model Ensemble (NMME; Kirtman et al., 2014) is an experimental multi-model system consisting of coupled dynamical models from various modeling centers in North America that includes seasonal predictions. To compare with statistical model predictions, NMME JJAS seasonal precipitation predictions (1˚×1˚ grid-cells) are extracted from model ensembles that cover the same time period (1983–2011), geographic region (western Ethiopia), and same lead time (predictions

made on June 1). A subset of 10 NMME models meet these criteria and are retained for further evaluation: (1) COLA-RSMAS-CCSM3, (2) COLA-RSMAS-CCSM4, (3) GFDL-CM2p1, (4) GFDL-CM2p1-are04, (5) GFDL-CM2p5-FLOR-A06, (6) GFDL-CM2p5-FLOR-B01, (7) IRI-ECHAM-AnomalyCoupled, (8) IRI-ECHAM-DirectCoupled, (9) NASA-GMAO, (10) NCEP-CFSv2. The names are kept the same as on the International Research Institute for Climate and Society (IRI) data repository website.

The NMME predictions for each of the 10 models are bias-corrected by applying probability mapping (e.g. Block et al., 2009; Teutschbein and Seibert, 2012; Chen et al., 2013), subject to the observational dataset from NMA. This is performed on a grid-cell by grid-cell basis on standardized data (the NMME dataset is reshaped to 0.1°×0.1° grid-cells to match the observational NMA dataset grid-cell size). The basic steps include:

(1) Fit gamma distributions to time-series from each observed and NMME grid-cell; for NMME this is performed on an individual model basis using all ensemble members available. (Goodness-of-fit tests indicate gamma distributions are appropriate; results not shown).

    (2) Translate gamma distributions into cumulative distribution functions (CDF).

    (3) For any given dynamical model prediction at the grid-cell level, a corrected prediction value is attained by mapping

from the modeled CDF to the observed CDF and applying the inverse gamma distribution. This is repeated for all grid-cells and all NMME models.

After correction, the gamma CDF of predictions and observations approximately match (Fig. 4a). Additionally, each ensemble still retains its variability over time, though the overall ensemble mean is shifted to closely match observation (Fig. 4b).





(a)

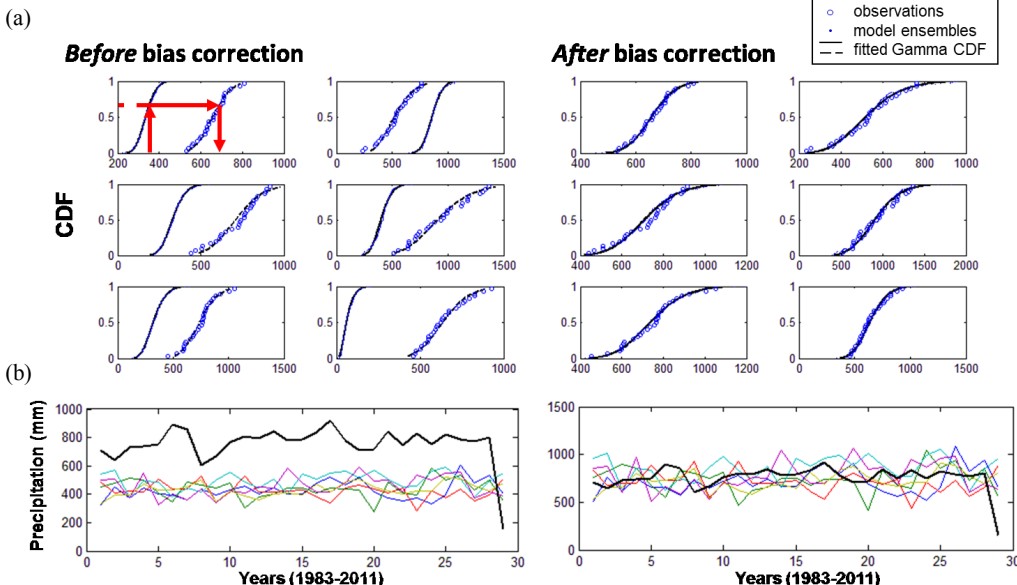

(b)

**Figure 4: (a) bias correction of NMME predictions using probability mapping; (b) precipitation time-series from NMME (colored lines) before and after correction, compared to observations (black line). Examples are shown for randomly selected six grid-cells.**

### 3.4 Performance metrics

5    Pearson correlations are used to measure the standardized covariance between observations and predictions. Ranked probability skill scores (RPSS; Wilks, 2011) are also evaluated to determine categorical skill based on probabilistic predictions. Here, the data are split into three equal terciles representing below-normal, near-normal, and above-normal conditions. A perfect prediction yields an RPSS of 100%, and a prediction with less skill than climatology (long-term averages) yields an RPSS of less than zero. Median RPSS values from all 29 years are reported.

Overall model superiority is evaluated by Akaike information criterion (AIC), Bayesian information criterion (BIC), and generalized cross validation (GCV) scores (Craven and Wahba, 1979; Manning et al., 2008). All metrics reward model parsimony by penalizing models with a larger number of predictors. Smaller AIC, BIC and GCV scores are preferred. The equations used to calculate AIC, BIC, and GCV, respectively, are given by:

$$AIC = N \times \log(RSS/N) + 2 \times K \qquad\qquad ...... (5)$$
$$BIC = N \times \log(RSS/N) + \log(N) \times K \qquad\qquad ...... (6)$$
$$GCV = RSS/[N \times (1 - K/N)^2] \qquad\qquad ...... (7)$$

where N is the number of years (29 years), K is the number of predictors used in the regression, and RSS is the residual sum of squares (equal to the difference between observations and predictions in each year squared, summed over all the years).





## 4 Results

### 4.1 Statistical model predictions

Using the scree test (Cattell, 1966), the first two PCs are retained as predictors for each cluster for the *clustered* case, and the first three PCs are retained for the *non-clustered* case. In all cases, the total variance explained by the PCs retained is approximately 5 70%.

Cluster-level model predictions demonstrate good skill, as most observations fall within the predicted 95% confidence envelope (Fig. 5), and strong positive correlations with observations – ranging from 0.68 to 0.84 – are evident across all clusters and the *non-clustered* study region (Table 3). Additionally, all RPSS values are positive, indicating superior prediction skill over 10 climatology. Among all clusters, *Cluster 5*, in agriculturally rich central-northwestern Ethiopia (Fig. 1), performs best, with correlation and RPSS values of 0.84 and 74.3%, respectively.

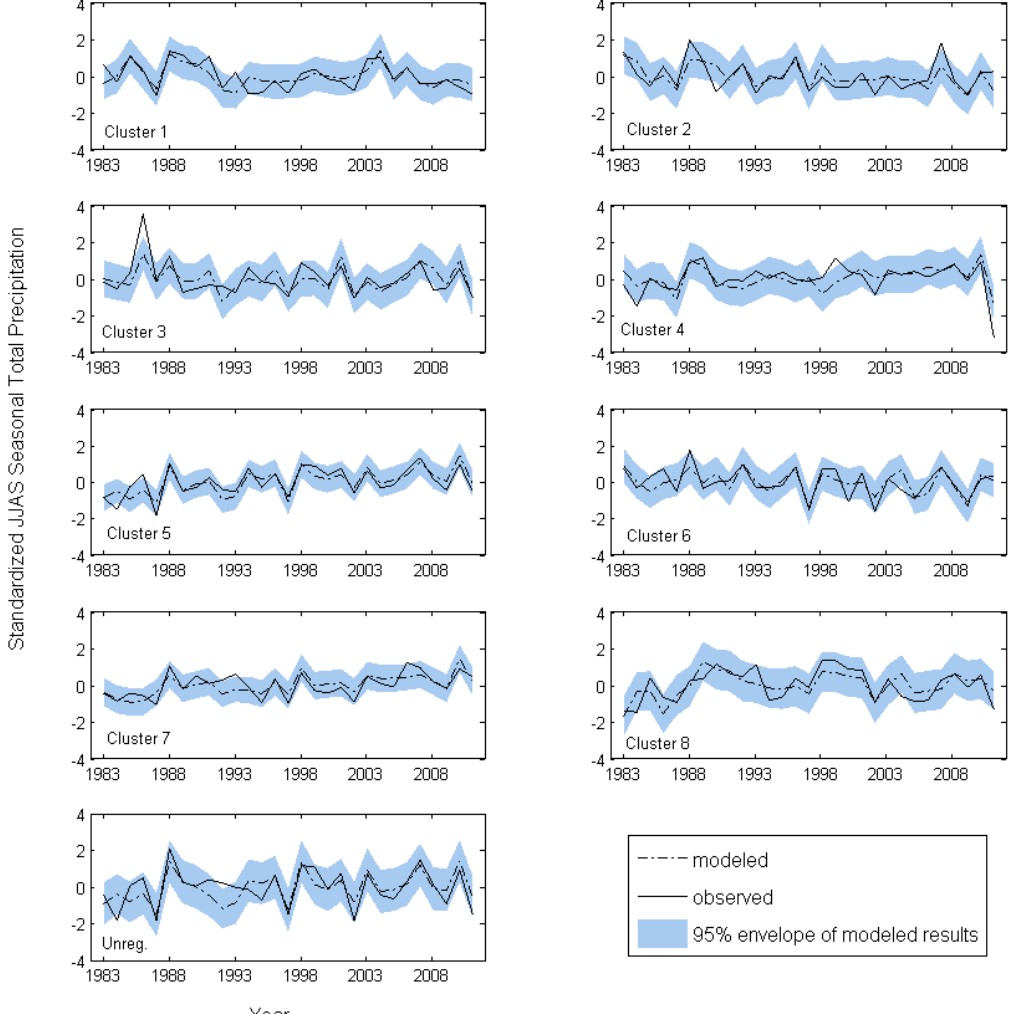

**Figure 5: cluster-level predictions and observations under C-I and NC-I scenario, with drop-one-year cross-validation. The 95% envelope shows the 95% confidence interval constructed using model errors.**



**Table 3: Correlation coefficients (Corr.) and RPSS for predictions (drop-one-year cross-validated) at cluster level compared to observations under C-I and NC-I scenario.**

| Cluster | C1 | C2 | C3 | C4 | C5 | C6 | C7 | C8 | Non-cluster |
|---|---|---|---|---|---|---|---|---|---|
| Corr. | 0.741 | 0.695 | 0.711 | 0.683 | 0.838 | 0.744 | 0.751 | 0.699 | 0.739 |
| RPSS (%) | 45.23 | 26.04 | 36.16 | 19.82 | 74.30 | 5.44 | 51.91 | 48.21 | 48.72 |

At the grid-scale, correlations between predictions and observations are clearly superior for the *clustered* case versus the *non-clustered* case (Fig. 6). Some parts of the region reach a correlation of 0.9, such as central-northwestern Ethiopia, which is consistent with the region of high cluster-level prediction skill (*Cluster 5*). The average correlation over all grid-cells is approximately 0.51 (*direct*) and 0.53 (*indirect*) for *clustered* predictions, compared to 0.24 (*direct*) and 0.27 (*indirect*) for the *non-clustered* predictions (Table 4), although spatial differences are clearly apparent (Fig. 6). In addition to higher average correlations, standard deviations of correlations are also lower in the *clustered* case than in the *non-clustered* case, indicating a more concentrated correlation distribution for these higher values. The percentage of grid-cells with correlations passing the 95% significance test increases from approximately 30% in the *non-clustered* case to more than 80% in the *clustered* case (Table 4).

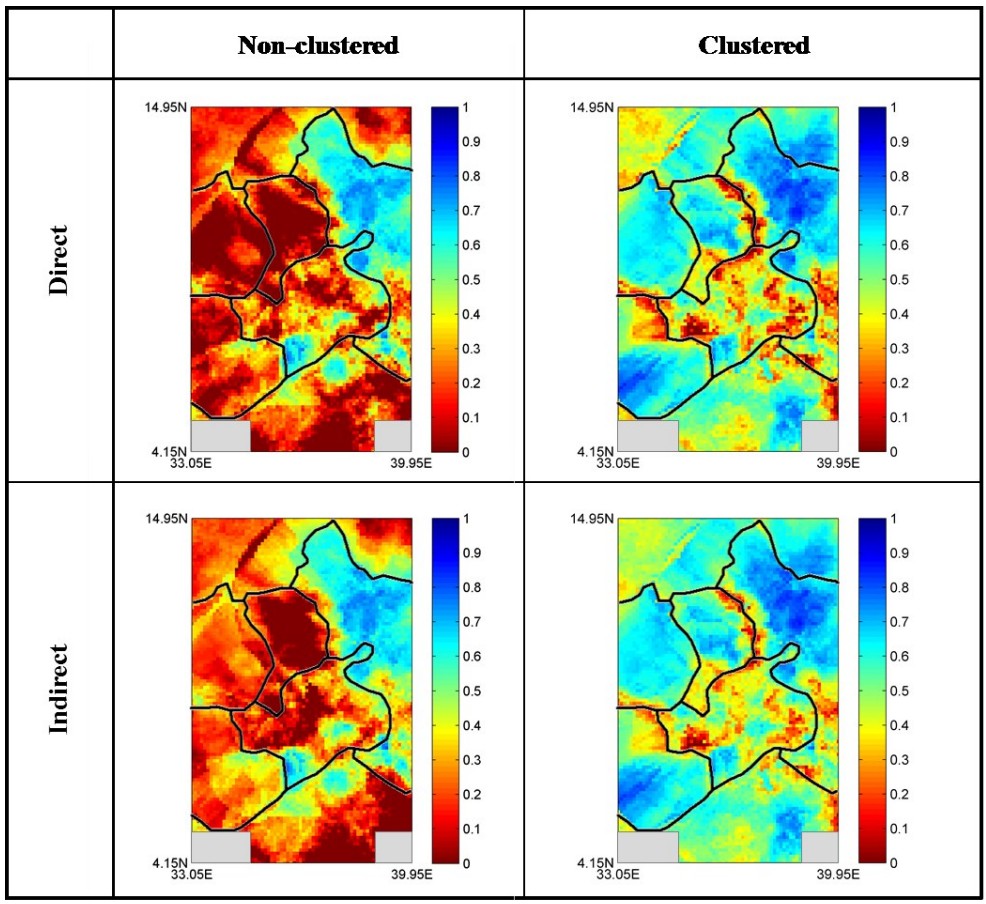

**Figure 6: Pearson correlations between grid-level observations and predictions under four scenarios, with the clustering boundary delineated roughly in black.**




**Table 4: Grid-level Pearson correlation and RPSS statistics**

| Statistical Model | Grid-level correlations | | | Grid-level RPSS | | |
|---|---|---|---|---|---|---|
| | mean | stdev | significant corr % | mean (%) | stdev (%) | positive RPSS % |
| NC-D | 0.237 | 0.245 | 28.1% | 5.42 | 18.46 | 58.8% |
| NC-I | 0.272 | 0.247 | 32.3% | 5.32 | 17.09 | 60.7% |
| C-D | 0.509 | 0.172 | 80.8% | 19.16 | 19.18 | 84.4% |
| C-I | 0.532 | 0.146 | 87.1% | 26.47 | 21.47 | 90.0% |
| Dynamical Model | | | | | | |
| (9) NASA-GMAO | 0.300 | 0.149 | 36.1% | 2.32 | 21.20 | 54.3% |
| (10) NCEP CFSv2 | 0.310 | 0.155 | 37.3% | 3.66 | 16.61 | 61.0% |

Similar findings are evident by evaluating the RPSS. The *non-clustered* predictions are modestly skillful, particularly for the same region of central-northwestern Ethiopia (Fig. 7), with an average RPSS of approximately 5.4% (both *direct* and *indirect*) over the entire study region, however RPSS values improve nearly fourfold to 19.2% and 26.5% in the *clustered* case (*indirect* and *direct*, respectively). Additionally, the percentage of grid-cells with positive RPSS values reaches 84.4% - 90.0% in the *clustered* case (Table 4).

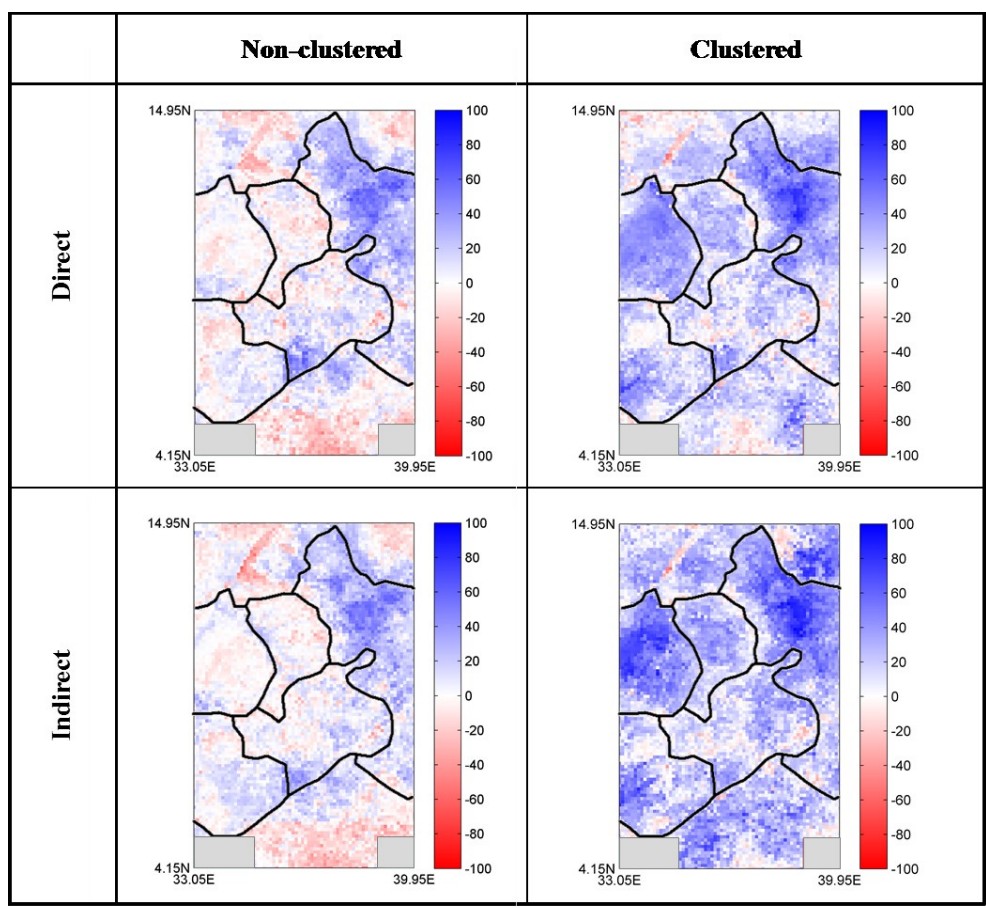

**Figure 7: grid-level RPSS (%) under four scenarios using climate variables as predictors, with the clustering boundary delineated roughly in black.**





At the grid-scale, predictions by the *indirect* approach generally outperform *direct* approach predictions, based on AIC, BIC and GCV values (Table 5), as well as correlation and RPSS values (Tables 4). Using the predicted cluster-level precipitation to predict grid-level precipitation (the *indirect* case) appears to help reduce the effect of over-fitting and smooth grid-scale noise.

From another perspective, the results also suggest that precipitation signals at the regional scale are better explained by large-scale climate variables, while at highly localized scales the signal is less evident. Obviously, however, this is dependent on cluster size and the degree of spatial coherence within each cluster, as demonstrated in this study.

**Table 5: grid-level AIC, BIC, and GCV value statistics**

| Statistical Model | AIC | | BIC | | GCV | |
|---|---|---|---|---|---|---|
| | mean | stdev | mean | stdev | mean | stdev |
| NC-D | 281.05 | 17.86 | 285.15 | 17.86 | 2.00E+04 | 1.46E+04 |
| NC-I | 277.47 | 17.37 | 280.20 | 17.37 | 1.73E+04 | 1.20E+04 |
| C-D | 272.79 | 15.38 | 276.89 | 15.38 | 1.41E+04 | 7.53E+03 |
| C-I | 270.05 | 15.33 | 272.78 | 15.33 | 1.27E+04 | 6.84E+03 |

**4.2 Dynamical model predictions**

The RPSS values based on the prediction ensembles of each dynamical model improve significantly after bias correction, however, the median RPSS values over all the grid-cells are still close to zero (Fig. 8). Only two models, NASA-GMAO and NCEP-CFSv2, show a positive mean RPSS value (2.32% and 3.66%, respectively; Table 4). These two dynamical models also exhibit generally higher grid-level correlations over the study region (averaging 0.30 and 0.31, respectively; Table 4 and Fig. 9), as compared with other NMME models. However, their overall prediction performance is still clearly inferior to that of the

*clustered* statistical models, as assessed by correlation and RPSS metrics (Table 4).

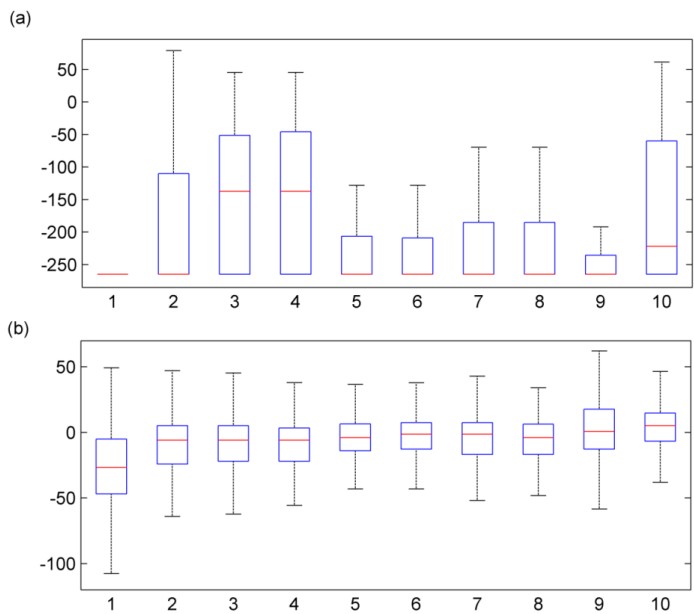

**Figure 8: Boxplots of grid-level RPSS (%) for 10 dynamical models from NMME (a) before and (b) after bias correction, labeled with the same number as listed in the context. Note: For each box plot, the line inside the box is the median, the box edges represent the 25th and 75th percentiles, and the whiskers extend to the most extreme data points not considered outliers (outliers not shown).**





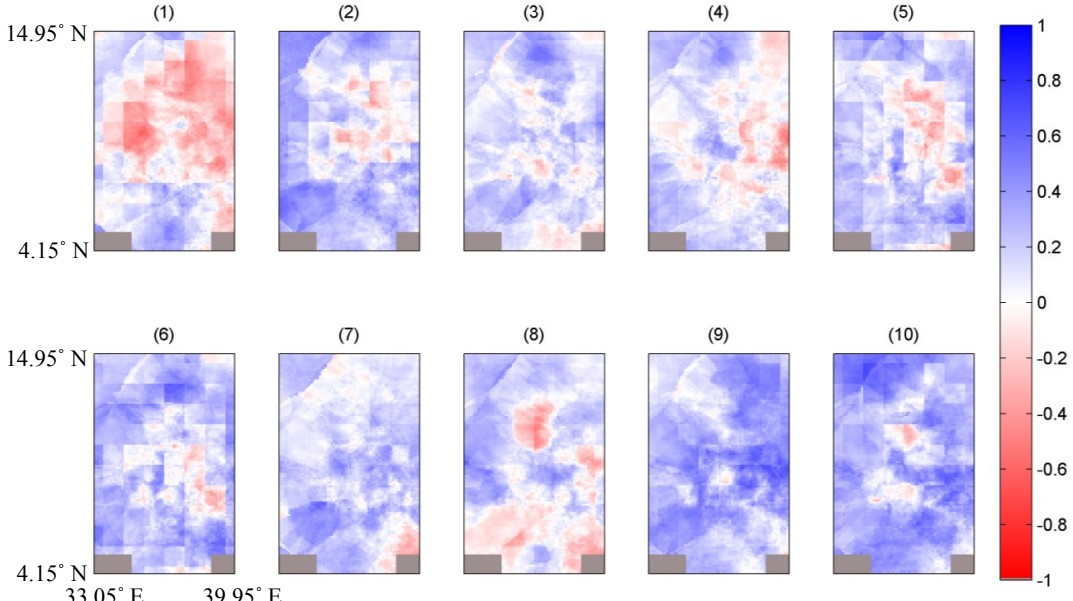

**Figure 9: Pearson correlations between grid-level observations and ensemble mean of bias-corrected predictions for 10 dynamical models from NMME, labeled with the same number as listed in the context. Note that the scale ranges from -1 to 1.**

**5 Conclusions and discussion**

This study demonstrates the potential for season-ahead large-scale climate information to produce skillful and credible high-resolution precipitation predictions under a *clustered indirect* approach in western Ethiopia. At the regional scale, the approach shows promise, particularly compared to current NMA operational forecasts, which are only moderately more skillful than climatology (Korecha and Sorteberg, 2013). The approach adopted here also advances on previous studies (Gissila et al., 2004; Block and Rajagopalan, 2007; Korecha and Barnston, 2007; Diro et al., 2011b; Segele et al., 2015) by first applying an objective

cluster analysis and then conditionally constructing high-resolution predictions. A unique set of predictors is applied to each cluster, which contributes to superior prediction performance at both cluster and grid levels, as compared with predictions from the non-clustered approach. Grid-level prediction under the *indirect* case also reduces the effect of over-fitting relative to the *direct* case.

Although predictions from the statistical model *clustered* approach are superior to all dynamical predictions for this study, improvements in dynamical models continues , and their application to seasonal precipitation prediction is likely to grow (e.g. Palmer et al., 2004; Saha et al., 2006; Lim et al., 2009). Multi-model combinations of statistical and dynamical models were also investigated for potential improvement of prediction skills through pooling, linear regression, and Bayesian model averaging (BMA; Raftery et al., 1997) using the best statistical model (C-I) and two best dynamical models (NASA-GMAO and NCEP-

CFSv2), however, the overall performance was inferior to the single statistical C-I model (results not shown).

Even though *clustered* statistical model predictions are promising overall, it is worth noting that relatively poor prediction performance is evident in some locations. One such place is along cluster boundaries, where assignment of grid-cells to one cluster versus the neighboring cluster is almost arbitrary, and clearly less certain than grid-cells falling within the central parts of



clusters (Fig. 6 and 7). Poor prediction skill is also evident in some of the mountainous regions of the study area, where the hydroclimatic processes that produce precipitation are likely driven by orographic and other local factors rather than large-scale climate variables. To test the prospects for improving prediction performance by including season-ahead local variables, soil moisture and spring rains were investigated; however, no significant improvement was found for the *clustered* case, and

correlations actually deteriorate for the *direct* case. Thus adding local predictors in this case may simply serve to introduce more noise and encourage over-fitting.

Additional prediction features also warrant future attention, including longer prediction lead times and evaluation of other relevant characteristics (e.g. intra-seasonal dry spells, seasonal onset or cessation, etc.). Improving predictive capabilities may

not be a complete panacea, but it can continue to be an important part of a decisions-maker's portfolio as they cope with hydroclimatic variability and its inherent risks.

**6 Data availability**

The National Centers for Environmental Prediction and National Center for Atmospheric Research (NCEP/NCAR) reanalysis dataset can be accessed through the National Oceanic & Atmospheric Administration (NOAA) Earth System Research

Laboratory (ESRL) website (https://www.esrl.noaa.gov/psd/data/reanalysis/).

The NMME hindcasts are available through the International Research Institute for Climate and Society (IRI) website (http://iridl.ldeo.columbia.edu/SOURCES/.Models/.NMME/).

The gridded precipitation dataset in western Ethiopia is available upon request from NMA (http://www.ethiomet.gov.et/).

**Competing interests**

The authors declare that they have no conflict of interest.

**Acknowledgements**

This study was supported by NASA Project NNX14AD30G and NSF PIRE Project 1545874. We acknowledge the National

Meteorological Agency of Ethiopia for sharing data.

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
