# Peer review of "Does objective cluster analysis serve as a useful precursor to seasonal precipitation prediction at local scale? Application to western Ethiopia"

_Hydrology and Earth System Sciences, 2017_

## Referee Comment (RC1) · Anonymous Referee #1 · 1 Mar 2017

The manuscript aims at developing a statistically based seasonal precipitation forecast model for Western Ethiopia. The target area is separated into homogeneous regions by means of a k-means cluster analysis of summer precipitation amounts. Eight regions with similar precipitation variability are defined. For each of them, a linear regression based forecast model is calibrated. Results are compared with a general forecast for the entire region and are found to be superior. In a final step the forecast is down-scaled to a high resolution grid, again by means of a liner regression approach. The target of the study is timely, since local precipitation predictions are often required for water management and planning, and the manuscript is well structured and easy to follow. However I have some serious concerns about the calibration and particularly

the evaluation of the statistical model. Further I would recommend to give some detailed information on the climate characteristics of the cluster regions and the major large scale influences.

1) Introduction, clustering and different predictor variables: An introduction into the climate of the target region is missing. Further, a detailed analysis of the precipitation characteristics of each cluster would be a basis for the interpretation of the modeling results. Some of the precipitation time series in Fig. 5 look highly correlated. Are simple statistical techniques really able to forecast those slight differences? And if, which predictor variables are responsible for the spatial variations of precipitation in Western Ethiopia? An analysis of the predictor-predictant relationships for each cluster would not only give some insights into the model structure and the large scale climate mechanisms of the target area, but also help to support (or scrutinize) the results of the modeling exercise.

2) Calibration of the statistical model and overfitting: Correlations between cross-validated modeling results and observations in the order of 0.7-0.85 are very high (in fact they exceed the skills of well known forecast models) and are questionable. I believe, that those results are due to overfitting (particularly due to the predictor selection). The predictors for each of the clusters are selected based on all years, the cross-validation is only performed for the calibration of the linear model. In order to fully evaluate the model skill, the predictor selection must be included in the cross-validation (i.e. chose predictors at each step of the cross validation, e.g. based on a correlation threshold). Most likely the model skill will significantly drop, I could imagine that a step wise selection of predictors might slightly improve the results.

3) Evaluation of the Downscaling approach: As the predictor selection, the downscaling procedure is not included in the cross-validation. I recommend to conduct the cross-evaluation for the entire modeling chain. That means, the predictor selection, cluster forecast and downscaling approach need to be calibrated based on (n-1) years, in order to forecast gridded precipitation for the remaining year.

In general, one should be aware, that there is a linear dependence of the cluster based and the gridded forecast. Thus, the downscaling approach will better reproduce the local climate, however the variability (drought and moist years) will be equal to the culstered result. The term "gridded" forecast is somehow misleading – I would prefer "downscaling of regional forecast"

Detailed remarks: 1) The abstract is very short and could certainly be more informative (e.g. by including some results)

2) The discussion of state of the art forecasting models in the introduction is very short. Particularly during recent years, several studies investigated the skill of statistical models for regional scale precipitation forecasts (some of them are even based on clustering or PCA). I would recommend to better discuss the literature and the advantage of your approach in the introduction. See for example:

Hertig, E. and Jacobeit, J.: Predictability of Mediterranean climate variables from oceanic variability. Part II: Statistical models for monthly precipitation and temperature in the Mediterranean area, Clim. Dynam., 36, 825–843, doi:10.1007/s00382-010-0821-3, 2010.

Suárez-Moreno, R. and Rodríguez-Fonseca, B.: S4CAST v2.0: sea surface temperature based statistical seasonal forecast model, Geosci. Model Dev., 8, 3639–3658, doi:10.5194/gmd-8-3639-2015, 2015.

Gerlitz, L., Vorogushyn, S., Apel, H., Gafurov, A., Unger-Shayesteh, K. & Merz, B.: A statistically based seasonal precipitation forecast model with automatic predictor selection and its application to central and south Asia, HESS 20, 4605–4623 , doi:10.5194/hess-20-4605-2016 , 2016.

3) The predictor selection is based on correlation maps, and regions with potential forecast skill are identified (see Tab.1). Please map the regions and show the correlation maps for some clusters.

[Figure]

4) P7,l10: PCAs are cross-validated. This is somehow unclear to me. PCA is usually used for dimension reduction. Is the cross-validation done for the loadings of the pca in order to investigate how these change based on different input data?

5) Dynamical Models: The section on dynamical models is poorly integrated. Please give some more information on the models in general. If (as expected) the skill of the statistical model drops as a consequence of the cross-evaluation, a more detailed comparison of skills might be interesting.

6) P9,l15: Please give some more information on the performance measures (BIC, AIC, GCV).

7) P10: How exactly is the envelope (uncertainty interval) calculated? Is this based on the assumption that cross-validated residuals of the regression are normal distributed?

---

## Referee Comment (RC2) · Anonymous Referee #2 · 9 Mar 2017

The authors proposed local level seasonal predictions of precipitation in the Western Ethiopia using the statistical approach at eight homogeneous regions clustered using k-means clustering technique. Large scale climate variables are used as potential predictors in developing the statistical model and unique set of predictors were assigned for each region. Results are compared with the dynamical prediction at regional scale and reported as the statistical approach is superior. This study is timely and helpful for the study region where rainfall is highly variable in space. The manuscript is well written and easy to understand and follow. I think this study will be a nice addition however a few moderate issues need to be resolved first. Please see my comments below.

1. The statistical approach is fully data driven approach that depends on the quality and length of the data. So how efficient is this technique in the area where there is sparse and poor quality data in the case of developing country like Ethiopia? The authors did not show how good the gridded rainfall data is through either validating with gauged data for selected weather stations or previous literatures that support the quality of this data.

2. The author argue that this study gives perdition of seasonal precipitation at high resolution in the region. However, the classification of the homogeneous regions by NMA, Ethiopia, for the study region is almost equivalent (Koricha et al., 2007, pg 7685). I do not see any benefits of this study interms of the spatial resolutions at regional scale for Western Ethiopia.

3. No effort have been made on finding out the time lag between the predictor variables and the seasonal rainfall in the study area. For example which month of sea surface temperature really affects the seasonal rainfall in the study area.

4. I don't see any comparison of the result with the current operational NMA forecast in the manuscript, however, it is reported in the conclusion section asif the result is compared at the regional scale with NMA operational forecast.

5. The abstract is too short and mainly focused on the merit of conducting this study. It would be good if it is supported with some finding. There is no introduction given about the study area interms of the rainfall patter, topography etc.

Reference

Korecha, D., and Sorteberg, A.: Validation of operational seasonal rainfall forecast in Ethiopia, Water Resources Research, 49, 20 7681-7697, 10.1002/2013wr013760, 2013.

---

## Editor Comment (EC1) · Q. J. Wang (Editor) · 8 May 2017

Dear Authors

You will need to make initial replies to comments by the reviewers. After that, I will review both the reviewer comments and your replies and advise you on the next step. Thanks

Best regards

QJ Wang

---

## Author Comment (AC1) · 8 May 2017

Dear Editor,

We are preparing the responses to reviewers. We will submit our responses before the due date on May 17. Thank you for your kind reminder.

Regards,

Ying Zhang
* * *

---

## Short Comment (SC1) · 17 May 2017

Dear Editor,

We have submitted our responses to the comments. As instructed, the revised manuscript are not prepared at this stage, although we include some revised texts in the responses. We look forward to hearing from you. Thank you.

Regards,

Ying Zhang

---

## Author Comment (AC2) · 17 May 2017

The manuscript aims at developing a statistically based seasonal precipitation forecast model for Western Ethiopia. The target area is separated into homogeneous regions by means of a k-means cluster analysis of summer precipitation amounts. Eight regions with similar precipitation variability are defined. For each of them, a linear regression based forecast model is calibrated. Results are compared with a general forecast for the entire region and are found to be superior. In a final step the forecast is downscaled to a high resolution grid, again by means of a liner regression approach. The target of the study is timely, since local precipitation predictions are often required for water management and planning, and the manuscript is well structured and easy to follow. However I have some serious concerns about the calibration and particularly the evaluation of the statistical model. Further I would recommend to give some detailed information on the climate characteristics of the cluster regions and the major large scale influences.

1) Introduction, clustering and different predictor variables: An introduction into the climate of the target region is missing. Further, a detailed analysis of the precipitation characteristics of each cluster would be a basis for the interpretation of the modeling results. Some of the precipitation time series in Fig. 5 look highly correlated. Are simple statistical techniques really able to forecast those slight differences? And if, which predictor variables are responsible for the spatial variations of precipitation in Western Ethiopia? An analysis of the predictor-predictant relationships for each cluster would not only give some insights into the model structure and the large scale climate mechanisms of the target area, but also help to support (or scrutinize) the results of the modeling exercise.

We thank the reviewer for the comment. To address it, firstly, we provide additional description of the climate in the study region. The texts are inserted into Section 2 "Application to western Ethiopia and objectives of the study" (Page 3 Line 5):

"Precipitation in western Ethiopia peaks in the summer with approximately 70% of annual total precipitation falling during the main raining season - also known as the *Kiremt* season spanning from June to September (JJAS). On average, the seasonal total precipitation in the study region is approximately 760 mm; however in the northwest, precipitation can exceed 1200 mm (Fig. 1a). Along with the high spatial variability in this mountainous region, the temporal variability is also significant with spatial-average seasonal total precipitation ranging from 650 mm in dry years up to 900 mm in wet years (Fig. 1b). These highly variable spatial and temporal precipitation patterns have made skillful seasonal predictions challenging, particularly at local scales. "

[Figure]

Figure 1: Spatial and temporal variability of June-September seasonal total precipitation in western Ethiopia: (a) spatial pattern of temporal-average, and (b) spatial-average time series.

Secondly, the precipitation characteristics of each cluster is described in a previous publication on cluster analysis (Zhang et al., 2016), however we agree that a brief summary should be provided here to help set the content and make the work more integrated. Therefore, we have added the text below to the original manuscript at the end of Section 3.1 "Cluster analysis" (Page 4 Line 10):

"The mean time series of each cluster illustrates high intra-correlation within the cluster and low inter-correlation between any two clusters, indicating strong coherency of the clustering results. A detailed analysis including a complete correlation table and unique patterns for each cluster-level time series associated with large climate variables is provided in Zhang et al. (2016), which readers are referred to for more details."

As we can see from the analysis in Zhang et al. (2016), some of the clusters contain stronger tele-connections to equatorial Pacific SSTs (an indicator of El Niño or La Niña), while other clusters are more affected by regional/local climate variables such as the pressure systems surrounding the African continent. This motivates us to use cluster analysis as a precursor to find proper predictors for each homogeneous region, which can capture the differences between the targeted predictand in each cluster.

Additional discussion on which predictor variables may be responsible for the spatial variations of precipitation in western Ethiopia will be provided based on a combination of the previous concurrent connection to large-climate variable analysis (Zhang et al., 2016) and the prediction results.

While additional analysis into the predictor-predictand relationship is clearly possible, we believe associating precipitation patterns with concurrent climate variables provides a solid inference into this relationship and strong support for explaining climate mechanism.

2) Calibration of the statistical model and overfitting: Correlations between crossvalidated modeling results and observations in the order of 0.7-0.85 are very high (in fact they exceed the skills of well known forecast models) and are questionable. I believe, that those results are due to overfitting (particularly due to the predictor selection). The predictors for each of the clusters are selected based on all years, the cross-validation is only performed for the calibration of the linear model. In order to fully evaluate the model skill, the predictor selection must be included in the cross-validation (i.e. chose predictors at each step of the cross validation, e.g. based on a correlation threshold). Most likely the model skill will significantly drop, I could imagine that a step wise selection of predictors might slightly improve the results.

We thank the reviewer for the insightful comment. We admit that the model is overfit due to our methodological procedure in the predictor selection process as the reviewer notes. Therefore, we re-performed the entire process including predictor selection and regression based on cross-validation; that is, we dropped the target year when creating the global correlation map to search for predictors. As a result, there are total of 1044 (29 x 9 x 4) global correlation maps given 29-year time-series, 8 clusters plus 1 non-cluster scenario, and 4 climate variables. Hence, we developed a program to help automatically select highly correlated and justifiable regions as predictors. A description of the method is added to Section 3.2 "Statistical modeling approach" (Page 5 Line 15) and also included here:

"To avoid overfitting, the entire process including predictor selection and statistical modeling is processed using cross-validation. To start, drop-one-year precipitation observations for JJAS averaged across the region and each cluster are spatially correlated independently with each global climate variable. As a result, there are total of 1044 global correlation maps given the 29-year time-series, eight clusters plus one non-cluster, and four climate variables. Hence, a program to automatically select highly correlated and justifiable regions as predictors is developed. The following steps describe the statistical modeling process:

(1) Grid-cells within each justifiable region (e.g. equatorial Pacific; Fig. 2) with correlation above the 99% significance level are identified (Fig. 3).
(2) The top 10% of the identified grid-cells with the highest correlation in each region is then selected, in order to boost the potential model skill.
(3) For each region, data of the selected grid-cells within the region are spatially averaged (defined as "pre-predictors").
(4) Pre-predictors are combined and transformed (for each cluster or non-cluster, and each dropped year analysis separately) through principal component analysis (PCA; Jolliffe, 2002).
(5) The top principal components (PCs) from the PCA with 95% variance explained are used as predictors … (the following steps are the same as in the original manuscript)

[Figure]

Figure 2: Justifiable climate regions globally for selecting predictors: (a) For SLP and GH at 500 mb with regions including EP, ES, LO, AH, SH, MH, and AM. For SAT, only LO is included. (b) For SST with regions including EP, NI, SI, and AT. *Note: EP - equatorial Pacific region, ES – Tahiti island for ENSO measurement, LO - local region, AH - Azores High, SH - St Helena High, MH - Mascarene High, AM - SW Asian Monsoon, NI - North Indian Ocean, SI - South Indian Ocean, AT - Equatorial/South Atlantic Ocean.*

[Figure]

Figure 3: Correlation map between mean JJAS seasonal precipitation time series in Cluster 5 and global SST under cross-validation, with correlations lower than 99% significance level masked out (one-tail test).

The results are also updated accordingly. As the reviewer expected, the model skill decreases significantly. Under the same model of PCR, the correlations of cluster-level prediction and observation now range from -0.180 to 0.504 (compared to 0.683 - 0.838 originally), with Cluster 5 having the highest correlation while Cluster 6 showing the lowest. Similarly for RPSS, 5 out of 8 clusters show skillful prediction compared to climatology (Table 1). However, we do see improvement over non-cluster scenarios for some of the clusters.

Table 1: Correlation coefficients (Corr.) and RPSS for predictions (drop-one-year cross-validated) at cluster level compared to observations under C-I and NC-I scenario. (PCR model)

| Cluster | C1 | C2 | C3 | C4 | C5 | C6 | C7 | C8 | Non-cluster |
|---|---|---|---|---|---|---|---|---|---|
| Corr. | 0.163 | -0.010 | 0.179 | 0.188 | 0.504 | -0.180 | 0.351 | -0.122 | 0.297 |
| RPSS(%) | 33.41 | -21.66 | 43.01 | 12.46 | 27.40 | -37.79 | 20.63 | -55.96 | 13.25 |

We have also tried stepwise regression with forward selection and backward elimination algorithm (von Storch and Zwiers, 1999). The skill does increase a little for some cluster, but other clusters had a deterioration of prediction skills. One table showing the results from stepwise regression with forward selection probability of 0.05 and backward elimination probability of 0.05 (can also be understood as significance levels) is presented below.

Table 2: Correlation coefficients (Corr.) and RPSS for predictions (drop-one-year cross-validated) at cluster level compared to observations under C-I scenario. (Stepwise model)

| Cluster | 1 | 2 | 3 | 4 | 5 | 6 | 7 | 8 |
|---|---|---|---|---|---|---|---|---|
| Corr. | 0.154 | -0.094 | 0.175 | 0.248 | 0.508 | -0.197 | 0.305 | -0.044 |
| RPSS | -3.31 | -44.17 | 40.60 | -49.89 | 33.24 | -77.10 | 25.00 | -24.87 |

Comparing to PCR, stepwise regression overfits some specific predictors, such as the SST in the equatorial Pacific region or SLP for the St. Helena High, whereas PCR extracts the main signal first and then fits with a regression model. As a result, less noise is left in the predictors (PCs) using PCR than those in the stepwise regression. Hence, PCR is more reliable, regardless of the prescribed rule for selecting a certain number of predictors, while stepwise regress is extremely sensitive to different prescriptions of significance levels of selecting and eliminating probability. Therefore, we consider PCR a more reliable method in this case and keep the results from PCR for this work with added discussion on stepwise at the end. Table 1 above is included in the revised manuscript together with other updated tables, figures and result analysis, which are also included here:

"Correlations between cluster-level model predictions and observations range from -0.180 to 0.504, with Cluster 5 having the highest correlation and Cluster 6 the lowest (Table 1). In approximately 1/5 of the 29 years, the observation falls outside the prediction envelope (Fig. 4), indicating model overfitting and an inability of the predictors to capture precipitation variability. For RPSS, 5 out of 8 clusters

indicate superior prediction skill over climatology (Table 1). Improvement in terms of RPSS over the non-cluster scenario is evident for Cluster 1, 3, 5 and 7. Among all clusters, Cluster 5, in agriculturally rich central-northwestern Ethiopia (Fig. 1 in original manuscript), performs best, with correlation and RPSS values of 0.5 and 27.4%, respectively. "

[Figure]

Figure 4: Cluster-level predictions and observations under C-I and NC-I scenario, with drop-one-year cross-validation. The 95% envelope shows the 95% confidence interval constructed using model errors.

"At the grid-scale, depending on the case (*direct* or *indirect)*, and for different clusters, correlations between predictions and observations can favor the clustered case or the non-clustered case (Fig. 5). In general, the *indirect* model provides a smoother pattern of correlations, with grid-cells showing a negative correlation in the *direct* case now improved to near or above zero (Fig. 5). For example, Cluster 5 under the *indirect* case illustrates a more consistent positive correlation within the cluster. Some parts

of the region reach a correlation of 0.6, such as central-northwestern Ethiopia, which is consistent with the region of high cluster-level prediction skill (Cluster 5). The percentage of grid-cells with correlations passing the 95% significance test is the highest for the NC-D case (Table 3); however, if only comparing within a clustered region, skills can be higher for the *clustered* case."

[Figure]

Figure 5: Pearson correlations between grid-level observations and predictions under four scenarios, with the clustering boundary delineated roughly in black.

"Similar findings are evident by evaluating the RPSS. The predictions are most skillful for the same region of central-northwestern Ethiopia (Cluster 5; Fig. 6). The percentage of grid-cells with positive RPSS values is the highest for the C-I case (Table 4), indicating the *clustered indirect* case is superior in terms of RPSS metrics."

[Figure]

Figure 6: Grid-level RPSS (%) under four scenarios using climate variables as predictors, with the clustering boundary delineated roughly in black.

Table 3: Grid-level Pearson correlation and RPSS statistics

| Statistical Model | Grid-level correlations | | | Grid-level RPSS | | |
|---|---|---|---|---|---|---|
| | mean | stdev | significant corr % | mean (%) | stdev (%) | positive RPSS % |
| NC-D | 0.128 | 0.258 | 19.3% | -5.21 | 26.97 | 42.8% |
| NC-I | 0.063 | 0.186 | 3.1% | -2.26 | 14.62 | 43.9% |
| C-D | 0.055 | 0.230 | 10.6% | -13.97 | 30.97 | 33.9% |
| C-I | 0.081 | 0.206 | 12.4% | -9.60 | 29.39 | 44.4% |
| **Dynamical Model** | | | | | | |
| (9) NASA-GMAO | 0.3 | 0.149 | 36.10% | 2.32 | 21.2 | 54.30% |
| (10) NCEP CFSv2 | 0.31 | 0.155 | 37.30% | 3.66 | 16.61 | 61.00% |

3) Evaluation of the Downscaling approach: As the predictor selection, the downscaling procedure is not included in the cross-validation. I recommend to conduct the crossevaluation for the entire modeling chain. That means, the predictor selection, cluster forecast and downscaling approach need to be calibrated based on (n-1) years, in order to forecast gridded precipitation for the remaining year.

In general, one should be aware, that there is a linear dependence of the cluster based and the gridded forecast. Thus, the downscaling approach will better reproduce the local climate, however the variability (drought and moist years) will be equal to the culstered result. The term "gridded" forecast is somehow misleading – I would prefer "downscaling of regional forecast"

The downscaling procedure is included in the cross validation. We apologize if this is not clear and added one sentence to the statistical model steps (Page 5 Line 30):

"For the indirect case only, cluster-level predictions are regressed to the grid-level. Note that the downscaling of cluster-level predictions to grid-level predictions is also cross-validated to avoid overfitting."

We also revised "gridded forecast" to "prediction at grid level" and "downscaling of cluster-level to grid-level prediction" to provide more appropriate descriptions.

Detailed remarks: 1) The abstract is very short and could certainly be more informative (e.g. by including some results)

We thank the reviewer for the comment and we have revised the abstract with highlights on results as the reviewer suggested. The added texts are also included here:

"… makes clear advances in modeling methodology and resolution, as compared with previous studies. The statistical model prediction results show improvements over non-clustered case for some clusters. Among those clusters, Cluster 5, in agriculturally rich central-northwestern Ethiopia, performs best, with correlation and RPSS values of 0.5 and 27.4%, respectively. The general skill of dynamical models over the entire study region is higher than statistical models, although dynamical models produce predictions at a lower resolution. However, for some specific clustered regions such as Cluster 5, the statistical model outperforms dynamical models for grid-level predictions producing higher correlations and RPSS at a finer resolution. "

2) The discussion of state of the art forecasting models in the introduction is very short. Particularly during recent years, several studies investigated the skill of statistical models for regional scale precipitation forecasts (some of them are even based on clustering or PCA). I would recommend to better discuss the literature and the advantage of your approach in the introduction. See for example:

Hertig, E. and Jacobeit, J.: Predictability of Mediterranean climate variables from oceanic variability. Part II: Statistical models for monthly precipitation and temperature in the Mediterranean area, Clim. Dynam., 36, 825–843, doi:10.1007/s00382-010- 0821-3, 2010.

Suárez-Moreno, R. and Rodríguez-Fonseca, B.: S4CAST v2.0: sea surface temperature based statistical seasonal forecast model, Geosci. Model Dev., 8, 3639–3658, doi:10.5194/gmd-8-3639-2015, 2015.

Gerlitz, L., Vorogushyn, S., Apel, H., Gafurov, A., Unger-Shayesteh, K. & Merz, B.: A statistically based seasonal precipitation forecast model with automatic predictor selection and its application to central and south Asia, HESS 20, 4605–4623 , doi:10.5194/hess-20-4605-2016 , 2016.

We thank the reviewer for the comment. The recommended literatures are closely related to our methodology, although they are not based on the same study region. We have added them to the current literature review collection under Section 3.2 "statistical modeling approach" (Page 4 Line 20) with the following texts inserted:

"Many studies have investigated statistical models for seasonal climate prediction. The variety of those studies lies in the pre-classification of predictor or predictand regime, predictor selection process, as well as statistical methods. For example, Hertig and Jacobeit (2011a) investigate sea surface temperature (SST) regimes as potential predictors for subsequent precipitation and temperature in the Mediterranean area. Through techniques including multiple applications of PCA, 17 stationary SST regimes were identified. Gerlitz et al. (2016) apply a k-means cluster analysis to grid-cells identified with significant correlations in the predictor field in order to facilitate predictor selection. Suárez-Moreno and Rodríguez-Fonseca (2015) investigate stationarity based on a sufficient long time series using a 21-year moving correlation window. The statistical prediction models are then applied to each stationary period respectively and the entire period for comparison. Despite diverse methods in seasonal prediction, multiple linear regression (MLR) is favored by many as a statistical modeling approach given its well-developed theory, simple model structure, efficient processing, and often skillful outcomes (e.g. Omondi et al., 2013; Camberlin and Philippon, 2002; Diro et al., 2008; Hertig and Jacobeit, 2011b). As mentioned, only a few studies have focused on seasonal precipitation prediction in Ethiopia (Gissila et al., 2004; Block and Rajagopalan, 2007; Korecha and Barnston, 2007; Diro et al., 2008; Diro et al., 2011; Segele et al., 2015), and almost all of them include the applications of MLR. This study also applies MLR to predict seasonal precipitation in combination with principal component analysis (PCA), yet differentiates from other studies by applying predictions to pre-defined homogeneous predictand regions and further translating to local-level predictions."

3) The predictor selection is based on correlation maps, and regions with potential forecast skill are identified (see Tab.1). Please map the regions and show the correlation maps for some clusters.

We have added one figure showing regions for selecting predictors and one correlation map for a cluster as an example. Please refer to Figure 2 and 3 in the responses above.

4) P7,l10: PCAs are cross-validated. This is somehow unclear to me. PCA is usually used for dimension reduction. Is the cross-validation done for the loadings of the pca in order to investigate how these change based on different input data?

PCR is cross-validated as indicated on Page 7 Line 10. To be specific, we performed PCA on the time series with the target-year data dropped, and then use the resultant PCs as predictors for the regression model. After the regression coefficient estimates are obtained, the principal components for the dropped year are *reconstructed*, and then multiplied with the coefficient estimates respectively in order to obtain the final predicted value for the dropped year. Please see more details in Wilks (2011).

5) Dynamical Models: The section on dynamical models is poorly integrated. Please give some more information on the models in general. If (as expected) the skill of the statistical model drops as a consequence of the cross-evaluation, a more detailed comparison of skills might be interesting.

We apologize for our poorly integrated section on dynamical model and have improved it with an additional introduction of the NMME models:

"The North American Multi-Model Ensemble (NMME) (Kirtman et al., 2014) is an experimental multi-model seasonal forecasting system consisting of dynamical coupled models from various modeling centers in the North America. To our knowledge, it is also the most extensive multi-model seasonal prediction archive. The NMME provides gridded climate predictions that cover regions globally and with different lead times. The hindcasts of monthly mean precipitations are easily accessible through the International Research Institute for Climate and Society (IRI) website (http://iridl.ldeo.columbia.edu/SOURCES/.Models/.NMME/), and can be easily aggregated to seasonal totals for comparison with the statistical model results in this study."

We agree that after the model is revised, the skills of statistical model and dynamical model are comparative now. We have provided additional result analysis with comparison between them.

"The general skill of dynamical models over the entire study region is higher than statistical models, although dynamic models produce predictions at a lower resolution. However, for some specific clustered regions such as Cluster 5, statistical models outperform dynamical models with grid-level predictions demonstrating higher correlations and RPSS at a higher resolution."

6) P9,l15: Please give some more information on the performance measures (BIC, AIC, GCV).

We have eliminated the performance measures BIC AIC GCV, as the revised predictor selection rule produces a dynamic number of predictors depending on which years is dropped and for different clusters (for the indirect case) or grid-cells (for the direct case).

7) P10: How exactly is the envelope (uncertainty interval) calculated? Is this based on the assumption that cross-validated residuals of the regression are normal distributed?

Yes. Q-Q plots are evaluated to verify normally distributed residuals (results not included) as indicated on Page 7 Line 14. To make it clear, we have added the following text to the same line:

"A 95% confidence interval of the cross-validated predictions is also constructed conditioned on model errors. Q-Q plots are evaluated to verify normally distributed residuals (results not included)."

References

Block, P. J., and Rajagopalan, B.: Interannual Variability and Ensemble Forecast of Upper Blue Nile Basin Kiremt Season Precipitation, J. Hydrometeor, 8, 327-343, http://dx.doi.org/10.1175/JHM580.1, 2007.

Camberlin, P., and Philippon, N.: The East African March–May Rainy Season: Associated Atmospheric Dynamics and Predictability over the 1968–97 Period, Journal of Climate, 15, 1002-1019, 10.1175/1520-0442(2002)015<1002:TEAMMR>2.0.CO;2, 2002.

Diro, G. T., Black, E., and Grimes, D. I. F.: Seasonal forecasting of Ethiopian spring rains, Meteorological Applications, 15, 73-83, 10.1002/met.63, 2008.

Diro, G. T., Grimes, D. I. F., and Black, E.: Teleconnections between Ethiopian summer rainfall and sea surface temperature: part II. Seasonal forecasting, Climate Dynamics, 37, 121-131, 10.1007/s00382-010-0896-x, 2011.

Gerlitz, L., Vorogushyn, S., Apel, H., Gafurov, A., Unger-Shayesteh, K., and Merz, B.: A statistically based seasonal precipitation forecast model with automatic predictor selection and its application to central and south Asia, Hydrol. Earth Syst. Sci., 20, 4605-4623, 10.5194/hess-20-4605-2016, 2016.

Gissila, T., Black, E., Grimes, D. I. F., and Slingo, J. M.: Seasonal forecasting of the Ethiopian summer rains, International Journal of Climatology, 24, 1345-1358, 10.1002/joc.1078, 2004.

Hertig, E., and Jacobeit, J.: Predictability of Mediterranean climate variables from oceanic variability. Part I: Sea surface temperature regimes, Climate Dynamics, 36, 811-823, 10.1007/s00382-010-0819-x, 2011a.

Hertig, E., and Jacobeit, J.: Predictability of Mediterranean climate variables from oceanic variability. Part II: Statistical models for monthly precipitation and temperature in the Mediterranean area, Climate Dynamics, 36, 825-843, 10.1007/s00382-010-0821-3, 2011b.

Jolliffe, I.: Principal component analysis, Wiley Online Library, 2002.

Kirtman, B. P., Min, D., Infanti, J. M., Kinter, J. L., Paolino, D. A., Zhang, Q., van den Dool, H., Saha, S., Mendez, M. P., Becker, E., Peng, P., Tripp, P., Huang, J., DeWitt, D. G., Tippett, M. K., Barnston, A. G., Li, S., Rosati, A., Schubert, S. D., Rienecker, M., Suarez, M., Li, Z. E., Marshak, J., Lim, Y.-K., Tribbia, J., Pegion, K., Merryfield, W. J., Denis, B., and Wood, E. F.: The North American Multimodel Ensemble: Phase-1 Seasonal-to-Interannual Prediction; Phase-2 toward Developing Intraseasonal Prediction, Bulletin of the American Meteorological Society, 95, 585-601, 10.1175/BAMS-D-12-00050.1, 2014.

Korecha, D., and Barnston, A. G.: Predictability of June–September Rainfall in Ethiopia, Monthly Weather Review, 135, 628-650, 10.1175/mwr3304.1, 2007.

Omondi, P., Ogallo, L. A., Anyah, R., Muthama, J. M., and Ininda, J.: Linkages between global sea surface temperatures and decadal rainfall variability over Eastern Africa region, International Journal of Climatology, 33, 2082-2104, 10.1002/joc.3578, 2013.

Segele, Z. T., Richman, M. B., Leslie, L. M., and Lamb, P. J.: Seasonal-to-Interannual Variability of Ethiopia/Horn of Africa Monsoon. Part II: Statistical Multi-Model Ensemble Rainfall Predictions, Journal of Climate, 150129124820009, 10.1175/jcli-d-14-00476.1, 2015.

Suárez-Moreno, R., and Rodríguez-Fonseca, B.: S$^4$CAST v2.0: sea surface temperature based statistical seasonal forecast model, Geosci. Model Dev., 8, 3639-3658, 10.5194/gmd-8-3639-2015, 2015.

von Storch, H., and Zwiers, F. W.: Statistical analysis in climate research, Cambridge University Press, Cambridge, 1999.

Wilks, D. S.: Statistical methods in the atmospheric sciences, Academic press, 2011.

Zhang, Y., Moges, S., and Block, P.: Optimal Cluster Analysis for Objective Regionalization of Seasonal Precipitation in Regions of High Spatial-Temporal Variability: Application to Western Ethiopia, Journal of Climate, 29, 3697-3717, 10.1175/Jcli-D-15-0582.1, 2016.

---

## Author Comment (AC3) · 17 May 2017

The authors proposed local level seasonal predictions of precipitation in the Western Ethiopia using the statistical approach at eight homogeneous regions clustered using k-means clustering technique. Large scale climate variables are used as potential predictors in developing the statistical model and unique set of predictors were assigned for each region. Results are compared with the dynamical prediction at regional scale and reported as the statistical approach is superior. This study is timely and helpful for the study region where rainfall is highly variable in space. The manuscript is well written and easy to understand and follow. I think this study will be a nice addition however a few moderate issues need to be resolved first. Please see my comments below.

1. The statistical approach is fully data driven approach that depends on the quality and length of the data. So how efficient is this technique in the area where there is sparse and poor quality data in the case of developing country like Ethiopia? The authors did not show how good the gridded rainfall data is through either validating with gauged data for selected weather stations or previous literatures that support the quality of this data.

We thank the reviewer for the comment. The dataset we use has been shown to reproduce station data over areas with both densely and sparsely distributed station networks. The original data is at a 10-day time interval, and we aggregate it to JJAS seasonal total precipitation, which may help offset some random errors and better represent the observations. Regarding the technique itself, as the cluster analysis and statistical model are purely data-driven, the data length and quality is essential to produce skillful results. With time, as the data length and availability improve, results are expected to become more skillful.

To justify the data quality, the following texts are added to the manuscript (Page 4 Line 7):
"…1983–2011 (29 years). This product has been verified against station data and has been deemed representative of observed precipitation in western Ethiopia (Dinku et al., 2014). "

And in the discussion (Page 15 Line 10):

"With additional data length in the future, the data-driven cluster analysis and statistical modeling approach are also expected to produce more confident, and perhaps skillful, results."

2. The author argue that this study gives perdition of seasonal precipitation at high resolution in the region. However, the classification of the homogeneous regions by NMA, Ethiopia, for the study region is almost equivalent (Koricha et al., 2007, pg 7685). I do not see any benefits of this study interms of the spatial resolutions at regional scale for Western Ethiopia.

We thank the reviewer for the comment. We agree that classifications of homogeneous regions are similar but still different. The prediction at regional scale in this study helps to verify our classification method, which is distinct from NMA's classification (see details in Zhang et al. (2016)). The regional-scale prediction also serves as an intermediate predictand for the grid-scale prediction under the indirect case. However, like the reviewer indicated, the highlight of the work should be the high-resolution prediction at grid scale.

3. No effort have been made on finding out the time lag between the predictor variables and the seasonal rainfall in the study area. For example which month of sea surface temperature really affects the seasonal rainfall in the study area.

We confirm that only one lead time is investigated in this work, and mention in the discussion that longer prediction lead times and evaluation of other relevant characteristics (e.g. intra-seasonal dry spells, seasonal onset or cessation, etc.) warrant future attention. However, for the purposes of this study, we consider only one lead time and instead focus on whether cluster analysis serves as a useful precursor to seasonal precipitation prediction at the local scale. Additional lead times can also be

readily applied to the current framework and are likely to be informative as to which months in the season-ahead are most related to JJAS seasonal precipitation.

4.   I don't see any comparison of the result with the current operational NMA forecast in the manuscript, however, it is reported in the conclusion section asif the result is compared at the regional scale with NMA operational forecast.

We have compared our results and NMA's results qualitatively only, and briefly mention in the discussion that at the regional scale, our results are more skillful than NMA's prediction, based on (Korecha and Sorteberg, 2013). We have revised the sentence to avoid confusion:

"At the regional scale, the approach shows qualitatively comparable results with current NMA operational forecasts, demonstrating moderately more skill than climatology (Korecha and Sorteberg, 2013)"

Upon the reviewer and editor's request, we can add more details comparing against NMA's prediction for each clustered region.

5.   The abstract is too short and mainly focused on the merit of conducting this study. It would be good if it is supported with some finding. There is no introduction given about the study area interms of the rainfall patter, topography etc.

We thank the reviewer for the comment. More content is added to the original manuscript (also included here):

In introduction:

"Precipitation in western Ethiopia peaks in the summer with approximately 70% of annual total precipitation falling during the main raining season - also known as the Kiremt season spanning from June to September (JJAS). On average, the seasonal total precipitation in the study region is approximately 760 mm; however in the northwest, precipitation can exceed 1200 mm (Fig. 1a). Along with the high spatial variability in this mountainous region, the temporal variability is also significant with spatial-average

seasonal total precipitation ranging from 650 mm in dry years up to 900 mm in wet years (Fig. 1b). These highly variable spatial and temporal precipitation patterns have made skillful seasonal predictions challenging, particularly at local scales. ”

[Figure]

Figure 1: Spatial and temporal variability of June-September seasonal total precipitation in western Ethiopia: (a) spatial pattern of temporal-average, and (b) spatial-average time series.

In abstract:

“… makes clear advances in modeling methodology and resolution, as compared with previous studies. The statistical model prediction results show improvements over non-clustered case for some clusters. Among those clusters, Cluster 5, in agriculturally rich central-northwestern Ethiopia, performs best, with correlation and RPSS values of 0.5 and 27.4%, respectively. The general skill of dynamical models over the entire study region is higher than statistical models, although dynamic models produce a lower resolution prediction. However, for some specific clustered regions such as Cluster 5, statistical model outperforms dynamical models with grid-level predictions demonstrating higher correlations and RPSS in a finer resolution. ”

Reference from reviewer's comments:
Korecha, D., and Sorteberg, A.: Validation of operational seasonal rainfall forecast in

Ethiopia, Water Resources Research, 49, 20 7681-7697, 10.1002/2013wr013760, 2013.

Reference from the author's responses:

Dinku, T., Hailemariam, K., Maidment, R., Tarnavsky, E., and Connor, S.: Combined use of satellite estimates and rain gauge observations to generate high-quality historical rainfall time series over Ethiopia, International Journal of Climatology, 34, 2489-2504, 10.1002/joc.3855, 2014.
Korecha, D., and Sorteberg, A.: Validation of operational seasonal rainfall forecast in Ethiopia, Water Resources Research, 49, 7681-7697, 10.1002/2013wr013760, 2013.
Zhang, Y., Moges, S., and Block, P.: Optimal Cluster Analysis for Objective Regionalization of Seasonal Precipitation in Regions of High Spatial-Temporal Variability: Application to Western Ethiopia, Journal of Climate, 29, 3697-3717, 10.1175/Jcli-D-15-0582.1, 2016.

---

## Author Response (AR1)

Response to Reviewer 1

The manuscript aims at developing a statistically based seasonal precipitation forecast model for Western Ethiopia. The target area is separated into homogeneous regions by means of a k-means cluster analysis of summer precipitation amounts. Eight regions with similar precipitation variability are defined. For each of them, a linear regression based forecast model is calibrated. Results are compared with a general forecast for the entire region and are found to be superior. In a final step the forecast is downscaled to a high resolution grid, again by means of a liner regression approach. The target of the study is timely, since local precipitation predictions are often required for water management and planning, and the manuscript is well structured and easy to follow. However I have some serious concerns about the calibration and particularly the evaluation of the statistical model. Further I would recommend to give some detailed information on the climate characteristics of the cluster regions and the major large scale influences.

1) Introduction, clustering and different predictor variables: An introduction into the climate of the target region is missing. Further, a detailed analysis of the precipitation characteristics of each cluster would be a basis for the interpretation of the modeling results. Some of the precipitation time series in Fig. 5 look highly correlated. Are simple statistical techniques really able to forecast those slight differences? And if, which predictor variables are responsible for the spatial variations of precipitation in Western Ethiopia? An analysis of the predictor-predictant relationships for each cluster would not only give some insights into the model structure and the large scale climate mechanisms of the target area, but also help to support (or scrutinize) the results of the modeling exercise.

We thank the reviewer for the comment. To address it, firstly, we provide additional description of the climate in the study region. The texts are inserted into Section 2 "Application to western Ethiopia and objectives of the study" (Page 3 Line 2):

"Precipitation in western Ethiopia peaks in the summer with approximately 70% of annual total precipitation falling during the main raining season - also known as the *Kiremt* season spanning from June to September (JJAS). On average, the seasonal total precipitation in the study region is approximately 760 mm; however in the northwest, precipitation can exceed 1200 mm (Fig. 1a). Along with the high spatial variability in this mountainous region, the temporal variability is also significant with spatial-average seasonal total precipitation ranging from 650 mm in dry years up to 900 mm in wet years (Fig. 1b). These highly variable spatial and temporal precipitation patterns have made skillful seasonal predictions challenging, particularly at local scales (e.g. Gissila et al., 2004, Block and Rajagopalan, 2007)."

[Figure]

(a)                                                      (b)

Figure 1: Spatial and temporal variability of June-September seasonal total precipitation in western Ethiopia: (a) spatial pattern of temporal-average, and (b) spatial-average time series.

Secondly, the precipitation characteristics of each cluster is described in a previous publication on cluster analysis (Zhang et al., 2016), however we agree that a brief summary should be provided here to help set the content and make the work more integrated. Therefore, we have added the text below to the original manuscript at the end of Section 3.1 "Cluster analysis" (Page 4 Line 26):

"The mean time series of each cluster illustrates high intra-correlation within the cluster and low inter-correlation between any two clusters, indicating strong coherency of the clustering results. For a detailed analysis including a complete correlation table and unique patterns for each cluster-level time series associated with large climate variables, readers are referred to Zhang et al. (2016)."

As we can see from the analysis in Zhang et al. (2016), some of the clusters contain stronger tele-connections to equatorial Pacific SSTs (an indicator of El Niño or La Niña), while other clusters are more affected by regional/local climate variables such as the pressure systems surrounding the African continent. This motivates us to use cluster analysis as a precursor to find proper predictors for each homogeneous region, which can capture the differences between the targeted predictand in each cluster.

Additional discussion on which predictor variables may be responsible for the spatial variations of precipitation in western Ethiopia is provided based on a combination of the previous concurrent connection to large-climate variable analysis (Zhang et al., 2016) and the prediction results in the discussion section (Page 15 Line 24):

Response to Reviewer 1

"A previous study (Zhang et al., 2016) has shown that the influence of ENSO on JJAS precipitation in western Ethiopia decreases generally from north to south, and is likely one of the reasons why skills are relatively low in southwestern Ethiopia. Cluster 5 was also identified with the strongest connection to equatorial Pacific SST (Zhang et al., 2016), which is consistent with the highest skill found in this study. Other regions with low prediction skill show relatively strong connections to SST and pressure systems in neighboring oceanic regions. However, connections with those climate patterns appear to be less robust than with ENSO, making the predictions in their associated regions less skillful."

While additional analysis into the predictor-predictand relationship is clearly possible, we believe associating precipitation patterns with concurrent climate variables provides a solid inference into this relationship and strong support for explaining climate mechanism.

2) Calibration of the statistical model and overfitting: Correlations between crossvalidated modeling results and observations in the order of 0.7-0.85 are very high (in fact they exceed the skills of well known forecast models) and are questionable. I believe, that those results are due to overfitting (particularly due to the predictor selection). The predictors for each of the clusters are selected based on all years, the cross-validation is only performed for the calibration of the linear model. In order to fully evaluate the model skill, the predictor selection must be included in the cross-validation (i.e. chose predictors at each step of the cross validation, e.g. based on a correlation threshold). Most likely the model skill will significantly drop, I could imagine that a step wise selection of predictors might slightly improve the results.

We thank the reviewer for the insightful comment. We admit that the model is overfit due to our methodological procedure in the predictor selection process as the reviewer notes. Therefore, we re-performed the entire process including predictor selection and regression based on cross-validation; that is, we dropped the target year when creating the global correlation map to search for predictors. As a result, there are total of 1044 (29 x 9 x 4) global correlation maps given 29-year time-series, 8 clusters plus 1 non-cluster scenario, and 4 climate variables. Hence, we developed a program to help automatically select highly correlated and justifiable regions as predictors. A description of the method is added to Section 3.2 "Statistical modeling approach" (Page 6 Line 11) and also included here with new figures:

"To avoid overfitting, the entire process including predictor selection and statistical modeling is processed using cross-validation. To start, drop-one-year precipitation observations for JJAS averaged across the region and each cluster are spatially correlated independently with each global climate variable. As a result, there are total of 1044 global correlation maps given the 29-year time-series, eight clusters plus one non-cluster, and four climate variables. Hence, a program to automatically select highly correlated and justifiable regions as predictors is developed. The following steps describe the subsequent statistical modeling process (Fig. 3):

(1) Grid-cells within each justifiable region (e.g. equatorial Pacific; Fig. 4) with correlation above the 99% significance level are identified (Fig. 5).

(2) The top 10% of the identified grid-cells with the highest correlation in each region is then selected, in order to boost the potential model skill.

(3) For each region, data of the selected grid-cells within the region are spatially averaged (defined as "pre-predictors").

(4) Pre-predictors are combined and transformed (for each cluster or non-cluster, and each dropped year analysis separately) through principal component analysis (PCA; Jolliffe, 2002).

(5a) The top principal components (PCs) from the PCA with a total of 95% variance explained are used as predictors – the direct inputs into the MLR model, otherwise known as the principal component regression (PCR). For the *direct* case, PCR is used to directly predict the grid-level precipitation; for the *indirect* case, PCR is used to predict the intermediate cluster-level precipitation.

(5b) For the *indirect* case only, cluster-level predictions are regressed to the grid-level. Note that the downscaling of cluster-level predictions to grid-level predictions is also cross-validated to avoid overfitting."

[Figure]

Figure 4: Justifiable climate regions globally for selecting predictors: (a) For SLP and GH at 500 mb with regions including EP, ES, LO, AH, SH, MH, and AM. For SAT, only LO is included. (b) For SST with regions including EP, NI, SI, and AT. *Note: EP - equatorial Pacific region, ES – Tahiti island for ENSO measurement, LO - local region, AH - Azores High, SH - St Helena High, MH - Mascarene High, AM - SW Asian Monsoon, NI - North Indian Ocean, SI - South Indian Ocean, AT - Equatorial/South Atlantic Ocean.*

[Figure]

Figure 5: Correlation map between mean JJAS seasonal precipitation time series in Cluster 5 and global SST under cross-validation, with correlations lower than 99% significance level masked out (one-tail test).

The results are also updated accordingly. As the reviewer expected, the model skill decreases significantly. Under the same model of PCR, the correlations of cluster-level prediction and observation now range from -0.180 to 0.504 (compared to 0.683 - 0.838 originally), with Cluster 5 having the highest correlation while Cluster 6 showing the lowest. Similarly for RPSS, 5 out of 8 clusters show skillful prediction compared to climatology (Table 2).  However, we do see improvement over non-cluster scenarios for some of the clusters.

Table 2: Correlation coefficients (Corr.) and RPSS for predictions (drop-one-year cross-validated) at cluster level compared to observations under C-I and NC-I scenario. (PCR model)

| Cluster | C1 | C2 | C3 | C4 | C5 | C6 | C7 | C8 | Non-cluster |
|---------|------|--------|-------|-------|-------|--------|-------|--------|-------------|
| Corr. | 0.163 | -0.010 | 0.179 | 0.188 | 0.504 | -0.180 | 0.351 | -0.122 | 0.297 |
| RPSS(%) | 33.41 | -21.66 | 43.01 | 12.46 | 27.40 | -37.79 | 20.63 | -55.96 | 13.25 |

We have also tried stepwise regression with forward selection and backward elimination algorithm (von Storch and Zwiers, 1999). The skill does increase a little for some cluster, but other clusters had a deterioration of prediction skills.  One table showing the results from stepwise regression with forward

selection probability of 0.05 and backward elimination probability of 0.05 (can also be understood as significance levels) is presented below.

Table 2b: Correlation coefficients (Corr.) and RPSS for predictions (drop-one-year cross-validated) at cluster level compared to observations under C-I scenario. (Stepwise model)

| Cluster | 1 | 2 | 3 | 4 | 5 | 6 | 7 | 8 |
|---|---|---|---|---|---|---|---|---|
| Corr. | 0.154 | -0.094 | 0.175 | 0.248 | 0.508 | -0.197 | 0.305 | -0.044 |
| RPSS | -3.31 | -44.17 | 40.60 | -49.89 | 33.24 | -77.10 | 25.00 | -24.87 |

Comparing to PCR, stepwise regression overfits some specific predictors, such as the SST in the equatorial Pacific region or SLP for the St. Helena High, whereas PCR extracts the main signal first and then fits with a regression model. As a result, less noise is left in the predictors (PCs) using PCR than those in the stepwise regression. Hence, PCR is more reliable, regardless of the prescribed rule for selecting a certain number of predictors, while stepwise regress is extremely sensitive to different prescriptions of significance levels of selecting and eliminating probability. Therefore, we consider PCR a more reliable method in this case and keep the results from PCR for this work with added discussion on stepwise at the end. Table 2 above is included in the revised manuscript together with other updated tables, figures and result analysis, which are also included here:

(Page 10 Line 5) "Correlations between cluster-level model predictions and observations range from - 0.18 to 0.50, with Cluster 5 having the highest correlation and Cluster 6 the lowest (Table 2). In approximately 1/5 of the 29 years, the observation falls outside the prediction envelope (Fig. 7), indicating model overfitting and an inability of the predictors to capture precipitation variability. For RPSS, 5 out of 8 clusters indicate superior prediction skill over climatology (Table 2). Improvement in terms of RPSS over the non-cluster scenario is evident for Cluster 1, 3, 5 and 7. Among all clusters, Cluster 5, in agriculturally rich central-northwestern Ethiopia (Fig. 2), performs the best, with correlation and RPSS values of 0.5 and 27%, respectively. Cluster 2, 4, 6 and 8, however, show deteriorated RPSS compared to non-cluster scenario, although those clusters are mainly regions outside Ethiopia and southern Ethiopia (Fig. 2) where water resources and agricultural activities are considerably less (Fig. 1)."

Response to Reviewer 1

[Figure]

Figure 7: Cluster-level predictions and observations under C-I and NC-I scenario, with drop-one-year cross-validation. The 95% envelope shows the 95% confidence interval constructed using model errors.

(Page 11 Line 5) "At the grid scale, depending on the case (*direct* or *indirect)*, and for different clusters, correlations between predictions and observations can favor the *clustered* case or the *non-clustered* case (Fig. 8). In general, the *indirect* model provides a smoother pattern of correlations, with grid-cells showing a negative correlation in the *direct* case now improved to near or above zero (Fig. 8). For example, Cluster 5 under the *indirect* case illustrates a more consistent positive correlation within the cluster. Some parts of the region reach a correlation over 0.6, such as central-northwestern Ethiopia (Cluster 5), which is consistent with the region of high cluster-level prediction skill. The percentage of grid-cells with correlations passing the 95% significance test is the highest for the NC-D case (Table 3); however, some locations demonstrate the lowest skills among all four scenarios."

"

[Figure]

Figure 8: Pearson correlations between grid-level observations and predictions under four scenarios, with the clustering boundary delineated roughly in black.

(Page 12 Line 3) "Similar findings are evident by evaluating the RPSS except for Cluster 8; instead of improving with increased RPSS in the *indirect* case, the grid-scale predictions deteriorate given poor cluster-level prediction (for the C-I case). However, the percentage of grid-cells with positive RPSS values overall is still the highest for the C-I case (Table 3), indicating the *clustered indirect* case is superior in terms of the number of grid-cells with improved skill compared to using climatology, particularly for grid-cells associated with skillful intermediate cluster-level predictions. The predictions are most skillful for the same region of central-northwestern Ethiopia (Cluster 5; Fig. 9) with 87% of its grid-cells showing positive RPSS and a spatial average RPSS value of 15% under the C-I scenario (Table 4). "

Table 3: Grid-level Pearson correlation and RPSS statistics

| Statistical Model | Grid-level correlations | | | Grid-level RPSS | | |
|---|---|---|---|---|---|---|
| | mean | stdev | significant corr % | mean (%) | stdev (%) | positive RPSS % |
| NC-D | 0.128 | 0.258 | 19.3% | -5.21 | 27.0 | 42.8% |
| NC-I | 0.063 | 0.186 | 3.13% | -2.26 | 14.6 | 43.9% |
| C-D | 0.055 | 0.230 | 10.6% | -14.0 | 31.0 | 33.9% |
| C-I | 0.081 | 0.206 | 12.4% | -9.60 | 29.4 | 44.4% |
| **Dynamical Model** | | | | | | |
| (1) | -0.105 | 0.209 | 0.51% | -31.4 | 25.4 | 5.70% |
| (2) | 0.133 | 0.171 | 6.26% | -14.2 | 24.6 | 27.0% |
| (3) | 0.086 | 0.130 | 2.08% | -14.9 | 25.2 | 26.2% |
| (4) | 0.027 | 0.156 | 0.38% | -14.4 | 19.3 | 22.6% |
| (5) | 0.067 | 0.170 | 1.64% | -9.66 | 17.0 | 28.4% |
| (6) | 0.139 | 0.165 | 6.53% | -5.66 | 16.7 | 38.1% |
| (7) | 0.102 | 0.130 | 1.67% | -8.64 | 17.6 | 31.7% |
| (8) | 0.009 | 0.185 | 0.90% | -10.3 | 14.8 | 26.7% |
| (9) | 0.244 | 0.149 | 23.1% | -2.33 | 21.8 | 46.0% |
| (10) | 0.244 | 0.149 | 21.2% | -1.09 | 16.8 | 48.9% |

Table 4: Grid-level Pearson correlation and RPSS statistics for grid-cells within Cluster 5

| Statistical Model | Grid-level correlations | | | Grid-level RPSS | | |
|---|---|---|---|---|---|---|
| | mean | stdev | significant corr % | mean (%) | stdev (%) | positive RPSS % |
| NC-D | 0.378 | 0.211 | 60.7% | 19.1 | 22.9 | 80.3% |
| NC-I | 0.265 | 0.111 | 12.8% | 8.33 | 14.8 | 70.3% |
| C-D | 0.229 | 0.244 | 30.5% | 6.91 | 24.1 | 62.3% |
| C-I | 0.345 | 0.165 | 55.7% | 14.7 | 13.3 | 87.1% |
| **Dynamical Model** | | | | | | |
| (9) | 0.353 | 0.110 | 46.8% | 8.21 | 18.2 | 65.7% |
| (10) | 0.248 | 0.130 | 18.4% | 3.92 | 16.2 | 59.5% |

[Figure]

Figure 9: Grid-level RPSS (%) under four scenarios using climate variables as predictors, with the clustering boundary delineated roughly in black.

3) Evaluation of the Downscaling approach: As the predictor selection, the downscaling procedure is not included in the cross-validation. I recommend to conduct the crossevaluation for the entire modeling chain. That means, the predictor selection, cluster forecast and downscaling approach need to be calibrated based on (n-1) years, in order to forecast gridded precipitation for the remaining year.

In general, one should be aware, that there is a linear dependence of the cluster based and the gridded forecast. Thus, the downscaling approach will better reproduce the local climate, however the variability (drought and moist years) will be equal to the culstered result. The term "gridded" forecast is somehow misleading – I would prefer "downscaling of regional forecast"

The downscaling procedure is included in the cross validation. We apologize if this is not clear and added one sentence to the statistical model steps (Page 6 Line 29):

Response to Reviewer 1

"(5b) For the *indirect* case only, cluster-level predictions are regressed to the grid-level. Note that the downscaling of cluster-level predictions to grid-level predictions is also cross-validated to avoid overfitting."

We also revised "gridded forecast" to "prediction at grid level" and "downscaling of cluster-level to grid-level prediction" for more appropriate descriptions.

Detailed remarks: 1) The abstract is very short and could certainly be more informative (e.g. by including some results)

We thank the reviewer for the comment and we have revised the abstract with highlights on results as the reviewer suggested. The added texts are also included here:

"… makes clear advances in prediction skill and resolution, as compared with previous studies. The statistical model improves versus the non-clustered case or dynamical models for a number of specific clusters in northwestern Ethiopia, with some cluster having regional average correlation and RPSS values of approximately 0.5 and 27%, respectively. The general skill of the two best performing dynamical models over the entire study region is superior to that of the statistical models, although dynamical models issue predictions at a lower resolution."

2) The discussion of state of the art forecasting models in the introduction is very short. Particularly during recent years, several studies investigated the skill of statistical models for regional scale precipitation forecasts (some of them are even based on clustering or PCA). I would recommend to better discuss the literature and the advantage of your approach in the introduction. See for example:

Hertig, E. and Jacobeit, J.: Predictability of Mediterranean climate variables from oceanic variability. Part II: Statistical models for monthly precipitation and temperature in the Mediterranean area, Clim. Dynam., 36, 825–843, doi:10.1007/s00382-010- 0821-3, 2010.

Suárez-Moreno, R. and Rodríguez-Fonseca, B.: S4CAST v2.0: sea surface temperature based statistical seasonal forecast model, Geosci. Model Dev., 8, 3639–3658, doi:10.5194/gmd-8-3639-2015, 2015.

Gerlitz, L., Vorogushyn, S., Apel, H., Gafurov, A., Unger-Shayesteh, K. & Merz, B.: A statistically based seasonal precipitation forecast model with automatic predictor selection and its application to central and south Asia, HESS 20, 4605–4623 , doi:10.5194/hess-20-4605-2016 , 2016.

We thank the reviewer for the comment. The recommended literatures are closely related to our methodology, although they are not based on the same study region. We have added them to the current literature review collection under Section 3.2 "statistical modeling approach" (Page 5 Line 6) with the following texts inserted:

Response to Reviewer 1

"Many studies have investigated statistical models for seasonal climate prediction. These studies vary by pre-classification of predictor or predictand regime, predictor selection process, and statistical methods. For example, Hertig and Jacobeit (2011) investigate sea surface temperature (SST) regimes as potential predictors for subsequent precipitation and temperature in the Mediterranean region. Through techniques including multiple applications of PCA, 17 stationary SST regimes were identified. Gerlitz et al. (2016) apply a k-means cluster analysis to grid-cells identified with significant correlations in the predictor field in order to facilitate predictor selection. Suárez-Moreno and Rodríguez-Fonseca (2015) investigate stationarity based on a long time series using a 21-year moving correlation window. The statistical prediction models are then applied to each stationary period respectively and the entire period for comparison. Despite diverse methods in seasonal prediction, multiple linear regression (MLR) is favored by many as a statistical modeling approach given its well-developed theory, simple model structure, efficient processing, and often skillful outcomes (e.g. Omondi et al., 2013, Camberlin and Philippon, 2002, Diro et al., 2008). As mentioned, only a few studies have focused on seasonal precipitation prediction in Ethiopia (Gissila et al., 2004, Block and Rajagopalan, 2007, Korecha and Barnston, 2007, Diro et al., 2008, Diro et al., 2011, Segele et al., 2015), and almost all of them include the applications of MLR. This study also applies MLR to predict seasonal precipitation, yet differentiates from other studies by applying predictions to pre-defined homogeneous regions and further translating to local-level predictions."

3) The predictor selection is based on correlation maps, and regions with potential forecast skill are identified (see Tab.1). Please map the regions and show the correlation maps for some clusters.

We have added one figure showing regions for selecting predictors and one correlation map for a cluster as an example. Please refer to Figure 4 and 5.

4) P7,l10: PCAs are cross-validated. This is somehow unclear to me. PCA is usually used for dimension reduction. Is the cross-validation done for the loadings of the pca in order to investigate how these change based on different input data?

PCR is cross-validated as indicated on Page 7 Line 17. To be specific, we performed PCA on the time series with the target-year data dropped, and then use the resultant PCs as predictors for the regression model. After the regression coefficient estimates are obtained, the principal components for the dropped year are *reconstructed*, and then multiplied with the coefficient estimates respectively in order to obtain the final predicted value for the dropped year. Please see more details in Wilks (2011).

5) Dynamical Models: The section on dynamical models is poorly integrated. Please give some more information on the models in general. If (as expected) the skill of the statistical model drops as a consequence of the cross-evaluation, a more detailed comparison of skills might be interesting.

Response to Reviewer 1

We apologize for our poorly integrated section on dynamical model and have improved it with an additional introduction of the NMME models (Page 8 Line 22):

"The North American Multi-Model Ensemble (NMME; Kirtman et al., 2014) is an experimental multi-model system consisting of coupled dynamical models from various modeling centers in North America. To our knowledge, it is also the most extensive multi-model seasonal prediction archive. The NMME provides gridded climate predictions that cover regions globally and with different lead times. The hindcasts of monthly mean precipitations are easily accessible through the International Research Institute for Climate and Society (IRI) website (http://iridl.ldeo.columbia.edu/SOURCES/.Models/.NMME/), and can be easily aggregated to seasonal totals for comparison with the statistical model results in this study."

We agree that after the model is revised, the skills of statistical model and dynamical model are comparative now. We have updated with Table 3 and 4 (see above), and provided additional result analysis with comparison between them. Texts are also copied here (Page 13 Line 7):

"The RPSS values based on the prediction ensembles of each dynamical model improve significantly after bias correction. The median RPSS values over all the grid-cells are now close to zero (Fig. 10) with two models, NASA-GMAO and NCEP-CFSv2, showing the highest RPSS value (-2.3% and -1.1%, respectively; Table 3). These two dynamical models also exhibit generally higher grid-level correlations over the study region (averaging 0.24 for both models; Table 3 and Fig. 11), as compared with other NMME models. The two best performing dynamical models after bias correction show advantage over statistical models, as assessed by correlation and RPSS metrics; however, all other dynamical models are inferior to the statistical models under NC-D and C-I scenarios, particularly given the percent of grid-cells with significant correlation and positive RPSS metrics (Table 3).

Within a certain cluster, statistical models may perform better than all dynamical models. For example, for Cluster 5, all statistical models show higher average RPSS values than that of all dynamical models (Table 4). The percentage of grid-cells with significant correlation reaches 61% for the statistical model under NC-D scenario, compared to the highest value of 47% among all the dynamical models. Similarly, the percentage with positive RPSS achieves 87% under C-I scenario as opposed to 66% for dynamical models. Note that the dynamical models also produce predictions in a lower spatial resolution (1°×1°) than the statistical models (0.1°×0.1°)."

6) P9,l15: Please give some more information on the performance measures (BIC, AIC, GCV).

We have eliminated the performance measures BIC AIC GCV, as the revised predictor selection rule produces a dynamic number of predictors depending on which years is dropped and for different clusters (for the *indirect* case) or grid-cells (for the *direct* case).

Response to Reviewer 1

7) P10: How exactly is the envelope (uncertainty interval) calculated? Is this based on the assumption that cross-validated residuals of the regression are normal distributed?

Yes. Q-Q plots are evaluated to verify normally distributed residuals (results not included) as indicated on Page 8 Line 3. To make it clear, we have added the following text to the same line:

"A 95% confidence interval of the cross-validated predictions is also constructed conditioned on model errors. Q-Q plots are evaluated to verify normally distributed residuals (results not included)."

[Figure]

Figure 1: Spatial and temporal variability of June-September seasonal total precipitation in western Ethiopia: (a) spatial pattern of temporal-average, and (b) spatial-average time series.

In abstract:

"… makes clear advances in prediction skill and resolution, as compared with previous studies. The statistical model improves versus the non-clustered case or dynamical models for a number of specific clusters in northwestern Ethiopia, with some cluster having regional average correlation and RPSS values of approximately 0.5 and 27%, respectively. The general skill of the two best performing dynamical models over the entire study region is superior to that of the statistical models, although dynamical models issue predictions at a lower resolution."

Reference from reviewer's comments:

Korecha, D., and Sorteberg, A.: Validation of operational seasonal rainfall forecast in

Ethiopia, Water Resources Research, 49, 20 7681-7697, 10.1002/2013wr013760, 2013.

Reference from the author's responses:

[revised manuscript text omitted]

**2 Application to western Ethiopia and objectives of the study**

Precipitation in western Ethiopia peaks in the summer with approximately 70% of annual total precipitation falling during the main raining season - also known as the *Kiremt* season spanning from June to September (JJAS). On average, the seasonal total precipitation in the study region is approximately 760 mm; however in the northwest, precipitation can exceed 1200 mm (Fig. 1a). Along with the high spatial variability in this mountainous region, the temporal variability is also significant with spatial-average seasonal total precipitation ranging from 650 mm in dry years up to 900 mm in wet years (Fig. 1b). These highly variable spatial and temporal precipitation patterns have made skillful seasonal predictions challenging, particularly at local scales (e.g. Gissila et al., 2004, Block and Rajagopalan, 2007).

[Figure]

(a)                                            (b)

Figure 1: Spatial and temporal variability of June-September seasonal total precipitation in western Ethiopia: (a) spatial pattern of temporal-average, and (b) spatial-average time series.

[revised manuscript text omitted]

**3.1 Cluster analysis**

Using a k-means clustering technique, western Ethiopia – the major agricultural region of the country – is divided into eight homogeneous regions (Fig. 2+), conditioned on the interannual variability of total precipitation in JJAS, the same variable that is to be predicted. Precipitation is based on a 0.1˚×0.1˚ gridded precipitation dataset from NMA (Dinku et al., 2014), consisting of 7320 grid-cells across 1983–2011 (29 years). This product has been verified against station data and has been deemed representative of observed precipitation in western Ethiopia (Dinku et al., 2014). Given the high-resolution gridded dataset, k-means clustering is performed for a range of predefined numbers of clusters; the optimal number of clusters is identified by comparing the within-cluster sum of square errors (WSS). During the clustering process, each grid-cell is assigned and reassigned to clusters until the WSS is minimized. This does not require any subjective delineation or manual delineation of boundaries between clustered stations or grid-cells; instead, an automated and objective delineation is performed. The mean time series of each cluster illustrates high intra-correlation within the cluster and low inter-correlation between any two clusters, indicating strong coherency of the clustering results. For a detailed analysis including a complete correlation table and unique patterns for each cluster-level time series associated with large climate variables, readers are referred to Zhang et al. (2016).

[Figure]

**Figure 2: Regionalization map of 8 homogeneous regions marked by different colors, with country boundary and river profile. After Zhang et al. (2016)**

**3.2 Statistical modeling approach**

5   Many studies have investigated statistical models for seasonal climate prediction. These studies vary by pre-classification of predictor or predictand regime, predictor selection process, and statistical methods. For example, Hertig and Jacobeit (2011) investigate sea surface temperature (SST) regimes as potential predictors for subsequent precipitation and temperature in the Mediterranean region. Through techniques including multiple applications of PCA, 17 stationary SST regimes were identified. Gerlitz et al. (2016) apply a k-means cluster analysis to grid-cells identified with significant correlations in the predictor field in

10  order to facilitate predictor selection. Suárez-Moreno and Rodríguez-Fonseca (2015) investigate stationarity based on a long time series using a 21-year moving correlation window. The statistical prediction models are then applied to each stationary period respectively and the entire period for comparison. Despite diverse methods in seasonal prediction, multiple linear regression (MLR)  is favored by many as a statistical modeling approach given its well-developed theory, simple model structure, efficient processing, and often skillful outcomes (e.g. Omondi et al., 2013, Camberlin and Philippon, 2002, Diro

15  et al., 2008). As mentioned, only a few studies have focused on seasonal precipitation prediction in Ethiopia (Gissila et al., 2004, Block and Rajagopalan, 2007, Korecha and Barnston, 2007, Diro et al., 2008, Diro et al., 2011b, Segele et al., 2015), and almost all of them include the applications of MLR. This study also applies MLR to predict seasonal precipitation, yet differentiates from other studies by applying predictions to pre-defined homogeneous regions and further translating to local-level predictions.

20  Large-scale climate variables are often evaluated as potential predictors in statistical seasonal precipitation prediction models, commonly including sea surface temperatures (SST) in the equatorial Pacific Ocean representing the well-known of the El Nino-Southern Oscillation (ENSO) (Stone et al., 1996). For Ethiopia, the ENSO phenomenon is considered a significant indicator of

precipitation variability, particularly in the main JJAS rainy season (e.g. NMSA, 1996, Camberlin, 1997, Bekele, 1997, Segele and Lamb, 2005, Diro et al., 2011a, Elagib and Elhag, 2011). In addition to ENSO, the effect of Indian Ocean SST and regional atmospheric pressure systems such as the St. Helena, Azores, and Mascarene Highs also have notable influence on Ethiopia's precipitation variability (e.g. Kassahun, 1987, Tadesse, 1994, NMSA, 1996, Shanko and Camberlin, 1998, Goddard and Graham, 1999, Latif et al., 1999, Black et al., 2003, Segele and Lamb, 2005). Consequently, season-ahead (March-May) or month-ahead (May) large-scale climate variables that are physically relevant in potentially modulating moisture transport to the basin (or cluster) are selected as potential predictors. Four climate variables are selected here for further evaluation based on outcomes of the aforementioned prediction studies: SST, sea level pressure (SLP), geopotential height (GH) at 500mb, and surface air temperature (SAT). All climate variables are from the National Centers for Environmental Prediction and National Center for Atmospheric Research (NCEP/NCAR) reanalysis dataset (Kalnay et al., 1996) at a 2.5˚×2.5˚ grid scale.

To avoid overfitting, the entire process including predictor selection and statistical modeling is processed using cross-validation. To start, drop-one-year precipitation observations for JJAS averaged across the region and each cluster are spatially correlated independently with each global climate variable. As a result, there are total of 1044 global correlation maps given the 29-year time-series, eight clusters plus one non-cluster, and four climate variables. Hence, a program to automatically select highly correlated and justifiable regions as predictors is developed. The following steps describe the subsequent statistical modeling process (Fig. 3):

(1) Grid-cells within each justifiable region (e.g. equatorial Pacific; Fig. 4) with correlation above the 99% significance level are identified (Fig. 5).

(2) The top 10% of the identified grid-cells with the highest correlation in each region is then selected, in order to boost the potential model skill.

(3) For each region, data of the selected grid-cells within the region are spatially averaged (defined as "pre-predictors").

(4) Pre-predictors are combined and transformed (for each cluster or non-cluster, and each dropped year analysis separately) through principal component analysis (PCA; Jolliffe, 2002).

(5a) The top principal components (PCs) from the PCA with a total of 95% variance explained are used as predictors – the direct inputs into the MLR model, otherwise known as the principal component regression (PCR). For the *direct* case, PCR is used to directly predict the grid-level precipitation; for the *indirect* case, PCR is used to predict the intermediate cluster-level precipitation.

(5b) For the *indirect* case only, cluster-level predictions are regressed to the grid-level. Note that the downscaling of cluster-level predictions to grid-level predictions is also cross-validated to avoid overfitting.

[Figure]

**Figure 2̶3: Flow chat of data processing for predictors into the statistical model. Numbers framed by dash lines correspond to the procedures listed in the context. Note: pre. – precipitation, t-s – time-series, avg. – average.**

[Figure]

**Figure 4: Justifiable climate regions globally for selecting predictors: (a) For SLP and GH at 500 mb with regions including EP, ES, LO, AH, SH, MH, and AM. For SAT, only LO is included. (b) For SST with regions including EP, NI, SI, and AT. Note: EP - equatorial Pacific region, ES – Tahiti island for ENSO measurement, LO - local region, AH - Azores High, SH - St Helena High, MH - Mascarene High, AM - SW Asian Monsoon, NI - North Indian Ocean, SI - South Indian Ocean, AT - Equatorial/South Atlantic Ocean.**

[Figure]

**Figure 5: Correlation map between mean JJAS seasonal precipitation time series in Cluster 5 and global SST under cross-validation, with correlations lower than 99% significance level masked out (one-tail test).**

[revised manuscript text omitted]

At the grid scale, correlations between predictions and observations are clearly superior for the *clustered* case versus the *non-clustered* case (Fig. 6). Some parts of the region reach a correlation of 0.9, such as central-northwestern Ethiopia, which is consistent with the region of high cluster-level prediction skill (*Cluster 5*). The average correlation over all grid cells is approximately 0.51 (*direct*) and 0.53 (*indirect*) for *clustered* predictions, compared to 0.24 (*direct*) and 0.27 (*indirect*) for the *non-clustered* predictions (Table 4), although spatial differences are clearly apparent (Fig. 6). In addition to higher average correlations, standard deviations of correlations are also lower in the *clustered* case than in the *non-clustered* case, indicating a more concentrated correlation distribution for these higher values. The percentage of grid cells with correlations passing the 95% significance test increases from approximately 30% in the *non-clustered* case to more than 80% in the *clustered* case (Table 4). At the grid scale, depending on the case (*direct* or *indirect*), and for different clusters, correlations between predictions and observations can favor the *clustered* case or the *non-clustered* case (Fig. 8). In general, the *indirect* model provides a smoother pattern of correlations, with grid-cells showing a negative correlation in the *direct* case now improved to near or above zero (Fig. 8). For example, Cluster 5 under the *indirect* case illustrates a more consistent positive correlation within the cluster. Some parts of the region reach a correlation over 0.6, such as central-northwestern Ethiopia (Cluster 5), which is consistent with the region of high cluster-level prediction skill. The percentage of grid-cells with correlations passing the 95% significance test is the highest for the NC-D case (Table 3); however, some locations demonstrate the lowest skills among all four scenarios.

|  | **Non–clustered** | **Clustered** |
|---|---|---|
| **Direct** |
[Figure]
 | |
| **Indirect** | | |

[Figure]

**Figure 68: Pearson correlations between grid-level observations and predictions under four scenarios, with the clustering boundary delineated roughly in black.**

**Table 43: Grid-level Pearson correlation and RPSS statistics**

| Statistical Model | Grid-level correlations | | | Grid-level RPSS | | |
|---|---|---|---|---|---|---|
| | mean | stdev | significant corr % | mean (%) | stdev (%) | positive RPSS % |
| NC-D | 0.237 | 0.245 | 28.1% | 5.42 | 18.46 | 58.8% |
| NC-I | 0.272 | 0.247 | 32.3% | 5.32 | 17.09 | 60.7% |
| C-D | 0.509 | 0.172 | 80.8% | 19.16 | 19.18 | 84.4% |
| C-I | 0.532 | 0.146 | 87.1% | 26.47 | 21.47 | 90.0% |
| Dynamical Model | | | | | | |
| (9) NASA GMAO | 0.300 | 0.149 | 36.1% | 2.32 | 21.20 | 54.3% |
| (10) NCEP CFSv2 | 0.310 | 0.155 | 37.3% | 3.66 | 16.61 | 61.0% |

| Statistical Model | Grid-level correlations | | | Grid-level RPSS | | |
|---|---|---|---|---|---|---|
| | mean | stdev | significant corr % | mean (%) | stdev (%) | positive RPSS % |
| NC-D | 0.128 | 0.258 | 19.3% | -5.21 | 27.0 | 42.8% |
| NC-I | 0.063 | 0.186 | 3.13% | -2.26 | 14.6 | 43.9% |
| C-D | 0.055 | 0.230 | 10.6% | -14.0 | 31.0 | 33.9% |
| C-I | 0.081 | 0.206 | 12.4% | -9.60 | 29.4 | 44.4% |

**Dynamical Model**

| | | | | | | |
|---|---|---|---|---|---|---|
| (1) | -0.105 | 0.209 | 0.51% | -31.4 | 25.4 | 5.70% |
| (2) | 0.133 | 0.171 | 6.26% | -14.2 | 24.6 | 27.0% |
| (3) | 0.086 | 0.130 | 2.08% | -14.9 | 25.2 | 26.2% |
| (4) | 0.027 | 0.156 | 0.38% | -14.4 | 19.3 | 22.6% |
| (5) | 0.067 | 0.170 | 1.64% | -9.66 | 17.0 | 28.4% |
| (6) | 0.139 | 0.165 | 6.53% | -5.66 | 16.7 | 38.1% |
| (7) | 0.102 | 0.130 | 1.67% | -8.64 | 17.6 | 31.7% |
| (8) | 0.009 | 0.185 | 0.90% | -10.3 | 14.8 | 26.7% |
| (9) | 0.244 | 0.149 | 23.1% | -2.33 | 21.8 | 46.0% |
| (10) | 0.244 | 0.149 | 21.2% | -1.09 | 16.8 | 48.9% |

~~Similar findings are evident by evaluating the RPSS. The *non-clustered* predictions are modestly skillful, particularly for the same region of central-northwestern Ethiopia (Fig. 7), with an average RPSS of approximately 5.4% (both *direct* and *indirect*) over the entire study region, however RPSS values improve nearly fourfold to 19.2% and 26.5% in the *clustered* case (*indirect* and *direct*, respectively). Additionally, the percentage of grid-cells with positive RPSS values reaches 84.4% - 90.0% in the *clustered* case (Table 4).~~

Similar findings are evident by evaluating the RPSS except for Cluster 8; instead of improving with increased RPSS in the *indirect* case, the grid-scale predictions deteriorate given poor cluster-level prediction (for the C-I case). However, the percentage of grid-cells with positive RPSS values overall is still the highest for the C-I case (Table 3), indicating the *clustered indirect* case is superior in terms of the number of grid-cells with improved skill compared to using climatology, particularly for grid-cells associated with skillful intermediate cluster-level predictions. The predictions are most skillful for the same region of central-northwestern Ethiopia (Cluster 5; Fig. 9) with 87% of its grid-cells showing positive RPSS and a spatial average RPSS value of 15% under the C-I scenario (Table 4).

Table 4: Grid-level Pearson correlation and RPSS statistics for grid-cells within Cluster 5

| **Statistical Model** | Grid-level correlations | | | Grid-level RPSS | | |
|---|---|---|---|---|---|---|
| | mean | stdev | significant corr % | mean (%) | stdev (%) | positive RPSS % |
| NC-D | 0.378 | 0.211 | 60.7% | 19.1 | 22.9 | 80.3% |
| NC-I | 0.265 | 0.111 | 12.8% | 8.33 | 14.8 | 70.3% |
| C-D | 0.229 | 0.244 | 30.5% | 6.91 | 24.1 | 62.3% |
| C-I | 0.345 | 0.165 | 55.7% | 14.7 | 13.3 | 87.1% |
| **Dynamical Model** | | | | | | |
| (9) | 0.353 | 0.110 | 46.8% | 8.21 | 18.2 | 65.7% |
| (10) | 0.248 | 0.130 | 18.4% | 3.92 | 16.2 | 59.5% |

|  | **Non-clustered** | **Clustered** |
|---|---|---|
| **Direct** |
[Figure]
 | |
| **Indirect** | | |

[Figure]

|  | Non-Clustered | Clustered |
|---|---|---|
| Direct | | |
| Indirect | | |

**Figure 9: grid-level RPSS (%) under four scenarios using climate variables as predictors, with the clustering boundary delineated roughly in black.**

5 ~~At the grid-scale, predictions by the *indirect* approach generally outperform *direct* approach predictions, based on AIC, BIC and GCV values (Table 5), as well as correlation and RPSS values (Tables 4). Using the predicted cluster-level precipitation to predict grid-level precipitation (the *indirect* case) appears to help reduce the effect of over-fitting and smooth grid-scale noise. From another perspective, the results also suggest that precipitation signals at the regional scale are better explained by large-scale climate variables, while at highly localized scales the signal is less evident. Obviously, however, this is dependent on cluster size~~

10

|  |  | |  | |  | |
|---|---|---|---|---|---|---|
| |  |  |  |  |  |  |
|  |  |  |  |  |  |  |
|  |  |  |  |  |  |  |
|  |  |  |  |  |  |  |
|  |  |  |  |  |  |  |

**4.2 Dynamical model predictions**

The RPSS values based on the prediction ensembles of each dynamical model improve significantly after bias correction The median RPSS values over all the grid-cells are  now close to zero (Fig. 10) .  two models, NASA-GMAO and

NCEP-CFSv2, showing the highest RPSS value (2.3% and 1.1%, respectively; Table 4). These two dynamical models also exhibit generally higher grid-level correlations over the study region (averaging 0.24 for both models; Table 4 and Fig. 11), as compared with other NMME models. The two best performing dynamical models after bias correction show advantage over statistical models, as assessed by correlation and RPSS metrics; however, all other dynamical models are inferior to the statistical models under NC-D and C-I scenarios, particularly given the percent of grid-cells with significant correlation and positive RPSS metrics (Table 3).

Within a certain cluster, statistical models may perform better than all dynamical models. For example, for Cluster 5, all statistical models show higher average RPSS values than that of all dynamical models (Table 4). The percentage of grid-cells with significant correlation reaches 61% for the statistical model under NC-D scenario, compared to the highest value of 47% among all the dynamical models. Similarly, the percentage with positive RPSS achieves 87% under C-I scenario as opposed to 66% for dynamical models. Note that the dynamical models also produce predictions in a lower spatial resolution (1°×1°) than the statistical models (0.1°×0.1°).

[Figure]

**Figure 10: Boxplots of grid-level RPSS (%) for 10 dynamical models from NMME (a) before and (b) after bias correction, labeled with the same number as listed in the context. Note: For each box plot, the line inside the box is the median, the box edges represent the 25th and 75th percentiles, and the whiskers extend to the most extreme data points not considered outliers (outliers not shown).**

[Figure]

**Figure 11: Pearson correlations between grid-level observations and ensemble mean of bias-corrected predictions for 10 dynamical models from NMME, labeled with the same number as listed in the context. Note that the scale ranges from -1 to 1.**

**5 Conclusions and discussion**

5    This study demonstrates the potential for season-ahead large-scale climate information to produce skillful and credible high-resolution precipitation predictions under a *clustered indirect* approach in western Ethiopia.  At the regional scale, the approach shows promise for northwestern Ethiopia (Cluster 1, 3, 5, and 7), particularly compared to current NMA operational forecasts, which are only moderately more skillful than climatology (Korecha and Sorteberg,

10   2013). The regional average RPSS in this study under the *clustered* case ranges from 21% to 43% for northwestern Ethiopia, as opposed to values under 6% for NMA operational forecast (Korecha and Sorteberg, 2013). The approach adopted here also advances on previous studies (Gissila et al., 2004, Block and Rajagopalan, 2007, Korecha and Barnston, 2007, Diro et al., 2011b, Segele et al., 2015) by first applying an objective cluster analysis and then conditionally constructing high-resolution predictions. A unique set of predictors is applied to each cluster, which contributes to superior prediction performance at  cluster

15   levels in northwestern Ethiopia, as compared with predictions from the *non-clustered* approach. Grid-level prediction under the *clustered* indirect case also reduces the effect of over-fitting relative to the *direct* case and improves negative RPSS values to near or above zeros; that said, the *non-clustered direct* case also illustrates higher correlation and RPSS values on average.

20    Two out of 10 NMME dynamical models, NASA-GMAO and NCEP-CFSv2, demonstrate overall superior performance to the statistical models; however, for certain regions such as Cluster 5, the performance of statistical models under *clustered indirect* and *non-clustered direct* cases is still superior. It is also worth noting that the statistical model predictions are at a one hundred times finer spatial resolution than the dynamical models providing additional advantages at the local scale, when skillful. Nevertheless,

improvements in dynamical models continue and their application to seasonal precipitation prediction is likely to grow (e.g. Palmer et al., 2004, Saha et al., 2006, Lim et al., 2009). ~~Multi model combinations of statistical and dynamical models were also investigated for potential improvement of prediction skills through pooling, linear regression, and Bayesian model averaging (BMA; Raftery et al., 1997) using the best statistical model (C-I) and two best dynamical models (NASA GMAO and NCEP CFSv2), however, the overall performance was inferior to the single statistical C-I model (results not shown).~~

Relatively poor prediction performance is evident in some locations such as southwestern Ethiopia and regions outside Ethiopia,

10  where the hydroclimatic processes that produce precipitation  might be driven by  local factors or other regional climate patterns rather than large-scale climate variables identified in this study. A previous study (Zhang et al., 2016) has shown that the influence of ENSO on JJAS precipitation in western Ethiopia decreases generally from north to south, and is likely one of the reasons why skills are relatively low in southwestern Ethiopia. Cluster 5 was

15 also identified with the strongest connection to equatorial Pacific SST (Zhang et al., 2016), which is consistent with the highest skill found in this study. Other regions with low prediction skill show relatively strong connections to SST and pressure systems in neighboring oceanic regions. However, connections with those climate patterns appear to be less robust than with ENSO, making the predictions in their associated regions less skillful. To test the prospects for improving prediction performance by including season-ahead local variables, soil moisture and spring rains were investigated; however, no significant improvement was found:

20  in fact, performance skill deteriorates when adding local predictors may simply serve to introduce more noise and encourage over-fitting.

Additional prediction features also warrant future attention, including longer prediction lead times and evaluation of other relevant

25 characteristics (e.g. intra-seasonal dry spells, seasonal onset or cessation, etc.). As observational datasets continue to grow, data-driven cluster analysis and statistical modeling approaches may be expected to improve. Improving predictive capabilities may not be a complete panacea, but it can continue to be an important part of a decisions-maker's portfolio as they cope with hydroclimatic variability and its inherent risks.

**6 Data availability**

30 The National Centers for Environmental Prediction and National Center for Atmospheric Research (NCEP/NCAR) reanalysis dataset can be accessed through the National Oceanic & Atmospheric Administration (NOAA) Earth System Research Laboratory (ESRL) website (https://www.esrl.noaa.gov/psd/data/reanalysis/).

The NMME hindcasts are available through the International Research Institute for Climate and Society (IRI) website

35 (http://iridl.ldeo.columbia.edu/SOURCES/.Models/.NMME/).

The gridded precipitation dataset in western Ethiopia is available upon request from NMA (http://www.ethiomet.gov.et/).

**Competing interests**

The authors declare that they have no conflict of interest.

**Acknowledgements**

This study was supported by NASA Project NNX14AD30G and NSF PIRE Project 1545874. We acknowledge the National
Meteorological Agency of Ethiopia for sharing data.

---

## Author Response (AR2)

The revised version of the manuscript by Zhang et al. definitively improved from a methodological point of view. The predictor selection is now based on a procedure, which automatically identifies pixels of large scale climate variables, which significantly correlate with summer precipitation anomalies. In a next step, the pixel anomalies in specific regions are averaged and different variables are combined by means of a pca analysis. PCA timeseries (explaining 95% of the overall variance), are eventually utilized as predictors for a linear regression based prediction. The procedure is now fully cross-validated which results in poor to moderate prediction results.

While the application of the method is solid (beside of some smaller issues, see specific remarks), the results of the statistical forecast model are worse compared with some dynamical models (and sometimes even worse compared with climatology). Beside of two clusters all correlations between observations and cross-validated forecast are below 0.2, which is not statistically significant for a period of 30 years. Thus in my opinion the study is not yet ready for publication in HESS. However, I see quite a big potential of the study from a hydro-climatological point of view. The major question for me is, why there is such a big difference in the skill of the forecast models, although the clusters are very close to each other. This could either be a statistical artefact or it could be explained by large/regional scale climate mechanisms. I believe that the results are robust and physically interpretable (all clusters with moderate forecast skill are windward of the East-African monsoon), but this should be somehow investigated. In the revised manuscript the authors give some more information on the general climate characteristics and also mention, that different cluster regions are correlated with different climate modes and sst patterns. I still feel that this insufficient for the interpretation of the results. I suggest to give a short literature overview of climatic mechanisms in the introduction. Which anomalies (pressure patterns, moisture fluxes etc) trigger precipitation surplus and deficits? How are these patterns related to ENSO and other potential predictors? There is quite a lot literature on the monsoon in general.

What kind of predictors are important for which cluster? Maybe this could eventually explain the different model skills and support the robustness of your models. Further it would definitively justify the clustering procedure. My impression from the gridded analysis is that it doesn't make a difference?

We truly appreciate the reviewer's help in improving this work. As the reviewer suggested, we added a more detailed literature review of climatic mechanisms affecting JJAS precipitation in Ethiopia at Page 5 Line 21:

"Large-scale climate variables are often evaluated as potential predictors in statistical seasonal precipitation prediction models, commonly including sea surface temperatures (SST) in the equatorial Pacific Ocean representing the well-known of the El Nino-Southern Oscillation (ENSO) (Stone et al., 1996). Sea level pressure (SLP) in the eastern Pacific Ocean at Tahiti as an critical and stable component for measuring an ENSO index (Torrence and Webster, 1999) warrants another potential predictor. For Ethiopia, the ENSO phenomenon is considered a primary indicator of precipitation variability, particularly in the main JJAS rainy season with El Nino/La Nina often associated with deficit/excess of precipitation amount in the study region (e.g. NMSA, 1996, Camberlin, 1997, Bekele, 1997, Segele and Lamb, 2005, Diro et al., 2011, Elagib and Elhag, 2011). Evidences have also shown a more direct moisture transport from the Gulf of Guinea (equatorial Atlantic Ocean), the Indian Ocean, and the Mediterranean Sea affecting Ethiopia's summer precipitation (Viste and Sorteberg, 2013a, Viste and Sorteberg, 2013b). Those moisture fluxes are often related to pressure patterns across the continent. For instance, the St. Helena high over the southern Atlantic Ocean or a high pressure over Gulf of Guinea, and a simultaneous low pressure over

Indian Ocean or a monsoon trough over Arabic Peninsula bring intensified westerlies and south-westerlies that transport moist air across the Congo Basin to the western Ethiopian highlands in the summer (Segele et al., 2009, Williams et al., 2011). Similarly, the southwest Asian monsoon at the Indian Ocean, which has a strong positive relationship with the concurrent JJAS rainfall in the western Ethiopia, is associated with the Mascarene high over the southern Indian Ocean and a low pressure system near Bombay. During this monsoon season, the southeast trades in the southern hemisphere are channeled by the east African highlands while crossing the equator and become a southwest monsoon flow. It is further diverted by the Turkana Channel, enhancing convergence with the westerlies/south-westerlies above the western Ethiopian highlands and bringing moisture to this region (Kinuthia, 1992, Nicholson, 1996, Camberlin, 1997, Slingo et al., 2005, Segele et al., 2009, Nicholson, 2014). In addition, the effect of other climate variables relevant to the aforementioned driven factors, such as the Indian Ocean SST, local and other regional atmospheric pressure systems such as Azores High also have notable influence on Ethiopia's precipitation variability (e.g. Kassahun, 1987, Tadesse, 1994, NMSA, 1996, Shanko and Camberlin, 1998, Goddard and Graham, 1999, Latif et al., 1999, Black et al., 2003, Segele and Lamb, 2005).

Consequently, season-ahead (March-May) or month-ahead (May) large-scale climate variables that are physically relevant in potentially modulating moisture transport to the basin (or cluster) are selected as potential predictors. Four climate variables are selected here for further evaluation based on outcomes of the aforementioned prediction studies: SST, SLP, geopotential height (GH) at 500mb, and surface air temperature (SAT). All climate variables are from the National Centers for Environmental Prediction and National Center for Atmospheric Research (NCEP/NCAR) reanalysis dataset (Kalnay et al., 1996) at a 2.5˚×2.5˚ grid scale. "

Regarding to the reasons why the prediction skills are different across the clusters, it is likely due to the complex climate mechanisms affecting this region, causing great spatial heterogeneity, although we cannot rule out the possibility that the linear structure of the model cannot capture some non-linear relationships, if any, between the climate predictors and JJAS precipitation in some clusters. As the reviewer suspects, the differences in skills are possibly related to the East-African monsoon (the southwest Asian monsoon over Indian Ocean) given that the clusters with moderate prediction skills lie on the windward pathway of the East-African monsoon. We inspected the concurrent correlation maps with sea level pressure (SLP) for each cluster and found that Cluster 5 and 7 – the two clusters with the best skills – are the only ones that are strongly and negatively corrected with SLP near Bombay (indicating a strong East-African monsoon is associated with higher JJAS precipitation amount, and vice versa), and meantime strongly and positively correlated with the SLP at eastern equatorial Pacific Ocean (indicating high surface pressure often accompanied with cold sea surface temperature and a raining pattern over the eastern equatorial Pacific Ocean – a La Nina phenomenon – somehow brings higher JJAS precipitation in western Ethiopia, and vice versa, i.e. El Nino associated with lower precipitation). Cluster 2 – one of the worst predicted clusters – shows moderately strong negative correlation with SLP near Bombay; however, it is also correlated strongly and negatively with SLP at southern Indian Ocean (indicating a possible weak gradient of East-African monsoon) and its correlation with SLP over equatorial Pacific Ocean is nonsignificant. Considering in general El Nino suppresses the monsoon and La Nina increases it (Kumar et al., 2006), the connections with both ENSO and monsoon in the right direction such as for Cluster 5 and 7, indicate a double insurance over their connection with the East-African monsoon. Therefore, we conclude

that clusters which are more affected by East-African monsoon, particularly coupled with the influence of ENSO, show more promises in prediction skills.

Additional discussions are provided at Page 16 Line 22:

"Relatively poor prediction performance is evident in some locations such as southwestern Ethiopia and regions outside Ethiopia, where the hydroclimatic processes that produce precipitation might be driven by local factors or other regional climate patterns rather than large-scale climate variables identified in this study. A previous study (Zhang et al., 2016) has shown that the influence of ENSO on JJAS precipitation in western Ethiopia decreases generally from north to south, and is likely one of the reasons why skills are relatively low in southwestern Ethiopia. Cluster 5 was also identified with the strongest connection to equatorial Pacific SST (Zhang et al., 2016), which is consistent with the highest skill found in this study. Other regions with low prediction skill show relatively strong connections to SST in neighboring oceanic regions. However, connections with those climate patterns appear to be less robust than with ENSO, making the predictions in their associated regions less skillful. This is also consistent with the findings from other studies that even though all three oceans (Indian, Atlantic, and Pacific Ocean) affect the JJAS precipitation in western Ethiopia, the Pacific Ocean still plays the greatest role (Segele et al., 2009, Omondi et al., 2013).

The southwest Asian monsoon over Indian Ocean may also be critical in determining the precipitation, given that the clusters with better prediction skills lie along the pathway of the monsoon. Based on the global concurrent correlation maps between JJAS precipitation and SLP for each cluster, Cluster 5 and 7 – the two clusters with the best skills – are the only ones that are strongly and negatively corrected with SLP near Bombay, and meantime strongly and positively correlated with the SLP at the eastern equatorial Pacific Ocean. The former indicates that a strong southwest Asian monsoon is associated with higher JJAS precipitation amount, and vice versa. The latter indicates that a high surface pressure over the eastern equatorial Pacific Ocean often accompanied with cold SST and a raining pattern – a La Nina phenomenon – also brings higher JJAS precipitation to the western Ethiopia, and vice versa. Cluster 2 – one of the worst predicted clusters – shows moderately strong negative correlation with SLP near Bombay; however, it is also correlated strongly and negatively with SLP at southern Indian Ocean, indicating a possible weak gradient of the southwest Asian monsoon. Moreover, its correlation with SLP over equatorial Pacific Ocean is nonsignificant. Considering in general El Nino suppresses the monsoon and La Nina increases it (Kumar et al., 2006), strong correlations with both ENSO and the monsoon in the correct direction, such as for Cluster 5 and 7, indicate a double insurance over their association with the southwest Asian monsoon. Therefore, clusters which are more affected by the southwest Asian monsoon over Indian Ocean, particularly coupled with the influence of ENSO, are likely to show more promises in their prediction skills."

We admit that except two clusters all correlations between observation and the cross-validated forecast are below 0.2; however, we want to point out that Cluster 5 and 7, which show correlations of 0.51 and 0.35 respectively, occupy a large portion of area inside Ethiopia, which is also the agricultural-richest region. Besides, we can also see positive RPSS values for Cluster 1, 3, and 4, although their correlations are below 0.2. As mentioned in the discussion section, for most clusters inside Ethiopia, the predictions in this study are superior to the current NMA operational forecast at regional scale. As observational

datasets continue to grow, this data-driven cluster analysis and statistical modeling approach may also be expected to improve.

Regarding to the gridded analysis, we want to emphasize that the grid-level prediction would be useful for operational purpose with more accurate spatial representation. While comparing to bias-corrected grid-level predictions based on dynamical models, our results show higher skills over some critical region that has a great deal of agricultural activities, such as Cluster 5. Gridded analysis also enables us to compare across *indirect* and *direct* method: whether cluster-level prediction can be interpreted properly to grid-level prediction helps to justify the clustering analysis – whether the clusters are robust and coherent. More responses regarding to gridded analysis are provided given the reviewer's minor remarks below.

Minor remarks:

- Fig. 1: Please show boarders and rivers to make orientation easier. The seasonal precipitation amounts might be rather shown as a barplot.

We appreciate the reviewer's suggestion and included an updated Fig. 1 in the manuscript (also included here):

[Figure]

**Figure 1: Spatial and temporal variability of June-September seasonal total precipitation in western Ethiopia: (a) spatial pattern of temporal-average, and (b) spatial-average time series.**

- P3L24 (and others): for me the term "scenario" stands for future climate change assessments. Do you mean similar large scale conditions? Likewise in Sec. 3 the term is misleading.

To avoid misunderstanding, we have changed to "climate condition" instead of "climate scenario". As for Sec. 3 and beyond, we kept the word as to our understanding "scenario" can also be used as "model scenarios".

- There's a bit more information on the clustering procedure now. However the selection of k is still not clear. WSS is smallest for the largest possible k, right? I see that the authors refer to their previous publication, but very general information should be provided.

We thank the reviewer for the comments. The following sentence is added to provide more information on the selection of *k* (Page 4 Line 25):

"…; the optimal number of clusters is identified by various evaluation metrics based on the within-cluster sum of square errors (WSS), including elbow method with *difference in WSS*, gap statistic with *difference in difference*, and qualitative analysis on post-visualization of clusters."

- I was a bit lost in the section on the statistical modelling approach. Would it be possible to structure it into predictor selection, model calibration and evaluation?

As model calibration (cross-validation) penetrates the entire statistical modeling process, we find it hard to structure it as suggested. Besides, model evaluation is for both statistical and dynamical models and is therefore separated as a third section "*Performance metrics*" after statistical and dynamical approaches.

- Predictor regions: Why are those regions chosen (Maybe a broader literature review could justify the selection). Please also name the regions in the map.

Thank you. We have provided a broader literature review as in the very first response above and updated the map (also included here):

[Figure]

**Figure 4: Justifiable climate regions globally for selecting predictors: (a) For SLP and GH at 500 mb with regions including EP, ES, LO, AH, SH, MH, and AM. For SAT, only LO is included. (b) For SST with regions including EP, NI, SI, and AT. Note: EP - equatorial Pacific region, ES – Tahiti island for ENSO measurement, LO - local region, AH - Azores High, SH -**

**St Helena High, MH - Mascarene High, AM - SW Asian Monsoon, NI - North Indian Ocean, SI - South Indian Ocean, AT - Equatorial/South Atlantic Ocean.**

\- Predictor selection approach: In step 3, the "best" grid cells are spatially averaged. This is not clear to me. If there are positive and negative correlations in the region, this might average out the predictive skill. For example in the southern Indian Ocean, both positive and negative correlations are detected in Fig. 5. Likewise the North Atlantic SLP domain, which contains not only the Azores (high) but also Iceland (low), might be problematic. Wouldn't it be straight forward, to use all gridcells directly for the PCA analysis?

We thank the reviewer for the comment. For regions containing grid-cells with both positive and negative correlations, the number of grid-cells with significant correlation in each sign is counted. If a greater number of grid-cells is associated with significant positive correlation, for example, we would average over only grid-cells with positive correlations, and vice versa. We use regional average instead of single grid-cell to hopefully filter out some noises.

To avoid confusion, the following sentence is added to the revised manuscript (Page 6 Line 34):

"… Grid-cells within each justifiable region (e.g. equatorial Pacific; Fig. 4) with correlation above the 99% significance level are identified (Fig. 5). For regions containing grid-cells with both positive and negative correlations, the number of the identified grid-cells in each sign is counted. If a greater number of grid-cells is associated with significant positive correlation, for example, only grid-cells with positive correlations are kept for the following steps, and vice versa."

\- PCA: No information is given on standardization of predictor variables. If different variables are not standardized, the final PCA might be much more affected by SLP (values around 1000hpa) then SST (around 25°C).

We thank the reviewer for the comment. We did perform standardization first before using PCA. To make it clear, the following sentence is revised (Page 7 Line 1):

"Pre-predictors are standardized, combined, and transformed through principal component analysis (PCA; Jolliffe, 2002) for each cluster or non-cluster, and each dropped-year analysis separately."

\- Predictor selection: How many PCA-predictors are used in the linear models at the moment? Are all of them correlated with the predictant? (Theoretically, the stepwise procedure, which the authors tested for the original predictors (author response), could also be used for the PCA based predictors. The predictors used in the linear regression (most likely very few) might then be easier interpretable. Further it might reduce overfitting, which is judged by the cross-validation.)

The number of PCs used in one model could be different for different dropped year under cross-validation. It also differs for different clusters and different model scenarios. The criterion of how many of them are kept as predictors is that the total variance explained by them should reach 95% (as stated at Page 7 Line 3).

PCA extracts the most dominant signals from the group of potential predictors and is independent of the predictand; that is, the selection of PCs does not depend on their correlation with the predictand. Therefore, PCA helps to reduce artificial skills. Combining PCA and stepwise regression could potentially select the PCs with the least dominant signals (likely noises) as predictors, which makes PCA analysis meaningless and could cause overfitting (to noises).

- P10L18: Please only use the term significant if you conducted a test (otherwise remarkable, great etc)

Thank you. We have changed related terms to "remarkable/remarkably", "evident/evidently", or "great" at appropriate places. Please see track-changes document for details.

- Dynamical Models: I am not sure if it's really necessary to do that comparison and I still feel that it is poorly integrated. The heart of the study is the statistical approach – and I think one can easily argue, that statistical predictions are good for operational use (without comparing with complex statistical models?)

As statistical prediction with gridded high spatial resolution in this study is the very first one for this region, it is hard to find similar statistical product to compare with. The NMME dynamical model outputs, in the same format of gridded prediction, within the same region and same time frame (lead time and target season) are therefore used to investigate whether our prediction makes some progress compared to the existing ones. The NMME dynamical models also produce timely outputs available for operational use.

- Cluster Results (Fig. 7) : How is the standardization conducted? The y-axis should be limited to -2/2 to better present the results.

We first standardized the time-series in each grid-cell and then average over all time-series in each cluster and the non-clustered region. We realize that in this way, some information of the target year, which is used at first for standardization at grid level, could leak through the prediction process. We apologize for the mistake (only made for *clustered indirect* case) and have rerun entire process (including grid-level predictions based on the cluster-level prediction) to ensure correct presentation of the method and outcome, even though the grid-level results do not alter much (visualization of grid-level correlation and RPSS maps is almost the same as before).

We have updated figures, tables, and some texts in result analysis (please see track-change document for details). For Fig. 7, instead of presenting standardized results, we present the actual values predicted compared to observations:

[Figure]

**Figure 7: cluster-level predictions and observations under C-I and NC-I scenario, with drop-one-year cross-validation. The 95% envelope shows the 95% confidence interval constructed using model errors.**

-        Precipitation trends (Fig. 7): The two clusters with skillfull models (5 and 7) seem to have positive precipitation trends. This might lead to an overestimation of skill, since every variable with a similar trend could be used as a predictor. Thus I suggest to test the model for detrended time-series.

We appreciate the comment. We performed linear trend test using student-t statistics and found that Cluster 5 and 7 have a significant trend (slope significantly differs from zero) at 95% significance level, while trends for all others including non-cluster are insignificant.

Ideally, for Cluster 5 and 7, the trend test would be performed under cross-validation, detrend the data, predict, add the trend on, and repeat with other dropped years. It would be also ideal to detrend the global predictors for each pixel, each variable, and each period with a different dropped year. As only 29 years of data is used in this study, the conclusion on significant trend could be unreliable and often data over 30-40 years are recommended for trend analysis in this field. Considering both the complexity of the process and the length of our data, we decided to keep trend analysis aside from the modeling process, but mentioned in the discussion section that if longer time series is available, one should consider using trend analysis first before prediction. We added the following sentence in the revised manuscript (Page 17 Line 10):

"As observational datasets continue to grow, data-driven cluster analysis and statistical modeling approaches may be expected to improve. The growing length of time series and climate change impacts also call for careful analysis on possible significant trends in the data."

From the perspective of possible overestimating of the skills, even though when selecting predictors, a biased region with pseudo-high correlation may be selected, this is not likely to lead to an overestimation of skill, as the PCA detrends data at the first step and produces detrended PCs. Besides, as mentioned above, the PCA process for getting the final predictors is completely independent of predictand; that is, PCs that explain the largest portion of total variance of all selected variables (not predictand) will be remain as predictors. Moreover, cross-validation also avoids overestimated skills. Therefore, the skillful results from Cluster 5 and 7 are more likely due to the climate mechanisms affecting them.

-       P23: The argument that the statistical models are of higher resolution than dynamical models is misleading. One could downscale the results the same way, as it is done with the cluster results. Further (as I mentioned during the first review round), the high resolution (indirect) forecast has the same temporal variability as the cluster forecast (due to the univariate linear relationship) and thus does not contain much additional information. The analysis of different statistical relationships at the cluster-level seem to be more relevant (in my point of view).

We agree with the reviewer that one could downscale the results the same way. In fact, we have bias-corrected the dynamical model outputs using our observation data set, and the bias-corrected outputs are in the same resolution as the statistical products. What we meant by "higher resolution" is relative to the original dynamical product. To avoid miscommunications, we have reconstructed the argument to (in abstract and elsewhere):

"The general skill (after bias-correction) of the two best performing dynamical models over the entire study region is superior to that of the statistical models, although the dynamical models issue predictions at a lower resolution and the raw predictions requires bias correction to guarantee comparable skills."

We also agree on univariate linear relationship of cluster-level prediction and regressed grid-level predictions for the *clustered* and *non-clustered indirect* cases. While comparing across *indirect* and *direct*

cases for grid-level prediction show promises. For example, as mentioned in the manuscript, the grid-level predictions under *indirect* cases perform better than *direct* cases in terms of categorical evaluation metrics (RPSS), with a higher percentage of grids demonstrating positive RPSS values.

[Figure]

**Figure 1: Spatial and temporal variability of June-September seasonal total precipitation in western Ethiopia: (a) spatial pattern of temporal-average, and (b) spatial-average time series.**

[revised manuscript text omitted]

35   grid-cells across 1983–2011 (29 years). This product has been verified against station data and has been deemed representative of observed precipitation in western Ethiopia (Dinku et al., 2014). Given the high-resolution gridded dataset, k-means clustering is performed for a range of predefined numbers of clusters; the optimal number of clusters is identified by various evaluation metrics

based on the within-cluster sum of square errors (WSS), including elbow method with *difference in WSS*, gap statistic with *difference in difference*, and qualitative analysis on post-visualization of clusters.. During the clustering process, each grid-cell is assigned and reassigned to clusters until the WSS is minimized. This does not require any subjective delineation or manual delineation of boundaries

5  between clustered stations or grid-cells; instead, an automated and objective delineation is performed. The mean time series of each cluster illustrates high intra-correlation within the cluster and low inter-correlation between any two clusters, indicating strong coherency of the clustering results. For a detailed analysis including a complete correlation table and unique patterns for each cluster-level time series associated with large climate variables, readers are referred to Zhang et al. (2016).

[Figure]

**Figure 2: Regionalization map of 8 homogeneous regions marked by different colors, with country boundary and river profile. After Zhang et al. (2016)**

**3.2 Statistical modeling approach**

15  Many studies have investigated statistical models for seasonal climate prediction. These studies vary by pre-classification of predictor or predictand regime, predictor selection process, and statistical methods. For example, Hertig and Jacobeit (2011) investigate sea surface temperature (SST) regimes as potential predictors for subsequent precipitation and temperature in the Mediterranean region. Through techniques including multiple applications of PCA, 17 stationary SST regimes were identified. Gerlitz et al. (2016) apply a k-means cluster analysis to grid-cells identified with significant correlations in the predictor field in

20  order to facilitate predictor selection. Suárez-Moreno and Rodríguez-Fonseca (2015) investigate stationarity based on a long time series using a 21-year moving correlation window. The statistical prediction models are then applied to each stationary period respectively and the entire period for comparison. Despite diverse methods in seasonal prediction, multiple linear regression (MLR) is favored by many as a statistical modeling approach given its well-developed theory, simple model structure, efficient processing,

and often skillful outcomes (e.g. Omondi et al., 2013, Camberlin and Philippon, 2002, Diro et al., 2008). As mentioned, only a few studies have focused on seasonal precipitation prediction in Ethiopia (Gissila et al., 2004, Block and Rajagopalan, 2007, Korecha and Barnston, 2007, Diro et al., 2008, Diro et al., 2011b, Segele et al., 2015), and almost all of them include the applications of MLR. This study also applies MLR to predict seasonal precipitation, yet differentiates from other studies by applying predictions
5    to pre-defined homogeneous regions and further translating to local-level predictions.

Large-scale climate variables are often evaluated as potential predictors in statistical seasonal precipitation prediction models, commonly including sea surface temperatures (SST) in the equatorial Pacific Ocean representing the well-known of the El Nino-Southern Oscillation (ENSO) (Stone et al., 1996). Sea level pressure (SLP) in the eastern Pacific Ocean at Tahiti as an critical and
10   stable component for measuring an ENSO index (Torrence and Webster, 1999) warrants another potential predictor. For Ethiopia, the ENSO phenomenon is considered a primary indicator of precipitation variability, particularly in the main JJAS rainy season with El Nino/La Nina often associated with deficit/excess of precipitation amount in the study region (e.g. NMSA, 1996, Camberlin, 1997, Bekele, 1997, Segele and Lamb, 2005, Diro et al., 2011, Elagib and Elhag, 2011). Evidences have also shown a more direct moisture transport from the Gulf of Guinea (equatorial Atlantic Ocean), the Indian Ocean, and the Mediterranean Sea
15   affecting Ethiopia's summer precipitation (Viste and Sorteberg, 2013a, Viste and Sorteberg, 2013b). Those moisture fluxes are often related to pressure patterns across the continent. For instance, the St. Helena high over the southern Atlantic Ocean or a high pressure over Gulf of Guinea, and a simultaneous low pressure over Indian Ocean or a monsoon trough over Arabic Peninsula bring intensified westerlies and south-westerlies that transport moist air across the Congo Basin to the western Ethiopian highlands in the summer (Segele et al., 2009, Williams et al., 2011). Similarly, the southwest Asian monsoon at the Indian Ocean, which has
20   a strong positive relationship with the concurrent JJAS rainfall in the western Ethiopia, is associated with the Mascarene high over the southern Indian Ocean and a low pressure system near Bombay. During this monsoon season, the southeast trades in the southern hemisphere are channeled by the east African highlands while crossing the equator and become a southwest monsoon flow. It is further diverted by the Turkana Channel, enhancing convergence with the westerlies/south-westerlies above the western Ethiopian highlands and bringing moisture to this region (Kinuthia, 1992, Nicholson, 1996, Camberlin, 1997, Slingo et al., 2005,
25   Segele et al., 2009, Nicholson, 2014). In addition, the effect of other climate variables relevant to the aforementioned driven factors, such as the Indian Ocean SST, local and other regional atmospheric pressure systems such as Azores High also have notable influence on Ethiopia's precipitation variability (e.g. Kassahun, 1987, Tadesse, 1994, NMSA, 1996, Shanko and Camberlin, 1998, Goddard and Graham, 1999, Latif et al., 1999, Black et al., 2003, Segele and Lamb, 2005).

30   Consequently, season-ahead (March-May) or month-ahead (May) large-scale climate variables that are physically relevant in potentially modulating moisture transport to the basin (or cluster) are selected as potential predictors. Four climate variables are selected here for further evaluation based on outcomes of the aforementioned prediction studies: SST, SLP, geopotential height (GH) at 500mb, and surface air temperature (SAT). All climate variables are from the National Centers for Environmental Prediction and National Center for Atmospheric Research (NCEP/NCAR) reanalysis dataset (Kalnay et al., 1996) at a 2.5˚×2.5˚
35   grid scale.

40

Lamb, 2005, Diro et al., 2011a, Elagib and Elhag, 2011). In addition to ENSO, the effect of Indian Ocean SST and regional atmospheric pressure systems such as the St. Helena, Azores, and Mascarene Highs also have notable influence on Ethiopia's precipitation variability (e.g. Kassahun, 1987, Tadesse, 1994, NMSA, 1996, Shanko and Camberlin, 1998, Goddard and Graham, 1999, Latif et al., 1999, Black et al., 2003, Segele and Lamb, 2005). Consequently, season ahead (March-May) or month ahead (May) large-scale climate variables that are physically relevant in potentially modulating moisture transport to the basin (or cluster) are selected as potential predictors. Four climate variables are selected here for further evaluation based on outcomes of the aforementioned prediction studies: SST, sea level pressure (SLP), geopotential height (GH) at 500mb, and surface air temperature (SAT). All climate variables are from the National Centers for Environmental Prediction and National Center for Atmospheric Research (NCEP/NCAR) reanalysis dataset (Kalnay et al., 1996) at a 2.5°×2.5° grid scale.

To avoid overfitting, the entire process including predictor selection and statistical modeling is processed using cross-validation. To start, drop-one-year precipitation observations for JJAS averaged across the region and each cluster are spatially correlated independently with each global climate variable. As a result, there are total of 1044 global correlation maps given the 29-year time-series, eight clusters plus one non-cluster, and four climate variables. Hence, a program to automatically select highly correlated and justifiable regions as predictors is developed. The following steps describe the subsequent statistical modeling process (Fig. 3):

(1) Grid-cells within each justifiable region (e.g. equatorial Pacific; Fig. 4) with correlation above the 99% significance level are identified (Fig. 5). For regions containing grid-cells with both positive and negative correlations, the number of the identified grid-cells in each sign is counted. If a greater number of grid-cells is associated with significant positive correlation, for example, only grid-cells with positive correlations are kept for the following steps, and vice versa.

(2) The top 10% of the identified grid-cells with the highest correlation in each region is then selected, in order to boost the potential model skill.

(3) For each region, data of the selected grid-cells within the region are spatially averaged (defined as "pre-predictors").

(4)  Pre-predictors are standardized, combined, and transformed through principal component analysis (PCA; Jolliffe, 2002) for each cluster or non-cluster, and each dropped-year analysis separately.

(5a) The top principal components (PCs) from the PCA with a total of 95% variance explained are used as predictors – the direct inputs into the MLR model, otherwise known as the principal component regression (PCR). For the *direct* case, PCR is used to directly predict the grid-level precipitation; for the *indirect* case, PCR is used to predict the intermediate cluster-level precipitation.

(5b) For the *indirect* case only, cluster-level predictions are regressed to the grid-level. Note that the downscaling of cluster-level predictions to grid-level predictions is also cross-validated to avoid overfitting.

[Figure]

**Figure 3: Flow chat of data processing for predictors into the statistical model. Numbers framed by dash lines correspond to the procedures listed in the context. Note: pre. – precipitation, t-s – time-series, avg. – average.**

[Figure]

(a)

(b)

**Figure 4: Justifiable climate regions globally for selecting predictors: (a) For SLP and GH at 500 mb with regions including EP, ES, LO, AH, SH, MH, and AM. For SAT, only LO is included. (b) For SST with regions including EP, NI, SI, and AT. Note: EP - equatorial Pacific region, ES – Tahiti island for ENSO measurement, LO - local region, AH - Azores High, SH - St Helena High, MH - Mascarene High, AM - SW Asian Monsoon, NI - North Indian Ocean, SI - South Indian Ocean, AT - Equatorial/South Atlantic Ocean.**

[Figure]

**Figure 5: Correlation map between mean JJAS seasonal precipitation time series in Cluster 5 and global SST under cross-validation, with correlations lower than 99% significance level masked out (one-tail test).**

5   PCA is a common approach in climate modeling to reduce the dimensionality of predictors and remove multi-collinearity, while simultaneously extracting the most dominant signals from the potential predictors, typically reflected in the first few PCs. Since PCA is independent of the predictand, retaining the first few PCs as predictors, in lieu of the original variables, also helps to reduce artificial prediction skill.

10   PCR is performed in a "drop-one-year" cross-validation mode to reduce over-fitting effects and therefore avoid overestimation of prediction skill. This requires reconstructing the principal components for the dropped year, and then multiplying the coefficient estimates with each reconstructed PC respectively in order to obtain the final predicted value for the dropped year (e.g. Block and Rajagopalan, 2009, Wilks, 2011). A 95% confidence interval of the cross-validated predictions is also constructed conditioned on model errors. Q-Q plots are evaluated to verify normally distributed residuals (results not included).

For the four scenarios, the model structures are quite similar but have subtle differences which could lead to  evidently different outcomes (Table 1). Under the NC-D (Eq. (1a, b)) and C-D scenarios (Eq. (2a, b)), the time-series of JJAS seasonal total precipitation in each grid-cell (i.e. at local level) is used as the direct predictand ($Y_{i,t}$); however, the NC-D and C-D scenarios differ, as the former uses the same predictors ($X_t$) across all the grid-cells, while the latter uses different predictors according to the cluster

20   to which the grid-cell is assigned ($X_{j,t}$). In the indirect case, the cluster-level time-series of JJAS seasonal total precipitation (the time-series averaged over all grid-cells that belong to a given cluster, $Y_{m,t}$ or $Y_{j,t}$) is first predicted (Eq. (3a, b) and (4a, b)). The predicted intermediate product ($\widetilde{Y}_{m,t}$ or $\widetilde{Y}_{j,t}$) is then used as the only regressor in the second step to estimate the grid-level precipitation ($\widetilde{Y}_{i,t}$ or $\widetilde{Y}_{i\in j,t}$ for every j; Eq. (3c, d) and (4c, d)). Again, for the C-I scenario, predictors in the first step are unique for

each of the eight clusters and grid-cells within that cluster ($X_{j,t}$), while predictors are identical for all grid-cells ($X_t$) under the NC-I scenario.

**Table 1: Equations of linear regression panel models under four scenarios**

| | Non-clustered | | Clustered | |
|---|---|---|---|---|
| **Direct** | $Y_{i,t} = \widetilde{\alpha}_i + \widetilde{\beta}_i X_t + \varepsilon_{i,t}$ | ...... (1a) | $Y_{i\in j,t} = \widetilde{\alpha}_i + \widetilde{\beta}_i X_{j,t} + \varepsilon_{i,t}$ | ...... (2a) |
| | $\widetilde{Y}_{i,t} = \widetilde{\alpha}_i + \widetilde{\beta}_i X_t$ | ...... (1b) | $\widetilde{Y}_{i\in j,t} = \widetilde{\alpha}_i + \widetilde{\beta}_i X_{j,t}$ | ...... (2b) |
| **Indirect** | $Y_{m,t} = \widetilde{\alpha} + \widetilde{\beta} X_t + \varepsilon_t$ | ...... (3a) | $Y_{j,t} = \widetilde{\alpha}_j + \widetilde{\beta}_j X_{j,t} + \varepsilon_{j,t}$ | ...... (4a) |
| | $\widetilde{Y}_{m,t} = \widetilde{\alpha} + \widetilde{\beta} X_t$ | ...... (3b) | $\widetilde{Y}_{j,t} = \widetilde{\alpha}_j + \widetilde{\beta}_j X_{j,t}$ | ...... (4b) |
| | $Y_{i,t} = \widetilde{\eta}_i + \widetilde{\gamma}_i \widetilde{Y}_{m,t} + \nu_{i,t}$ | ...... (3c) | $Y_{i\in j,t} = \widetilde{\eta}_i + \widetilde{\gamma}_i \widetilde{Y}_{j,t} + \nu_{i,t}$ | ...... (4c) |
| | $\widetilde{Y}_{i,t} = \widetilde{\eta}_i + \widetilde{\gamma}_i \widetilde{Y}_{m,t}$ | ...... (3d) | $\widetilde{Y}_{i\in j,t} = \widetilde{\eta}_i + \widetilde{\gamma}_i \widetilde{Y}_{j,t}$ | ...... (4d) |

where Y- predictand of JJAS seasonal total precipitation; X- two predictors of top two PCs; $\varepsilon, \nu$ - error terms; $\widetilde{Y}$ - predicted values of JJAS seasonal total precipitation; $\widetilde{\alpha}, \widetilde{\beta}, \widetilde{\eta}, \widetilde{\gamma}$- estimated coefficients; i- grid-cell index; t- time (year) index; j- cluster index; i ∈ j- grid-cell i that belongs to cluster j; m- mean over entire study region that is equivalently the only one cluster.

**3.3 Dynamical modeling approach**

The North American Multi-Model Ensemble (NMME; Kirtman et al., 2014) is an experimental multi-model system consisting of coupled dynamical models from various modeling centers in North America. To our knowledge, it is also the most extensive multi-model seasonal prediction archive. The NMME provides gridded climate predictions that cover regions globally and with different lead times. The hindcasts of monthly mean precipitations are easily accessible through the International Research Institute for Climate and Society (IRI) website (http://iridl.ldeo.columbia.edu/SOURCES/.Models/.NMME/), and can be easily aggregated to seasonal totals for comparison with the statistical model results in this study. Therefore, NMME JJAS seasonal precipitation predictions (1˚×1˚ grid-cells) are extracted from model ensembles that cover the same time period (1983–2011), geographic region (western Ethiopia), and with the same lead time (predictions made on June 1). A subset of 10 NMME models meet these criteria and are retained for further evaluation: (1) COLA-RSMAS-CCSM3, (2) COLA-RSMAS-CCSM4, (3) GFDL-CM2p1, (4) GFDL-CM2p1-are04, (5) GFDL-CM2p5-FLOR-A06, (6) GFDL-CM2p5-FLOR-B01, (7) IRI-ECHAM-AnomalyCoupled, (8) IRI-ECHAM-DirectCoupled, (9) NASA-GMAO, (10) NCEP-CFSv2. The names are kept the same as on the International Research Institute for Climate and Society (IRI) data repository website.

The NMME predictions for each of the 10 models are bias-corrected by applying probability mapping (e.g. Block et al., 2009, Teutschbein and Seibert, 2012, Chen et al., 2013) under cross-validation, subject to the observational dataset from NMA (Fig. 6). This is performed on a grid-cell by grid-cell basis on standardized data (the NMME dataset is reshaped to 0.1°×0.1° grid-cells to match the observational NMA dataset grid-cell size). The basic steps include:

(1) Fit gamma distributions to drop-one-year time-series from each observed and NMME grid-cell; for NMME this is performed on an individual model basis using all ensemble members available. (Goodness-of-fit tests indicate gamma distributions are appropriate; results not shown.)

(2) Translate gamma distributions into cumulative distribution functions (CDF).

(3) For any given dynamical model prediction at the grid-cell level, a corrected prediction value is attained by mapping from the modeled CDF to the observed CDF and applying the inverse gamma distribution. This is repeated for all grid-cells, all NMME models, and all dropped years.

5   After correction, the gamma CDF of predictions and observations approximately match (Fig. 6a). Additionally, each ensemble still retains its variability over time, though the overall ensemble mean is shifted to closely match observation (Fig. 6b).

[Figure]

Figure 6: (a) bias correction of NMME predictions using probability mapping; (b) precipitation time-series from NMME (colored lines) before and after correction, compared to observations (black line). Examples are shown for randomly selected six grid-cells.

**3.4 Performance metrics**

Pearson correlations are used to measure the standardized covariance between observations and predictions. Ranked probability skill scores (RPSS; Wilks, 2011) are also evaluated to determine categorical skill based on probabilistic predictions. Here, the data are split into three equal terciles representing below-normal, near-normal, and above-normal conditions. A perfect prediction yields

15   an RPSS of 100%, and a prediction with less skill than climatology (long-term averages) yields an RPSS of less than zero. Median RPSS values from all 29 years are reported.

**4 Results**

**4.1 Statistical model predictions**

Correlations between cluster-level model predictions and observations range from -0.18 to 0.51, with Cluster 5 having the

20   highest correlation and Cluster 6 the lowest (Table 2). In approximately 1/5 of the 29 years, the observation falls outside the prediction envelope (Fig. 7), indicating model overfitting and an inability of the predictors to capture precipitation variability. For RPSS, 3 out of 8 clusters indicate superior prediction skill over climatology (Table 2). Improvement in terms of RPSS over the non-cluster scenario is evident for Cluster 1, 3 and 7.  Although Cluster 5, in agriculturally rich centralnorthwestern Ethiopia (Fig. 2), shows a slightly deteriorated RPSS relative to non-cluster scenario, it still performs outstandinglythe best, with the highest correlation and a positive RPSS values of 0.51 and 2710%, respectively. Cluster 2, 4, 6 and 8, however, show deteriorated RPSS compared to non-cluster scenario, although those clusters are mainly regions outside Ethiopia and southern Ethiopia (Fig. 2) where water resources and agricultural activities are considerably less (Fig. 1).

[Figure]

[Figure]

**Figure 7: cluster-level predictions and observations under C-I and NC-I scenario, with drop-one-year cross-validation. The 95% envelope shows the 95% confidence interval constructed using model errors.**

**Table 2 Correlation coefficients (Corr.) and RPSS for predictions (drop-one-year cross-validated) at cluster level compared to observations under C-I and NC-I scenario.**

| Cluster | C1 | C2 | C3 | C4 | C5 | C6 | C7 | C8 | Non-cluster |
|---|---|---|---|---|---|---|---|---|---|
| **Corr.** | 0.137 | -0.027 | 0.171 | 0.184 | 0.514 | -0.157 | 0.353 | -0.108 | 0.297 |
| **RPSS(%)** | 22.88 | -26.14 | 33.32 | 12.74 | 10.02 | -43.61 | 20.92 | -26.40 | 13.25 |

At the grid scale, depending on the case (*direct* or *indirect)*, and for different clusters, correlations between predictions and observations can favor the *clustered* case or the *non-clustered* case (Fig. 8). In general, the *indirect* model provides a smoother pattern of correlations, with grid-cells showing a negative correlation in the *direct* case now improved to near or above zero (Fig. 8). For example, Cluster 5 under the *indirect* case illustrates a more consistent positive correlation within the cluster. Some parts of the region reach a correlation over 0.6, such as central-northwestern Ethiopia (Cluster 5), which is consistent with the region of high cluster-level prediction skill. The percentage of grid-cells with correlations passing the 95% significance test is the highest for the NC-D case (Table 3); however, some locations demonstrate the lowest skills among all four scenarios.

| | Non-Clustered | Clustered |
|---|---|---|
| Direct | | |
| Indirect | | |

[Figure]

[Figure]

**Figure 8: Pearson correlations between grid-level observations and predictions under four scenarios, with the clustering boundary delineated roughly in black.**

**Table 3: Grid-level Pearson correlation and RPSS statistics**

| Statistical Model | Grid-level correlations | | | Grid-level RPSS | | |
|---|---|---|---|---|---|---|
| | mean | stdev | significant corr % | mean (%) | stdev (%) | positive RPSS % |
| NC-D | 0.128 | 0.258 | 19.3% | -5.21 | 27.0 | 42.8% |
| NC-I | 0.063 | 0.186 | 3.13% | -2.26 | 14.6 | 43.9% |
| C-D | 0.055 | 0.230 | 10.6% | -14.0 | 31.0 | 33.9% |
| C-I | 0.081 | 0.206 | 12.4% | -9.60 | 29.4 | 44.4% |
| **Dynamical Model** | | | | | | |
| (1) | -0.105 | 0.209 | 0.51% | -31.4 | 25.4 | 5.70% |
| (2) | 0.133 | 0.171 | 6.26% | -14.2 | 24.6 | 27.0% |
| (3) | 0.086 | 0.130 | 2.08% | -14.9 | 25.2 | 26.2% |
| (4) | 0.027 | 0.156 | 0.38% | -14.4 | 19.3 | 22.6% |

| | | | | | | |
|---|---|---|---|---|---|---|
| (5) | 0.067 | 0.170 | 1.64% | -9.66 | 17.0 | 28.4% |
| (6) | 0.139 | 0.165 | 6.53% | -5.66 | 16.7 | 38.1% |
| (7) | 0.102 | 0.130 | 1.67% | -8.64 | 17.6 | 31.7% |
| (8) | 0.009 | 0.185 | 0.90% | -10.3 | 14.8 | 26.7% |
| (9) | 0.244 | 0.149 | 23.1% | -2.33 | 21.8 | 46.0% |
| (10) | 0.244 | 0.149 | 21.2% | -1.09 | 16.8 | 48.9% |

Similar findings are evident by evaluating the RPSS except for Cluster 8; instead of improving with increased RPSS in the *indirect* case, the grid-scale predictions deteriorate given poor cluster-level prediction (for the C-I case). However, the percentage of grid-cells with positive RPSS values overall for the C-I case is still the second highest after the NC-I case (Table 3),

5   indicating the  indirect case is superior in terms of the number of grid-cells with improved skill compared to using climatology, particularly for grid-cells associated with skillful intermediate cluster-level predictions. The predictions are most skillful for the same region of central-northwestern Ethiopia (Cluster 5; Fig. 9) with 87% of its grid-cells showing positive RPSS and a spatial average RPSS value of 15% under the C-I scenario (Table 4).

10   Table 4: Grid-level Pearson correlation and RPSS statistics for grid-cells *within Cluster 5*

| Statistical Model | Grid-level correlations | | | Grid-level RPSS | | |
|---|---|---|---|---|---|---|
| | mean | stdev | significant corr % | mean (%) | stdev (%) | positive RPSS % |
| NC-D | 0.378 | 0.211 | 60.7% | 19.1 | 22.9 | 80.3% |
| NC-I | 0.265 | 0.111 | 12.8% | 8.33 | 14.8 | 70.3% |
| C-D | 0.229 | 0.244 | 30.5% | 6.91 | 24.1 | 62.3% |
| C-I | 0.346 | 0.167 | 55.4% | 14.5 | 13.1 | 87.0% |
| **Dynamical Model** | | | | | | |
| (9) | 0.353 | 0.110 | 46.8% | 8.21 | 18.2 | 65.7% |
| (10) | 0.248 | 0.130 | 18.4% | 3.92 | 16.2 | 59.5% |

|  | **Non-Clustered** | **Clustered** |
|:---:|:---:|:---:|
| **Direct** | | |
| **Indirect** | | |

[Figure]

[Figure]

**Figure 9: grid-level RPSS (%) under four scenarios using climate variables as predictors, with the clustering boundary delineated roughly in black.**

**4.2 Dynamical model predictions**

The RPSS values based on the prediction ensembles of each dynamical model improve  remarkably after bias correction. The median RPSS values over all the grid-cells are now close to zero (Fig. 10) with two models, NASA-GMAO and NCEP-CFSv2, showing the highest RPSS value (-2.3% and -1.1%, respectively; Table 3). These two dynamical models also exhibit generally higher grid-level correlations over the study region (averaging 0.24 for both models; Table 3 and Fig. 11), as compared with other NMME models. The two best performing dynamical models after bias correction show advantage over statistical models, as assessed by correlation and RPSS metrics; however, all other dynamical models are inferior to the statistical models under NC-D and C-I scenarios, particularly given the percent of grid-cells with significant correlation and positive RPSS metrics (Table 3).

Within a certain cluster, statistical models may perform better than all dynamical models. For example, for Cluster 5, all statistical models show higher average RPSS values than that of all dynamical models (Table 4). The percentage of grid-cells with significant correlation reaches 61% for the statistical model under NC-D scenario, compared to the highest value of 47% among all the dynamical models. Similarly, the percentage with positive RPSS achieves 87% under C-I scenario as opposed to 66% for dynamical models. Note that the dynamical models also produce raw predictions in a lower spatial resolution (1°×1°) than the statistical models (0.1°×0.1°) and requires bias correction to guarantee comparable skills.

[Figure]

**Figure 10: Boxplots of grid-level RPSS (%) for 10 dynamical models from NMME (a) before and (b) after bias correction, labeled with the same number as listed in the context. Note: For each box plot, the line inside the box is the median, the box edges represent the 25th and 75th percentiles, and the whiskers extend to the most extreme data points not considered outliers (outliers not shown).**

[Figure]

**Figure 11: Pearson correlations between grid-level observations and ensemble mean of bias-corrected predictions for 10 dynamical models from NMME, labeled with the same number as listed in the context. Note that the scale ranges from -1 to 1.**

**5 Conclusions and discussion**

This study demonstrates the potential for season-ahead large-scale climate information to produce skillful and credible high-resolution precipitation predictions under a *clustered indirect* approach in western Ethiopia. At the regional scale, the approach shows promise for northwestern Ethiopia (Cluster 1, 3, 5, and 7), particularly compared to current NMA operational forecasts,

5 which are only moderately more skillful than climatology (Korecha and Sorteberg, 2013). The regional average RPSS in this study under the *clustered* case ranges from 2110% to 433% for northwestern Ethiopia, as opposed to values under 6% for NMA operational forecast (Korecha and Sorteberg, 2013). The approach adopted here also advances on previous studies (Gissila et al., 2004, Block and Rajagopalan, 2007, Korecha and Barnston, 2007, Diro et al., 2011b, Segele et al., 2015) by first applying an objective cluster analysis and then conditionally constructing high-resolution predictions. A unique set of predictors is applied to

10 each cluster, which contributes to superior prediction performance at cluster levels in northwestern Ethiopia, as compared with predictions from the *non-clustered* approach. Grid-level prediction under the *clustered indirect* case also reduces the effect of over-fitting relative to the *direct* case and improves negative RPSS values to near or above zeros; that said, the *non-clustered direct* case also illustrates higher correlation and RPSS values on average.

15 Two out of 10 NMME dynamical models, NASA-GMAO and NCEP-CFSv2, demonstrate overall superior performance to the statistical models; however, for certain regions such as Cluster 5, the performance of statistical models under *clustered indirect* and *non-clustered direct* cases is still superior. It is also worth noting that the statistical model predictions are at a one hundred times finer spatial resolution than the dynamical models providing additional advantages at the local scale, when skillful. Nevertheless, improvements in dynamical models continue and their application to seasonal precipitation prediction is likely to

20 grow (e.g. Palmer et al., 2004, Saha et al., 2006, Lim et al., 2009).

Relatively poor prediction performance is evident in some locations such as southwestern Ethiopia and regions outside Ethiopia, where the hydroclimatic processes that produce precipitation might be driven by local factors or other regional climate patterns rather than large-scale climate variables identified in this study. A previous study (Zhang et al., 2016) has shown that the influence

25 of ENSO on JJAS precipitation in western Ethiopia decreases generally from north to south, and is likely one of the reasons why skills are relatively low in southwestern Ethiopia. Cluster 5 was also identified with the strongest connection to equatorial Pacific SST (Zhang et al., 2016), which is consistent with the highest skill found in this study. Other regions with low prediction skill show relatively strong connections to SST in neighboring oceanic regions. However, connections with those climate patterns appear to be less robust than with ENSO, making the predictions in their associated regions less skillful. This is also consistent

30 with the findings from other studies that even though all three oceans (Indian, Atlantic, and Pacific Ocean) affect the JJAS precipitation in western Ethiopia, the Pacific Ocean still plays the greatest role (Segele et al., 2009, Omondi et al., 2013).

The southwest Asian monsoon over Indian Ocean may also be critical in determining the precipitation, given that the clusters with better prediction skills lie along the pathway of the monsoon. Based on the global concurrent correlation maps between JJAS

35 precipitation and SLP for each cluster, Cluster 5 and 7 – the two clusters with the best skills – are the only ones that are strongly and negatively corrected with SLP near Bombay, and meantime strongly and positively correlated with the SLP at the eastern equatorial Pacific Ocean. The former indicates that a strong southwest Asian monsoon is associated with higher JJAS precipitation amount, and vice versa. The latter indicates that a high surface pressure over the eastern equatorial Pacific Ocean often accompanied with cold SST and a raining pattern – a La Nina phenomenon – also brings higher JJAS precipitation to the western

40 Ethiopia, and vice versa. Cluster 2 – one of the worst predicted clusters – shows moderately strong negative correlation with SLP

near Bombay; however, it is also correlated strongly and negatively with SLP at southern Indian Ocean, indicating a possible weak gradient of the southwest Asian monsoon. Moreover, its correlation with SLP over equatorial Pacific Ocean is nonsignificant. Considering in general El Nino suppresses the monsoon and La Nina increases it (Kumar et al., 2006), strong correlations with both ENSO and the monsoon in the correct direction, such as for Cluster 5 and 7, indicate a double insurance over their association with the southwest Asian monsoon. Therefore, clusters which are more affected by the southwest Asian monsoon over Indian Ocean, particularly coupled with the influence of ENSO, are likely to show more promises in their prediction skills.

Relatively poor prediction performance is evident in some locations such as southwestern Ethiopia and regions outside Ethiopia, where the hydroclimatic processes that produce precipitation might be driven by local factors or other regional climate patterns rather than large-scale climate variables identified in this study. A previous study (Zhang et al., 2016) has shown that the influence of ENSO on JJAS precipitation in western Ethiopia decreases generally from north to south, and is likely one of the reasons why skills are relatively low in southwestern Ethiopia. Cluster 5 was also identified with the strongest connection to equatorial Pacific SST (Zhang et al., 2016), which is consistent with the highest skill found in this study. Other regions with low prediction skill show relatively strong connections to SST and pressure systems in neighboring oceanic regions. However, connections with those climate patterns appear to be less robust than with ENSO, making the predictions in their associated regions less skillful. To test the prospects for improving prediction performance by including season-ahead local variables, soil moisture and spring rains were investigated; however, no significant improvement was found; in fact, performance skill deteriorates when adding local predictors may simply serve to introduce more noise and encourage over-fitting.

Additional prediction features also warrant future attention, including longer prediction lead times and evaluation of other relevant characteristics (e.g. intra-seasonal dry spells, seasonal onset or cessation, etc.). As observational datasets continue to grow, data-driven cluster analysis and statistical modeling approaches may be expected to improve. The growing length of time series and climate change impacts also call for careful analysis on possible significant trends in the data. Improving predictive capabilities may not be a complete panacea, but it can continue to be an important part of a decisions-maker's portfolio as they cope with hydroclimatic variability and its inherent risks.

**6 Data availability**

The National Centers for Environmental Prediction and National Center for Atmospheric Research (NCEP/NCAR) reanalysis dataset can be accessed through the National Oceanic & Atmospheric Administration (NOAA) Earth System Research Laboratory (ESRL) website (https://www.esrl.noaa.gov/psd/data/reanalysis/).

The NMME hindcasts are available through the International Research Institute for Climate and Society (IRI) website (http://iridl.ldeo.columbia.edu/SOURCES/.Models/.NMME/).

The gridded precipitation dataset in western Ethiopia is available upon request from NMA (http://www.ethiomet.gov.et/).

**Competing interests**

The authors declare that they have no conflict of interest.

**Acknowledgements**

This study was supported by NASA Project NNX14AD30G and NSF PIRE Project 1545874. We acknowledge the National Meteorological Agency of Ethiopia for sharing data. We also want to thank the reviewers for their suggestions in improving this work.

---

## Author Response (AR3)

I had a look at the revised manuscript and I think, that the authors responded professionally to my comments. I only have a few suggestions, but in general I recommend to publish the manuscript in HESS.

Minor comments:

1) I appreciate that the authors give a general overview about the atmospheric mechanisms related with precipitation variability in the target region. For me the link with the southwest Asian monsoon is not trivial. Since it is an important point, as the autors argue in the conclusion, I think a bit more information in the introduction might be helpful.

We have moved the paragraph on atmospheric mechanisms to the introduction of the second section (Page 3 Line 14, *referring to the revised manuscript without track-changes*). We also edited the paragraph a little bit such that the contents flow well. The edited paragraph is pasted below:

"The climate mechanisms affecting JJAS precipitation patterns in western Ethiopia are quite complex. Sea surface temperatures (SST) in the equatorial Pacific Ocean representing the well-known El Nino-Southern Oscillation (ENSO) phenomena is considered a primary indicator of precipitation variability, with El Nino/La Nina often associated with deficit/excess of precipitation across the study region (e.g. NMSA, 1996, Camberlin, 1997, Bekele, 1997, Segele and Lamb, 2005, Diro et al., 2011, Elagib and Elhag, 2011). Additionally, there is evidence of direct moisture transport from the Gulf of Guinea (equatorial Atlantic Ocean), the Indian Ocean, and the Mediterranean Sea, affecting Ethiopia's summertime precipitation (Viste and Sorteberg, 2013a, Viste and Sorteberg, 2013b). These moisture fluxes are often related to pressure patterns across the continent. For instance, the St. Helena High over the southern Atlantic Ocean or a high pressure over the Gulf of Guinea, coupled with a simultaneous low pressure over the Indian Ocean or a monsoon trough over Arabic Peninsula, all bring intensified westerlies and south-westerlies that transport moist air across the Congo Basin to the western Ethiopian highlands (Segele et al., 2009, Williams et al., 2011). Similarly, the southwest Asian monsoon in the Indian Ocean, which has a strong positive relationship with concurrent JJAS precipitation in western Ethiopia, is associated with the Mascarene High over the southern Indian Ocean and a low pressure system near Bombay. During this monsoon season, the southeast trade winds in the southern hemisphere are channeled by the east African highlands while crossing the equator and become a southwest monsoon flow. They are further diverted by the Turkana Channel, enhancing convergence with the westerlies/south-westerlies above the western Ethiopian highlands and bringing moisture to the region (Kinuthia, 1992, Nicholson, 1996, Camberlin, 1997, Slingo et al., 2005, Segele et al., 2009, Nicholson, 2014). In addition, the effect of other hydro-climate variables, such as Indian Ocean SST, local and regional atmospheric pressure systems (e.g. Azores High) also have notable influence on Ethiopia's precipitation variability (e.g. Kassahun, 1987, Tadesse, 1994, NMSA, 1996, Shanko and Camberlin, 1998, Goddard and Graham, 1999, Latif et al., 1999, Black et al., 2003, Segele and Lamb, 2005). Consequently, these large-scale climate variables may serve as potential predictors in statistical seasonal precipitation prediction models."

2) The link with the Monsoon is investigated by means of the correlation of seasonal precipitation amounts with SLP near Bombay. Is this correlation simultanous or does it consider a lead time? If a lead time is considered, is the monsoonal circulation robust, i.e. stable during subsequent months?

This correlation is simultaneous, or concurrent, as we mentioned in the manuscript (Page 16 Line 35). No lead time is considered for this analysis.

3) structure of the introduction and methods section: I would give the climatic overview (atmospheric mechanisms) in the introduction.

Thank you. We agree with the reviewer and have moved the climatic overview in the introduction of Section 2. Please refer to the response to comment 1.

In the methods section it would be appropriate, to describe the PCA techniques for the predictor generation first and the calibration of the LRM afterwards.

Thank you. We agree with the reviewer and have moved the paragraphs explaining PCA and PCR forward. We also edited some text throughout the section to make sure it flows well (Page 6 Line 17). The edited paragraphs are pasted below:

"Season-ahead (March-May) or month-ahead (May) large-scale climate variables that are physically relevant in potentially modulating moisture transport to the basin (or cluster) are selected as potential predictors. Four climate variables are selected here for further evaluation based on outcomes of the aforementioned prediction studies: SST, SLP, geopotential height (GH) at 500mb, and surface air temperature (SAT). All climate variables are from the National Centers for Environmental Prediction and National Center for Atmospheric Research (NCEP/NCAR) reanalysis dataset (Kalnay et al., 1996) at a 2.5˚×2.5˚ grid scale.

Those potential predictors are first transformed through principal component analysis (PCA; Jolliffe, 2002). PCA is a common approach in climate modeling to reduce the dimensionality of predictors and remove multi-collinearity, while simultaneously extracting the most dominant signals from the potential predictors, typically reflected in the first few PCs. Since PCA is independent of the predictand, retaining the first few PCs as predictors, in lieu of the original variables, also helps to reduce artificial prediction skill.

Subsequently, a certain number of PCs are used as the direct inputs into a MLR model, otherwise known as the principal component regression (PCR). PCR is performed in a "drop-one-year" cross-validation mode to reduce over-fitting effects and therefore avoid overestimation of prediction skill. This requires reconstructing the principal components for the dropped year, and then multiplying the coefficient estimates with each reconstructed PC respectively in order to obtain the final predicted value for the dropped year (e.g. Block and Rajagopalan, 2009, Wilks, 2011). A detailed methodology is provided below.

To avoid overfitting, the entire process including predictor selection and statistical modeling is processed using cross-validation. To start, drop-one-year precipitation observations for JJAS averaged across the region and each cluster are spatially correlated independently with each global climate variable. As a result, there are total of 1044 global correlation maps given the 29-year time-series, eight clusters plus one non-cluster, and four climate variables. Hence, a program to automatically select highly correlated

and justifiable regions as predictors is developed. The following steps describe the subsequent statistical modeling process (Fig. 3):

(1) Grid-cells within each justifiable region (e.g. equatorial Pacific; Fig. 4) with correlation above the 99% significance level are identified (Fig. 5). For regions containing grid-cells with both positive and negative correlations, the number of the identified grid-cells in each sign is counted. If a greater number of grid-cells is associated with significant positive correlation, for example, only grid-cells with positive correlations are kept for the following steps, and vice versa.

(2) The top 10% of the identified grid-cells with the highest correlation in each region is then selected, in order to boost the potential model skill.

(3) For each region, data of the selected grid-cells within the region are spatially averaged (defined as "pre-predictors").

(4) Pre-predictors are standardized, combined, and transformed through PCA for each cluster or non-cluster, and each dropped-year analysis separately.

(5a) The top principal components (PCs) from the PCA with a total of 95% variance explained are used as predictors in PCR. For the *direct* case, PCR is used to directly predict the grid-level precipitation; for the *indirect* case, PCR is used to predict the intermediate cluster-level precipitation.

(5b) For the *indirect* case only, cluster-level predictions are regressed to the grid-level. Note that the downscaling of cluster-level predictions to grid-level predictions is also cross-validated to avoid overfitting."

4) I still assume that that trends in cluster 5 and 7 might lead to better forecast results. The PCAs are usualkly orthogonal, which does not mean that the predictors do not have any trends, Particularly for sst I assume strong trends during recent decades. I understand the point, that 29 years are to short for detrending (since decadal variations such as PDO are not captured otherwise), but I think this should be better communicated in the conclusions.

Thank you. We agree that PCs can still have trends; however, as the PCA process is completely independent of the predictand and the model is constructed under strict cross-validation from predictor selection to regression, we do believe the skillful results from Cluster 5 and 7 are more likely due to the climate mechanisms affecting them than the trends.

Regarding the comment on data length for trend analysis, a more comprehensive description on how possible trends in the time series could lead to better skill is provided in the conclusion (Page 17 Line 12). It is also pasted below:

"As observational datasets continue to grow, data-driven cluster analyses and statistical modeling approaches may be expected to improve. Careful analysis of possible significant trends in the data is also warranted; a region with a relatively high correlation may be selected solely based on trends in predictors and observations. For shorter time series, such as the data used in this study, trend analysis may not be reliable; detrending can also reduce evidence of large-scale decadal climate signals."

Technical corrections: (pages and lines are taken from the track change version)

p22, 1st sentence: I do not understand that sentence

The original sentence was "This study demonstrates the potential for season-ahead large-scale climate information to produce skillful and credible high-resolution precipitation predictions under a *clustered indirect* approach in western Ethiopia.", and we rephrased it to (Page 16 Line 2):

"This study demonstrates the potential for applying season-ahead large-scale climate information to predict high-resolution precipitation using a statistical modeling approach. Skillful and credible predictions are produced for some regions in western Ethiopia, particularly under a *clustered indirect* statistical approach."

p22, l. 36: corrects? Do you mean correlates?

Thank you. We have changed the "corrected" to "correlated" (Page 16 Line 37).

p23,l2: what do you mean by gradient?

By "gradient" we mean the difference in sea level pressure between a high pressure system in the southern Indian Ocean and a low pressure system near Bombay. If the gradient is weak, the southwest Asian monsoon may also be weak. We realize our original expression may be confusing; therefore, we have rephrased the sentence to (Page 17 Line 1):

"Cluster 2 – one of the worst predicted clusters – shows moderately strong negative correlation with SLP near Bombay; however, it is also correlated strongly and negatively with SLP in the southern Indian Ocean (a high pressure system that drives the monsoon toward the low pressure system near Bombay), indicating that high JJAS precipitation in Cluster 2 is not necessarily associated with a strong southwest Asian monsoon."

[Figure]

**Figure 1: Spatial and temporal variability of June-September seasonal total precipitation in western Ethiopia: (a) spatial pattern of temporal-average, and (b) spatial-average time series.**

The climate mechanisms affecting JJAS precipitation patterns in western Ethiopia are quite complex. Sea surface

15    temperatures (SST) in the equatorial Pacific Ocean representing the well-known  El Nino-Southern Oscillation (ENSO) phenomena is considered a primary indicator of precipitation variability,  with El Nino/La Nina often associated with deficit/excess of precipitation across the study region (e.g. NMSA, 1996, Camberlin,

20    1997, Bekele, 1997, Segele and Lamb, 2005, Diro et al., 2011a, Elagib and Elhag, 2011). Additionally, there is evidence  direct moisture transport from the Gulf of Guinea (equatorial Atlantic Ocean), the Indian Ocean, and the Mediterranean Sea affecting Ethiopia's summer precipitation (Viste and Sorteberg, 2013a, Viste and Sorteberg, 2013b). The moisture fluxes are often related to pressure patterns across the continent. For instance, the St. Helena High over the southern Atlantic Ocean or a high pressure over  Gulf of Guinea coupled with a simultaneous low pressure over the Indian

25    Ocean or a monsoon trough over Arabic Peninsula, all bring intensified westerlies and south-westerlies that transport moist air across the Congo Basin to the western Ethiopian highlands  (Segele et al., 2009, Williams et al., 2011). Similarly, the southwest Asian monsoon in the Indian Ocean, which has a strong positive relationship with  concurrent JJAS

precipitationrainfall in the western Ethiopia, is associated with the Mascarene hHigh over the southern Indian Ocean and a low pressure system near Bombay. During this monsoon season, the southeast trade winds in the southern hemisphere are channeled by the east African highlands while crossing the equator and become a southwest monsoon flow. It isThey are further diverted by the Turkana Channel, enhancing convergence with the westerlies/south-westerlies above the western Ethiopian highlands and bringing moisture to thisthe region (Kinuthia, 1992, Nicholson, 1996, Camberlin, 1997, Slingo et al., 2005, Segele et al., 2009, Nicholson, 2014). In addition, the effect of other hydro-climate variables relevant to the aforementioned driven factors, such as the Indian Ocean SST, local and other regional atmospheric pressure systems such as (e.g. Azores High) also have notable influence on Ethiopia's precipitation variability (e.g. Kassahun, 1987, Tadesse, 1994, NMSA, 1996, Shanko and Camberlin, 1998, Goddard and Graham, 1999, Latif et al., 1999, Black et al., 2003, Segele and Lamb, 2005). Consequently, these Large-scale climate variables may serve as potential predictors in statistical seasonal precipitation prediction models.

[revised manuscript text omitted]

10   grid-cells across 1983–2011 (29 years). This product has been verified against station data and has been deemed representative of observed precipitation in western Ethiopia (Dinku et al., 2014). Given the high-resolution gridded dataset, k-means clustering is performed for a range of predefined numbers of clusters; the optimal number of clusters is identified by various evaluation metrics based on the within-cluster sum of square errors (WSS), including elbow method with *difference in WSS*, gap statistic with *difference in difference*, and qualitative analysis on post-visualization of clusters.. During the clustering process, each grid-cell is

15   assigned and reassigned to clusters until the WSS is minimized. This does not require any subjective delineation or manual delineation of boundaries between clustered stations or grid-cells; instead, an automated and objective delineation is performed. The mean time series of each cluster illustrates high intra-correlation within the cluster and low inter-correlation between any two clusters, indicating strong coherency of the clustering results. For a detailed analysis including a complete correlation table and unique patterns for each cluster-level time series associated with large climate variables, readers are referred to Zhang et al. (2016).

[Figure]

**Figure 2: Regionalization map of 8 homogeneous regions marked by different colors, with country boundary and river profile. After Zhang et al. (2016)**

**3.2 Statistical modeling approach**

Many studies have investigated statistical models for seasonal climate prediction. These studies vary by pre-classification of predictor or predictand regime, predictor selection process, and statistical methods. For example, Hertig and Jacobeit (2011) investigate sea surface temperature (SST) regimes as potential predictors for subsequent precipitation and temperature in the Mediterranean region. Through techniques including multiple applications of PCA, 17 stationary SST regimes were identified. Gerlitz et al. (2016) apply a k-means cluster analysis to grid-cells identified with significant correlations in the predictor field in order to facilitate predictor selection. Suárez-Moreno and Rodríguez-Fonseca (2015) investigate stationarity based on a long time series using a 21-year moving correlation window. The statistical prediction models are then applied to each stationary period respectively and the entire period for comparison. Despite diverse methods in seasonal prediction, multiple linear regression (MLR) is favored by many as a statistical modeling approach given its well-developed theory, simple model structure, efficient processing, and often skillful outcomes (e.g. Omondi et al., 2013, Camberlin and Philippon, 2002, Diro et al., 2008). As mentioned, only a few studies have focused on seasonal precipitation prediction in Ethiopia (Gissila et al., 2004, Block and Rajagopalan, 2007, Korecha and Barnston, 2007, Diro et al., 2008, Diro et al., 2011b, Segele et al., 2015), and almost all of them include the applications of MLR. This study also applies MLR to predict seasonal precipitation, yet differentiates from other studies by applying predictions to pre-defined homogeneous regions and further translating to local-level predictions.

~~Large-scale climate variables are often evaluated as potential predictors in statistical seasonal precipitation prediction models, commonly including sea surface temperatures (SST) in the equatorial Pacific Ocean representing the well-known of the El Niño-Southern Oscillation (ENSO) (Stone et al., 1996). Sea level pressure (SLP) in the eastern Pacific Ocean at Tahiti as an critical and stable component for measuring an ENSO index (Torrence and Webster, 1999) warrants another potential predictor. For Ethiopia, the ENSO phenomenon is considered a primary indicator of precipitation variability, particularly in the main JJAS rainy season with El Niño/La Niña often associated with deficit/excess of precipitation amount in the study region. Evidences have also shown a more direct moisture transport from the Gulf of Guinea (equatorial Atlantic Ocean), the Indian Ocean, and the Mediterranean Sea affecting Ethiopia's summer precipitation (Viste and Sorteberg, 2013a, Viste and Sorteberg, 2013b). Those moisture fluxes are often related to pressure patterns across the continent. For instance, the St. Helena high over the southern Atlantic Ocean or a high pressure over Gulf of Guinea, and a simultaneous low pressure over Indian Ocean or a monsoon trough over Arabic Peninsula bring intensified westerlies and south-westerlies that transport moist air across the Congo Basin to the western Ethiopian highlands in the summer. Similarly, the southwest Asian monsoon at the Indian Ocean, which has a strong positive relationship with the concurrent JJAS rainfall in the western Ethiopia, is associated with the Mascarene high over the southern Indian Ocean and a low pressure system near Bombay. During this monsoon season, the southeast trades in the southern hemisphere are channeled by the east African highlands while crossing the equator and become a southwest monsoon flow. It is further diverted by the Turkana Channel, enhancing convergence with the westerlies/south-westerlies above the western Ethiopian highlands and bringing moisture to this region. In addition, the effect of other climate variables relevant to the aforementioned driven factors, such as the Indian Ocean SST, local and other regional atmospheric pressure systems such as Azores High also have notable influence on Ethiopia's precipitation variability (e.g. Kassahun, 1987, Tadesse, 1994, NMSA, 1996, Shanko and Camberlin, 1998, Goddard and Graham, 1999, Latif et al., 1999, Black et al., 2003, Segele and Lamb, 2005).~~

Consequently, season Season-ahead (March-May) or month-ahead (May) large-scale climate variables that are physically relevant in potentially modulating moisture transport to the basin (or cluster) are selected as potential predictors. Four climate variables are selected here for further evaluation based on outcomes of the aforementioned prediction studies: SST, SLP, geopotential height (GH) at 500mb, and surface air temperature (SAT). All climate variables are from the National Centers for Environmental Prediction and National Center for Atmospheric Research (NCEP/NCAR) reanalysis dataset (Kalnay et al., 1996) at a 2.5˚×2.5˚ grid scale.

Those potential predictors are first transformed through principal component analysis (PCA; Jolliffe, 2002). PCA is a common approach in climate modeling to reduce the dimensionality of predictors and remove multi-collinearity, while simultaneously extracting the most dominant signals from the potential predictors, typically reflected in the first few PCs. Since PCA is independent of the predictand, retaining the first few PCs as predictors, in lieu of the original variables, also helps to reduce artificial prediction skill.

Subsequently, a certain number of PCs are used as the direct inputs into a MLR model, otherwise known as the principal component regression (PCR). PCR is performed in a "drop-one-year" cross-validation mode to reduce over-fitting effects and therefore avoid overestimation of prediction skill. This requires reconstructing the principal components for the dropped year, and then multiplying the coefficient estimates with each reconstructed PC respectively in order to obtain the final predicted value for the dropped year (e.g. Block and Rajagopalan, 2009, Wilks, 2011). A detailed methodology is provided below.

To avoid overfitting, the entire process including predictor selection and statistical modeling is processed using cross-validation. To start, drop-one-year precipitation observations for JJAS averaged across the region and each cluster are spatially correlated independently with each global climate variable. As a result, there are total of 1044 global correlation maps given the 29-year time-series, eight clusters plus one non-cluster, and four climate variables. Hence, a program to automatically select highly correlated and justifiable regions as predictors is developed. The following steps describe the subsequent statistical modeling process (Fig. 3):

(1) Grid-cells within each justifiable region (e.g. equatorial Pacific; Fig. 4) with correlation above the 99% significance level are identified (Fig. 5). For regions containing grid-cells with both positive and negative correlations, the number of the identified grid-cells in each sign is counted. If a greater number of grid-cells is associated with significant positive correlation, for example, only grid-cells with positive correlations are kept for the following steps, and vice versa.

(2) The top 10% of the identified grid-cells with the highest correlation in each region is then selected, in order to boost the potential model skill.

(3) For each region, data of the selected grid-cells within the region are spatially averaged (defined as "pre-predictors").

(4) Pre-predictors are standardized, combined, and transformed through PCA principal component analysis (PCA; Jolliffe, 2002) for each cluster or non-cluster, and each dropped-year analysis separately.

(5a) The top principal components (PCs) from the PCA with a total of 95% variance explained are used as predictors – the direct inputs into the MLR model, otherwise known as the principal component regression (in PCR). For the *direct* case, PCR is used to directly predict the grid-level precipitation; for the *indirect* case, PCR is used to predict the intermediate cluster-level precipitation.

(5b) For the *indirect* case only, cluster-level predictions are regressed to the grid-level. Note that the downscaling of cluster-level predictions to grid-level predictions is also cross-validated to avoid overfitting.

[Figure]

**Figure 3: Flow chat of data processing for predictors into the statistical model. Numbers framed by dash lines correspond to the procedures listed in the context. Note: pre. – precipitation, t-s – time-series, avg. – average.**

[Figure]

**Figure 4: Justifiable climate regions globally for selecting predictors: (a) For SLP and GH at 500 mb with regions including EP, ES, LO, AH, SH, MH, and AM. For SAT, only LO is included. (b) For SST with regions including EP, NI, SI, and AT. Note: EP - equatorial Pacific region, ES – Tahiti island for ENSO measurement, LO - local region, AH - Azores High, SH - St Helena High, MH - Mascarene High, AM - SW Asian Monsoon, NI - North Indian Ocean, SI - South Indian Ocean, AT - Equatorial/South Atlantic Ocean.**

[Figure]

**Figure 5: Correlation map between mean JJAS seasonal precipitation time series in Cluster 5 and global SST under cross-validation, with correlations lower than 99% significance level masked out (one-tail test).**

5  ~~PCA is a common approach in climate modeling to reduce the dimensionality of predictors and remove multi-collinearity, while simultaneously extracting the most dominant signals from the potential predictors, typically reflected in the first few PCs. Since PCA is independent of the predictand, retaining the first few PCs as predictors, in lieu of the original variables, also helps to reduce artificial prediction skill.~~

10 ~~PCR is performed in a "drop-one-year" cross-validation mode to reduce over-fitting effects and therefore avoid overestimation of prediction skill. This requires reconstructing the principal components for the dropped year, and then multiplying the coefficient estimates with each reconstructed PC respectively in order to obtain the final predicted value for the dropped year (e.g. Block and Rajagopalan, 2009, Wilks, 2011). A 95% confidence interval of the cross-validated predictions is also constructed conditioned on model errors. Q-Q plots are evaluated to verify normally distributed residuals (results not included).~~

For the four scenarios, the model structures are quite similar but have subtle differences which could lead to evidently different outcomes (Table 1). Under the NC-D (Eq. (1a, b)) and C-D scenarios (Eq. (2a, b)), the time-series of JJAS seasonal total precipitation in each grid-cell (i.e. at local level) is used as the direct predictand ($Y_{i,t}$); however, the NC-D and C-D scenarios differ, as the former uses the same predictors ($X_t$) across all the grid-cells, while the latter uses different predictors according to the cluster

20  to which the grid-cell is assigned ($X_{j,t}$). In the indirect case, the cluster-level time-series of JJAS seasonal total precipitation (the time-series averaged over all grid-cells that belong to a given cluster, $Y_{m,t}$ or $Y_{j,t}$) is first predicted (Eq. (3a, b) and (4a, b)). The predicted intermediate product ($\widetilde{Y}_{m,t}$ or $\widetilde{Y}_{j,t}$) is then used as the only regressor in the second step to estimate the grid-level precipitation ($\widetilde{Y}_{i,t}$ or $\widetilde{Y}_{i\in j,t}$ for every j; Eq. (3c, d) and (4c, d)). Again, for the C-I scenario, predictors in the first step are unique for each of the eight clusters and grid-cells within that cluster ($X_{j,t}$), while predictors are identical for all grid-cells ($X_t$) under the NC-

I scenario. A 95% confidence interval of the cross-validated predictions is also constructed conditioned on model errors. Q-Q plots are evaluated to verify normally distributed residuals (results not included).

Table 1: Equations of linear regression panel models under four scenarios

| | Non-clustered | | Clustered | |
|---|---|---|---|---|
| **Direct** | $Y_{i,t} = \widetilde{\alpha}_i + \widetilde{\beta}_i X_t + \varepsilon_{i,t}$ | ...... (1a) | $Y_{i\in j,t} = \widetilde{\alpha}_i + \widetilde{\beta}_i X_{j,t} + \varepsilon_{i,t}$ | ...... (2a) |
| | $\widetilde{Y}_{i,t} = \widetilde{\alpha}_i + \widetilde{\beta}_i X_t$ | ...... (1b) | $\widetilde{Y}_{i\in j,t} = \widetilde{\alpha}_i + \widetilde{\beta}_i X_{j,t}$ | ...... (2b) |
| **Indirect** | $Y_{m,t} = \widetilde{\alpha} + \widetilde{\beta} X_t + \varepsilon_t$ | ...... (3a) | $Y_{j,t} = \widetilde{\alpha}_j + \widetilde{\beta}_j X_{j,t} + \varepsilon_{j,t}$ | ...... (4a) |
| | $\widetilde{Y}_{m,t} = \widetilde{\alpha} + \widetilde{\beta} X_t$ | ...... (3b) | $\widetilde{Y}_{j,t} = \widetilde{\alpha}_j + \widetilde{\beta}_j X_{j,t}$ | ...... (4b) |
| | $Y_{i,t} = \widetilde{\eta}_i + \widetilde{\gamma}_i \widetilde{Y}_{m,t} + \nu_{i,t}$ | ...... (3c) | $Y_{i\in j,t} = \widetilde{\eta}_i + \widetilde{\gamma}_i \widetilde{Y}_{j,t} + \nu_{i,t}$ | ...... (4c) |
| | $\widetilde{Y}_{i,t} = \widetilde{\eta}_i + \widetilde{\gamma}_i \widetilde{Y}_{m,t}$ | ...... (3d) | $\widetilde{Y}_{i\in j,t} = \widetilde{\eta}_i + \widetilde{\gamma}_i \widetilde{Y}_{j,t}$ | ...... (4d) |

where Y- predictand of JJAS seasonal total precipitation; X- two predictors of top two PCs;
$\varepsilon, \nu$ - error terms; $\widetilde{Y}$ - predicted values of JJAS seasonal total precipitation; $\widetilde{\alpha}, \widetilde{\beta}, \widetilde{\eta}, \widetilde{\gamma}$- estimated coefficients; i- grid-cell index; t- time (year) index; j- cluster index; i ∈ j- grid-cell i that belongs to clusterj; m- mean over entire study region that is equivalently the only one cluster.

**3.3 Dynamical modeling approach**

The North American Multi-Model Ensemble (NMME; Kirtman et al., 2014) is an experimental multi-model system consisting of coupled dynamical models from various modeling centers in North America. To our knowledge, it is also the most extensive multi-model seasonal prediction archive. The NMME provides gridded climate predictions that cover regions globally and with different lead times. The hindcasts of monthly mean precipitations are easily accessible through the International Research Institute for Climate and Society (IRI) website (http://iridl.ldeo.columbia.edu/SOURCES/.Models/.NMME/), and can be easily aggregated to seasonal totals for comparison with the statistical model results in this study. Therefore, NMME JJAS seasonal precipitation predictions (1˚×1˚ grid-cells) are extracted from model ensembles that cover the same time period (1983–2011), geographic region (western Ethiopia), and with the same lead time (predictions made on June 1). A subset of 10 NMME models meet these criteria and are retained for further evaluation: (1) COLA-RSMAS-CCSM3, (2) COLA-RSMAS-CCSM4, (3) GFDL-CM2p1, (4) GFDL-CM2p1-are04, (5) GFDL-CM2p5-FLOR-A06, (6) GFDL-CM2p5-FLOR-B01, (7) IRI-ECHAM-AnomalyCoupled, (8) IRI-ECHAM-DirectCoupled, (9) NASA-GMAO, (10) NCEP-CFSv2. The names are kept the same as on the International Research Institute for Climate and Society (IRI) data repository website.

The NMME predictions for each of the 10 models are bias-corrected by applying probability mapping (e.g. Block et al., 2009, Teutschbein and Seibert, 2012, Chen et al., 2013) under cross-validation, subject to the observational dataset from NMA (Fig. 6). This is performed on a grid-cell by grid-cell basis on standardized data (the NMME dataset is reshaped to 0.1°×0.1° grid-cells to match the observational NMA dataset grid-cell size). The basic steps include:

(1) Fit gamma distributions to drop-one-year time-series from each observed and NMME grid-cell; for NMME this is performed on an individual model basis using all ensemble members available. (Goodness-of-fit tests indicate gamma distributions are appropriate; results not shown.)

(2) Translate gamma distributions into cumulative distribution functions (CDF).

(3) For any given dynamical model prediction at the grid-cell level, a corrected prediction value is attained by mapping from the modeled CDF to the observed CDF and applying the inverse gamma distribution. This is repeated for all grid-cells, all NMME models, and all dropped years.

5  After correction, the gamma CDF of predictions and observations approximately match (Fig. 6a). Additionally, each ensemble still retains its variability over time, though the overall ensemble mean is shifted to closely match observation (Fig. 6b).

[Figure]

**Figure 6: (a) bias correction of NMME predictions using probability mapping; (b) precipitation time-series from NMME (colored lines)**
10  **before and after correction, compared to observations (black line). Examples are shown for randomly selected six grid-cells.**

**3.4 Performance metrics**

Pearson correlations are used to measure the standardized covariance between observations and predictions. Ranked probability skill scores (RPSS; Wilks, 2011) are also evaluated to determine categorical skill based on probabilistic predictions. Here, the data
15  are split into three equal terciles representing below-normal, near-normal, and above-normal conditions. A perfect prediction yields an RPSS of 100%, and a prediction with less skill than climatology (long-term averages) yields an RPSS of less than zero. Median RPSS values from all 29 years are reported.

**4 Results**

**4.1 Statistical model predictions**

20  Correlations between cluster-level model predictions and observations range from -0.16 to 0.51, with Cluster 5 having the highest correlation and Cluster 6 the lowest (Table 2). In approximately 1/5 of the 29 years, the observation falls outside the prediction envelope (Fig. 7), indicating model overfitting and an inability of the predictors to capture precipitation variability. For RPSS, 3 out of 8 clusters indicate superior prediction skill over climatology (Table 2). Improvement in terms of RPSS over the non-cluster

scenario is evident for Cluster 1, 3 and 7. Although Cluster 5, in agriculturally rich central-northwestern Ethiopia (Fig. 2), shows a slightly deteriorated RPSS relative to non-cluster scenario, it still performs outstandingly with the highest correlation and a positive RPSS value of 0.51 and 10%, respectively. Cluster 2, 4, 6 and 8 show deteriorated RPSS compared to non-cluster scenario, although those clusters are mainly regions outside Ethiopia and southern Ethiopia (Fig. 2) where water resources and agricultural activities are considerably less (Fig. 1).

[Figure]

**Figure 7: cluster-level predictions and observations under C-I and NC-I scenario, with drop-one-year cross-validation. The 95% envelope shows the 95% confidence interval constructed using model errors.**

10 **Table 2 Correlation coefficients (Corr.) and RPSS for predictions (drop-one-year cross-validated) at cluster level compared to observations under C-I and NC-I scenario.**

| Cluster | C1 | C2 | C3 | C4 | C5 | C6 | C7 | C8 | Non-cluster |
|---|---|---|---|---|---|---|---|---|---|
| **Corr.** | 0.137 | -0.027 | 0.171 | 0.184 | 0.514 | -0.157 | 0.353 | -0.108 | 0.297 |
| **RPSS(%)** | 22.88 | -26.14 | 33.32 | 12.74 | 10.02 | -43.61 | 20.92 | -26.40 | 13.25 |

At the grid scale, depending on the case (*direct* or *indirect)*, and for different clusters, correlations between predictions and observations can favor the *clustered* case or the *non-clustered* case (Fig. 8). In general, the *indirect* model provides a smoother pattern of correlations, with grid-cells showing a negative correlation in the *direct* case now improved to near or above zero (Fig. 8). For example, Cluster 5 under the *indirect* case illustrates a more consistent positive correlation within the cluster. Some parts

5 of the region reach a correlation over 0.6, such as central-northwestern Ethiopia (Cluster 5), which is consistent with the region of high cluster-level prediction skill. The percentage of grid-cells with correlations passing the 95% significance test is the highest for the NC-D case (Table 3); however, some locations demonstrate the lowest skills among all four scenarios.

[Figure]

10 **Figure 8: Pearson correlations between grid-level observations and predictions under four scenarios, with the clustering boundary delineated roughly in black.**

**Table 3: Grid-level Pearson correlation and RPSS statistics**

| Statistical Model | Grid-level correlations | | | Grid-level RPSS | | |
|---|---|---|---|---|---|---|
| | mean | stdev | significant corr % | mean (%) | stdev (%) | positive RPSS % |
| NC-D | 0.128 | 0.258 | 19.3% | -5.21 | 27.0 | 42.8% |
| NC-I | 0.063 | 0.186 | 3.13% | -2.26 | 14.6 | 43.9% |
| C-D | 0.055 | 0.230 | 10.6% | -14.0 | 31.0 | 33.9% |
| C-I | 0.080 | 0.205 | 12.3% | -9.93 | 29.3 | 43.7% |
| **Dynamical Model** | | | | | | |
| (1) | -0.105 | 0.209 | 0.51% | -31.4 | 25.4 | 5.70% |
| (2) | 0.133 | 0.171 | 6.26% | -14.2 | 24.6 | 27.0% |
| (3) | 0.086 | 0.130 | 2.08% | -14.9 | 25.2 | 26.2% |
| (4) | 0.027 | 0.156 | 0.38% | -14.4 | 19.3 | 22.6% |
| (5) | 0.067 | 0.170 | 1.64% | -9.66 | 17.0 | 28.4% |
| (6) | 0.139 | 0.165 | 6.53% | -5.66 | 16.7 | 38.1% |
| (7) | 0.102 | 0.130 | 1.67% | -8.64 | 17.6 | 31.7% |
| (8) | 0.009 | 0.185 | 0.90% | -10.3 | 14.8 | 26.7% |
| (9) | 0.244 | 0.149 | 23.1% | -2.33 | 21.8 | 46.0% |
| (10) | 0.244 | 0.149 | 21.2% | -1.09 | 16.8 | 48.9% |

Similar findings are evident by evaluating the RPSS except for Cluster 8; instead of improving with increased RPSS in the *indirect* case, the grid-scale predictions deteriorate given poor cluster-level prediction (for the C-I case). However, the percentage of grid-cells with positive RPSS values overall for the C-I case is still the second highest after the NC-I case(Table 3), indicating the *indirect* cases are superior in terms of the number of grid-cells with improved skill compared to using climatology, particularly for grid-cells associated with skillful intermediate cluster-level predictions. The predictions are most skillful for the same region of central-northwestern Ethiopia (Cluster 5; Fig. 9) with 87% of its grid-cells showing positive RPSS and a spatial average RPSS value of 15% under the C-I scenario (Table 4).

**Table 4: Grid-level Pearson correlation and RPSS statistics for grid-cells *within Cluster 5***

| Statistical Model | Grid-level correlations | | | Grid-level RPSS | | |
|---|---|---|---|---|---|---|
| | mean | stdev | significant corr % | mean (%) | stdev (%) | positive RPSS % |
| NC-D | 0.378 | 0.211 | 60.7% | 19.1 | 22.9 | 80.3% |
| NC-I | 0.265 | 0.111 | 12.8% | 8.33 | 14.8 | 70.3% |
| C-D | 0.229 | 0.244 | 30.5% | 6.91 | 24.1 | 62.3% |
| C-I | 0.346 | 0.167 | 55.4% | 14.5 | 13.1 | 87.0% |
| **Dynamical Model** | | | | | | |
| (9) | 0.353 | 0.110 | 46.8% | 8.21 | 18.2 | 65.7% |
| (10) | 0.248 | 0.130 | 18.4% | 3.92 | 16.2 | 59.5% |

[Figure]

**Figure 9: grid-level RPSS (%) under four scenarios using climate variables as predictors, with the clustering boundary delineated roughly in black.**

**4.2 Dynamical model predictions**

The RPSS values based on the prediction ensembles of each dynamical model improve remarkably after bias correction. The median RPSS values over all the grid-cells are now close to zero (Fig. 10) with two models, NASA-GMAO and NCEP-CFSv2, showing the highest RPSS value (-2.3% and -1.1%, respectively; Table 3). These two dynamical models also exhibit generally higher grid-level correlations over the study region (averaging 0.24 for both models; Table 3 and Fig. 11), as compared with other NMME models. The two best performing dynamical models after bias correction show advantage over statistical models, as assessed by correlation and RPSS metrics; however, all other dynamical models are inferior to the statistical models under NC-D and C-I scenarios, particularly given the percent of grid-cells with significant correlation and positive RPSS metrics (Table 3).

Within a certain cluster, statistical models may perform better than all dynamical models. For example, for Cluster 5, all statistical models show higher average RPSS values than that of all dynamical models (Table 4). The percentage of grid-cells with significant correlation reaches 61% for the statistical model under NC-D scenario, compared to the highest value of 47% among all the dynamical models. Similarly, the percentage with positive RPSS achieves 87% under C-I scenario as opposed to 66% for dynamical models. Note that the dynamical models also produce raw predictions in a lower spatial resolution (1°×1°) than the statistical models (0.1°×0.1°) and requires bias correction to guarantee comparable skills.

(a)

[Figure]

(b)

**Figure 10: Boxplots of grid-level RPSS (%) for 10 dynamical models from NMME (a) before and (b) after bias correction, labeled with the same number as listed in the context. Note: For each box plot, the line inside the box is the median, the box edges represent the 25th and 75th percentiles, and the whiskers extend to the most extreme data points not considered outliers (outliers not shown).**

[Figure]

**Figure 11: Pearson correlations between grid-level observations and ensemble mean of bias-corrected predictions for 10 dynamical models from NMME, labeled with the same number as listed in the context. Note that the scale ranges from -1 to 1.**

**5 Conclusions and discussion**

This study demonstrates the potential for applying season-ahead large-scale climate information to predict high-resolution precipitation using a statistical modeling approach. Skillful and credible predictions are produced for some regions in western Ethiopia, particularly under a *clustered indirect* statistical approach.  At the regional scale, the approach shows promise for northwestern Ethiopia (Cluster 1, 3, 5, and 7),
5   particularly compared to current NMA operational forecasts, which are only moderately more skillful than climatology (Korecha and Sorteberg, 2013). The regional average RPSS in this study under the *clustered* case ranges from 10% to 33% for northwestern Ethiopia, as opposed to values under 6% for NMA operational forecast (Korecha and Sorteberg, 2013). The approach adopted here
10   also advances on previous studies (Gissila et al., 2004, Block and Rajagopalan, 2007, Korecha and Barnston, 2007, Diro et al., 2011b, Segele et al., 2015) by first applying an objective cluster analysis and then conditionally constructing high-resolution predictions. A unique set of predictors is applied to each cluster, which contributes to superior prediction performance at cluster levels in northwestern Ethiopia, as compared with predictions from the *non-clustered* approach. Grid-level prediction under the *clustered indirect* case also reduces the effect of over-fitting relative to the *direct* case and improves negative RPSS values to near
15   or above zeros; that said, the *non-clustered direct* case also illustrates higher correlation and RPSS values on average.

Two out of 10 NMME dynamical models, NASA-GMAO and NCEP-CFSv2, demonstrate overall superior performance to the statistical models; however, for certain regions such as Cluster 5, the performance of statistical models under *clustered indirect* and *non-clustered direct* cases is still superior. It is also worth noting that the statistical model predictions are at a one hundred
20   times finer spatial resolution than the dynamical models providing additional advantages at the local scale, when skillful. Nevertheless, improvements in dynamical models continue and their application to seasonal precipitation prediction is likely to grow (e.g. Palmer et al., 2004, Saha et al., 2006, Lim et al., 2009).

Relatively poor prediction performance is evident in some locations such as southwestern Ethiopia and regions outside Ethiopia,
25   where the hydroclimatic processes that produce precipitation might be driven by local factors or other regional climate patterns rather than large-scale climate variables identified in this study. A previous study (Zhang et al., 2016) has shown that the influence of ENSO on JJAS precipitation in western Ethiopia decreases generally from north to south, and is likely one of the reasons why skills are relatively low in southwestern Ethiopia. Cluster 5 was also identified with the strongest connection to equatorial Pacific SST (Zhang et al., 2016), which is consistent with the highest skill found in this study. Other regions with low prediction skill
30   show relatively strong connections to SST in neighboring oceanic regions. However, connections with those climate patterns appear to be less robust than with ENSO, making the predictions in their associated regions less skillful. This is also consistent with the findings from other studies that even though all three oceans (Indian, Atlantic, and Pacific Ocean) affect the JJAS precipitation in western Ethiopia, the Pacific Ocean still plays the greatest role (Segele et al., 2009, Omondi et al., 2013).

35   The southwest Asian monsoon over Indian Ocean may also be critical in determining the precipitation, given that the clusters with better prediction skills lie along the pathway of the monsoon. Based on the global concurrent correlation maps between JJAS precipitation and SLP for each cluster, Cluster 5 and 7 – the two clusters with the best skills – are the only ones that are strongly and negatively  correlated with SLP near Bombay, and meantime strongly and positively correlated with the SLP at the eastern equatorial Pacific Ocean. The former indicates that a strong southwest Asian monsoon is associated with higher JJAS
40   precipitation amount, and vice versa. The latter indicates that a high surface pressure over the eastern equatorial Pacific Ocean

often accompanied with cold SST and a raining pattern – a La Nina phenomenon – also brings higher JJAS precipitation to the western Ethiopia, and vice versa. Cluster 2 – one of the worst predicted clusters – shows moderately strong negative correlation with SLP near Bombay; however, it is also correlated strongly and negatively with SLP in the southern Indian Ocean (a high pressure system that drives the monsoon toward the low pressure system near Bombay), indicating that high JJAS precipitation in

5     Cluster 2 is not necessarily associated with a strong southwest Asian monsoon.  Moreover, its correlation with SLP over equatorial Pacific Ocean is nonsignificant. Considering in general El Nino suppresses the monsoon and La Nina increases it (Kumar et al., 2006), strong correlations with both ENSO and the monsoon in the correct direction, such as for Cluster 5 and 7,

10     indicate a double insurance over their association with the southwest Asian monsoon. Therefore, clusters which are more affected by the southwest Asian monsoon over Indian Ocean, particularly coupled with the influence of ENSO, are likely to show more promises in their prediction skills.

Additional prediction features also warrant future attention, including longer prediction lead times and evaluation of other relevant

15     characteristics (e.g. intra-seasonal dry spells, seasonal onset or cessation, etc.). As observational datasets continue to grow, data-driven cluster analyse~i~s and statistical modeling approaches may be expected to improve. Careful analysis of possible significant trends in the data is also warranted; a region with a relatively high correlation may be selected solely based on trends in predictors and observations. For shorter time series, such as the data used in this study, trend analysis may not be reliable; detrending can also reduce evidence of large-scale decadal climate signals.

20      Improving predictive capabilities may not be a complete panacea, but it can continue to be an important part of a decisions-maker's portfolio as they cope with hydroclimatic variability and its inherent risks.

**6 Data availability**

The National Centers for Environmental Prediction and National Center for Atmospheric Research (NCEP/NCAR) reanalysis

25     dataset can be accessed through the National Oceanic & Atmospheric Administration (NOAA) Earth System Research Laboratory (ESRL) website (https://www.esrl.noaa.gov/psd/data/reanalysis/).

The NMME hindcasts are available through the International Research Institute for Climate and Society (IRI) website (http://iridl.ldeo.columbia.edu/SOURCES/.Models/.NMME/).

30

The gridded precipitation dataset in western Ethiopia is available upon request from NMA (http://www.ethiomet.gov.et/).

**Competing interests**

The authors declare that they have no conflict of interest.

**Acknowledgements**

This study was supported by NASA Project NNX14AD30G and NSF PIRE Project 1545874. We acknowledge the National Meteorological Agency of Ethiopia for sharing data. We also want to thank the reviewers for their suggestions in improving this work.